# Improved Best-of-Both-Worlds Guarantees for Multi-Armed Bandits: FTRL with General Regularizers and Multiple Optimal Arms

**Tiancheng Jin** [*]
University of Southern California
tiancheng.jin@usc.edu

**Junyan Liu** [*]
University of California, San Diego
jul037@ucsd.edu

**Haipeng Luo**
University of Southern California
haipengl@usc.edu

## Abstract

We study the problem of designing adaptive multi-armed bandit algorithms that perform optimally in both the stochastic setting and the adversarial setting simultaneously (often known as a best-of-both-world guarantee). A line of recent works shows that when configured and analyzed properly, the Follow-the-Regularized-Leader (FTRL) algorithm, originally designed for the adversarial setting, can in fact optimally adapt to the stochastic setting as well. Such results, however, critically rely on an assumption that there exists one unique optimal arm. Recently, Ito [2021] took the first step to remove such an undesirable uniqueness assumption for one particular FTRL algorithm with the $1/2$-Tsallis entropy regularizer. In this work, we significantly improve and generalize this result, showing that uniqueness is unnecessary for FTRL with a broad family of regularizers and a new learning rate schedule. For some regularizers, our regret bounds also improve upon prior results even when uniqueness holds. We further provide an application of our results to the decoupled exploration and exploitation problem, demonstrating that our techniques are broadly applicable.

## 1 Introduction

We study the problem of multi-armed bandits (MAB) where a learner sequentially interacts with an environment for $T$ rounds. In each round, the learner selects one of the $K$ arms and observes its loss. The goal of the learner is to minimize her regret, which measures the difference between her total loss and that of the best fixed arm in hindsight. Depending on how the losses are generated, two settings have been heavily studied in the literature: the *stochastic* setting, where the loss of each arm at each round is an i.i.d. sample of a fixed and unknown distribution, and the *adversarial* setting, where the losses can be arbitrarily decided by an adversary. In the stochastic setting, the UCB algorithm [Lai and Robbins, 1985, Auer et al., 2002a] attains the instance-optimal regret $\mathcal{O}(\sum_{i:\Delta_i>0} \frac{\log T}{\Delta_i})$, where the sub-optimality gap $\Delta_i$ is the difference between the expected loss of arm $i$ and that of the optimal arm. On the other hand, in the adversarial setting, the minimax-optimal regret is known to be of order $\Theta(\sqrt{KT})$ [Auer et al., 2002b, Audibert and Bubeck, 2009], achieved via the well-known Follow-the-Regularized-Leader (FTRL) framework.

---

[*]Equal contribution, in alphabetical order.

37th Conference on Neural Information Processing Systems (NeurIPS 2023).

Given the rather different algorithmic ideas in UCB and FTRL, it is natural to ask whether there exists an adaptive algorithm that achieves the *best of both worlds* (BOBW), simultaneously enjoying the instance-optimal $(\log T)$-regret in the stochastic setting and minimax-optimal $\sqrt{T}$-regret in the adversarial setting. This question was first answered affirmatively in [Bubeck and Slivkins, 2012], followed by a sequence of improvements and extensions in the past decade. Among these works, a somewhat surprising result by Wei and Luo [2018] shows that, when configured and analyzed properly, FTRL, an algorithm originally designed for the adversarial setting to achieve $\sqrt{T}$-type regret, in fact can also achieve $(\log T)$-type regret in the stochastic setting automatically. This result was latter significantly improved to optimal by Zimmert and Seldin [2019, 2021] using the $1/2$-Tsallis entropy regularizer and extended to many other problems. In fact, these algorithms not only achieve BOBW, but also automatically adapt to intermediate settings with a regret bound that interpolates smoothly between the two extremes.

A key drawback of such FTRL-based approaches, however, is that their analysis for the stochastic setting critically relies on a uniqueness assumption, that is, there exists one unique optimal arm (with $\Delta_i = 0$). A recent work by Ito [2021] took the first step to address this issue and proposed a novel analysis showing that the exact same Tsallis-INF algorithm of [Zimmert and Seldin, 2019] in fact works even without this assumption. Unfortunately, his analysis is specific to the $1/2$-Tsallis entropy regularizer with an arm-independent learning rate, and it is highly unclear how to extend it to other regularizers which often require an arm-dependent learning rate. For example, extending it to the log-barrier regularizer was explicitly mentioned as an open problem in [Ito, 2021].

In this work, we significantly improve and generalize the analysis of [Ito, 2021], greatly deepening our understanding on using FTRL to achieve BOBW. Our improved analysis allows us to obtain a suite of new results, all achieved *without* the uniqueness assumption. Specifically, we consider a new and unified arm-dependent learning rate schedule and the following regularizers (plus a small amount of extra log-barrier); see also Table 1 for a summary.

- Log-barrier: with our new learning rate schedule, we show a regret bound of order $\mathcal{O}\left(\mathbb{G}\log T\right)$ for the stochastic setting and $\mathcal{O}\left(\sqrt{KT\log T}\right)$ for the adversarial setting. Here, $\mathbb{G} = \frac{|U|}{\Delta_{\text{MIN}}} + \sum_{i \in V}\frac{1}{\Delta_i}$ measures the difficulty of the instance, with $U = \{i : \Delta_i = 0\}$ being the set of optimal-arms, $V$ being the set of all remaining sub-optimal arms, and $\Delta_{\text{MIN}} = \min_{i \in V}\Delta_i$ being the minimum non-zero sub-optimality gap. Our bound for the stochastic setting improves those in [Wei and Luo, 2018, Ito, 2021], both of which require the uniqueness assumption.

- Shannon entropy: we show a regret bound of order $\mathcal{O}\left(\mathbb{G}(\log T)^2\right)$ for the stochastic setting and $\mathcal{O}(\sqrt{KT}\log T)$ for the adversarial setting. This improves [Ito et al., 2022] (when applying their more general results to MAB) in two ways: first, their result requires the uniqueness assumption while ours does not; second, their $\mathbb{G}$ is defined as $\frac{K}{\Delta_{\text{MIN}}}$, strictly larger than ours.

- $\beta$-Tsallis entropy: we also consider Tsallis entropy with a general parameter $\beta \in (0,1)$, and show a regret bound of order $\mathcal{O}\left(\frac{\mathbb{G}\log T}{\beta(1-\beta)}\right)$ for the stochastic setting and $\mathcal{O}\left(\sqrt{\frac{KT(\log T)^{\mathbb{I}\{\beta \neq 1/2\}}}{\beta(1-\beta)}}\right)$ for the adversarial setting. The only prior work that uses $\beta$-Tsallis entropy for BOBW in MAB is [Zimmert and Seldin, 2021], but their algorithm is *infeasible* (unless $\beta = 1/2$) since the learning rate is tuned in terms of the unknown sub-optimality gaps. We not only address this issue with our new learning rate schedule, but also remove the uniqueness assumption. While $\beta = 1/2$ leads to be best bounds in MAB, following [Rouyer and Seldin, 2020], we showcase the importance of other values of $\beta$ (specifically, $\beta = 2/3$) in the so-called Decoupled Exploration and Exploitation (DEE-MAB) setting, and again significantly improve their results (see Table 2).

It is worth noting that, as it is common for FTRL-based approaches, our algorithms also automatically adapt to more general corrupted settings (or the so-called *adversarial regime with a self-bounding constraint* [Zimmert and Seldin, 2021]). The complete statements of our results can be found in Section 3.

**Techniques**    Inspired by [Ito, 2021], we decompose the regret into three parts: the regret related to the sub-optimal arms, the regret related to the optimal arms, and the residual regret. Bounding each of them requires new ideas, as discussed below (see Section 4 for details).

Table 1: Overview of our BOBW results for MAB, all achieved via Algorithm 1 and a unified learning rate $\gamma_i^t = \theta \sqrt{1 + \sum_{\tau < t} \left( \max \{ p_i^\tau, 1/T \} \right)^{1-2\alpha}}$ for arm $i$ in round $t$, with $p_i^\tau$ being the probability of picking arm $i$ in round $\tau$ and the values of $\theta$ and $\alpha$ specified in the table. "Sto." and "Adv." denote respectively the stochastic and the adversarial setting. $\mathbb{G}$ is defined as $\frac{|U|}{\Delta_{\text{MIN}}} + \sum_{i \in V} \frac{1}{\Delta_i}$, where $U$ is the set of optimal-arms, $V$ is the set of sub-optimal arms, and $\Delta_{\text{MIN}} = \min_{i \in V} \Delta_i$.

| Regularizer[a] | $\alpha, \theta$ | Regret (w/o uniqueness) | Comments |
|---|---|---|---|
| Log-barrier $-\sum_i \gamma_i^t \log p_i$ | $\alpha = 0$ $\theta = \sqrt{\frac{1}{\log T}}$ | Sto. $\mathcal{O}\left( \mathbb{G} \log T \right)$ Adv. $\mathcal{O}\left( \sqrt{KT \log T} \right)$ | First to remove uniqueness for log-barrier |
| $\beta$-Tsallis entropy $-\frac{1}{1-\beta} \sum_i \gamma_i^t p_i^\beta$ | $\alpha = \beta$ $\theta = \sqrt{\frac{1-\beta}{\beta}}$ | Sto. $\mathcal{O}\left( \frac{\mathbb{G} \log T}{\beta(1-\beta)} \right)$ Adv. $\mathcal{O}\left( \sqrt{\frac{KT(\log T)^{\mathbb{I}\{\beta \neq 1/2\}}}{\beta(1-\beta)}} \right)$ | First BOBW result for $\beta \neq 1/2$ (even with uniqueness) |
| Shannon entropy $\sum_i \gamma_i^t p_i \log \left( \frac{p_i}{e} \right)$ | $\alpha = 1$ $\theta = \sqrt{\frac{1}{\log T}}$ | Sto. $\mathcal{O}\left( \mathbb{G} (\log T)^2 \right)$ Adv. $\mathcal{O}\left( \sqrt{KT} \log T \right)$ | Improve [Ito et al., 2022] which defines $\mathbb{G}$ as $K/\Delta_{\text{MIN}}$ and requires uniqueness[b] |

[a]To be more precise, a small amount of extra log-barrier has been omitted in this table for simplicity.

[b]To be clear, the setting in [Ito et al., 2022] is more general than MAB. Here, the comparison is solely based on their results specified to MAB.

Table 2: Regret bounds of our algorithm using $2/3$-Tsallis entropy for the Decoupled Exploration and Exploitation MAB problem. $V$ is the set of sub-optimal arms and $\Delta_{\text{MIN}} = \min_{i \in V} \Delta_i$.

| Ours (w/o uniqueness) | [Rouyer and Seldin, 2020] (w/ uniqueness) |
|---|---|
| Sto. $\mathcal{O}\left( \sqrt{\sum_{i \in V} \frac{K}{\Delta_i^2}} \right)$ Adv. $\mathcal{O}(\sqrt{KT})$ | Sto. $\mathcal{O}\left( \sqrt{\sum_{i \in V} \frac{K}{\Delta_i \cdot \Delta_{\text{MIN}}}} \right)$ Adv. $\mathcal{O}(\sqrt{KT})$ |

To bound the regret related to the sub-optimal arms by a so-called self-bounding quantity (the key to achieve BOBW using FTRL), we design a novel arm-dependent learning rate schedule (which is also our key algorithmic contribution). For example, when using $\beta$-Tsallis entropy, this schedule balances the corresponding stability term and penalty term of a sub-optimal arm $i$ to $\mathcal{O}\left( \sum_{t=1}^{T} (p_i^t)^{1-\beta} / \gamma_i^{t+1} \right)$, which is then bounded by a self-bounding quantity. Apart from removing the uniqueness assumption, as mentioned this learning rate schedule also enables us to achieve the first BOBW guarantees for $\beta$-Tsallis entropy with any value of $\beta$, and also to improve the bound of Ito et al. [2022] for Shannon entropy, which are notable results on their own.

Then, to bound the regret related to the optimal arms, we greatly extend the idea of [Ito, 2021] that is highly specific to the simple form of $1/2$-Tsallis entropy with an arm-independent learning rate. Specifically, we develop a new analysis based on a key observation of a certain monotonicity of Bregman divergences. Such monotonicity only requires two mild conditions on the regularizer that are usually satisfied, allowing us to apply it to a broad spectrum of regularizers.

Our arm-dependent learning rate does make the residual regret much more complicated compared to [Ito, 2021]. To handle it, we carefully consider two cases and show that in both cases it can be related to some self-bounding quantities.

Finally, we note that various places of our analysis require the learner's distribution over arms to be stable in a multiplicative sense between two consecutive rounds. We achieve this by adding an extra small amount of log-barrier, a technique first proposed in [Bubeck et al., 2018]. While we do not know how to prove the same results without this extra tweak, we conjecture that it is indeed unnecessary.

**Related work**   For early results solely for the stochastic setting or solely for the adversarial setting, we refer the readers to the systematic survey in [Lattimore and Szepesvári, 2020]. The study of BOBW for MAB starts from the pioneering work of Bubeck and Slivkins [2012], followed by many improvements via different approaches [Seldin and Slivkins, 2014, Auer and Chiang, 2016, Seldin and Lugosi, 2017, Wei and Luo, 2018, Lykouris et al., 2018, Gupta et al., 2019, Zimmert and Seldin, 2019, 2021] and many extensions from MAB to other problems such as semi-bandits [Zimmert et al., 2019], linear bandits [Lee et al., 2021], MAB with feedback graphs [Ito et al., 2022, Erez and Koren, 2021, Rouyer et al., 2022], MAB with switching cost [Rouyer et al., 2021, Amir et al., 2022], model-selection [Pacchiano et al., 2022], partial monitoring [Tsuchiya et al., 2023], and Markov Decision Process (MDP) [Lykouris et al., 2019, Jin and Luo, 2020, Jin et al., 2021, Chen et al., 2021]. Among these works, the FTRL-based approach is particularly appealing since it is simple in both the algorithm design and the analysis, and also extends seamlessly to other more general settings (such as the corrupted setting). The uniqueness assumption used to be critical for the analysis of this approach, but plays no role in other methods such as [Auer and Chiang, 2016, Seldin and Lugosi, 2017]. Following the first step by [Ito, 2021], our work further demonstrates that this was merely due to the lack of a better analysis (and sometimes a better learning rate schedule). We believe that our techniques shed light on removing the uniqueness assumption for using FTRL in more complicated problems such as semi-bandits and MDPs.

## 2   Preliminaries

In multi-armed bandits (MAB), a learner is given a fixed set of arms $[K] = \{1, 2, \cdots, K\}$ and has to interact with an environment for $T \geq K$ rounds. In each round $t$, the learner chooses an arm $i^t \in [K]$ while simultaneously the environment decides a loss vector $\ell^t \in [0, 1]^K$. The learner then suffers and observes the loss $\ell^t_{i^t}$ of the selected arm for this round. The goal of the learner is to minimize her (pseudo) regret, which measures the difference between her expected cumulative loss and that of the best arm in hindsight. Formally, the regret is defined as $\text{Reg}^T = \mathbb{E}\left[\sum_{t=1}^T \ell^t_{i^t} - \sum_{t=1}^T \ell^t_{i^\star}\right]$, where $i^\star \in \text{argmin}_{i \in [K]} \mathbb{E}\left[\sum_{t=1}^T \ell^t_i\right]$ is one of the best arms in hindsight, and $\mathbb{E}[\cdot]$ denotes the expectation with respect to the internal randomness of both the algorithm and the environment.

**Adversarial setting versus stochastic setting**   We consider two different settings according to how the loss vectors are decided by the environment. In the adversarial setting, the environment decides the loss vectors in an arbitrary way with the knowledge of the learner's algorithm. In this case, the minimax optimal regret is known to be $\Theta(\sqrt{KT})$ [Audibert and Bubeck, 2009].

In the stochastic setting, following prior work such as [Zimmert and Seldin, 2021], we consider a situation much more general than the vanilla i.i.d. case (sometimes called the adversarial regime with a self-bounding constraint). Formally, we assume that the loss vectors satisfy the following condition: there exists a gap vector $\Delta \in [0, 1]^K$ and a constant $C \geq 0$ such that

$$\text{Reg}^T \geq \mathbb{E}\left[\sum_{t=1}^T \sum_{i \in [K]} \Pr\left[i^t = i\right] \Delta_i\right] - C, \tag{1}$$

where $\Pr[i^t = i]$ denotes the learner's probability of taking arm $i$ in round $t$. This condition subsumes the well-studied i.i.d. setting (discussed in Section 1) where the loss vectors are independently sampled from a fixed but unknown distribution and thus Condition (1) holds with equality, $C = 0$, and $\Delta_i = \mathbb{E}[\ell^t_i - \ell^t_{i^\star}]$ being the sub-optimality gap of arm $i$ (independent of $t$). In this case, the instance-optimal regret is $\mathcal{O}(\sum_{i:\Delta_i > 0} \frac{\log T}{\Delta_i})$, achieved by the UCB algorithm [Auer et al., 2002a]. More generally, Condition (1) covers the corrupted i.i.d. setting where the loss vectors are first sampled from a fixed distribution and then corrupted by an adversary in an arbitrary way as long as the expected cumulative $\ell_\infty$ distance between the corrupted loss vector and the original one is bounded by $C \in [0, T]$. While these two examples both involve iidness, note that Condition (1) itself is much more general and does not necessarily require that.

**Uniqueness assumption**   Prior work using FTRL to achieve BOBW crucially relies on a uniqueness assumption when analyzing the regret under Condition (1). Specifically, it is assumed that there exists one and only one arm $\mathring{i}$ with $\Delta_{\mathring{i}} = 0$. In the special i.i.d. case, this simply means that there

---

**Algorithm 1** FTRL for BOBW without Uniqueness

---

**Input**: coefficient $\theta$, learning rate $\alpha$, Tsallis entropy parameter $\beta$, log-barrier coefficient $C_{\text{LOG}}$, number of arms $K$, number of rounds $T$ (not necessary if a doubling trick is applied; see [Ito, 2021, Section 5.3])

**for** $t = 1, 2, \ldots, T$ **do**

$\quad$ Define regularizer $\phi^t(p) = \underbrace{-C_{\text{LOG}} \sum_{i \in [K]} \log p_i}_{\text{extra log-barrier}} + \begin{cases} -\sum_{i \in [K]} \gamma_i^t \log p_i, & \text{(log-barrier)} \\ -\frac{1}{1-\beta} \sum_{i \in [K]} \gamma_i^t p_i^\beta, & \text{($\beta$-Tsallis entropy)} \\ \sum_{i \in [K]} \gamma_i^t p_i \log(p_i/e), & \text{(Shannon entropy)} \end{cases}$

$\quad$ with learning rate $\gamma_i^t = \theta \sqrt{1 + \sum_{\tau < t} \left( \max \{p_i^\tau, 1/T\} \right)^{1-2\alpha}}$.

$\quad$ Compute $p^t = \operatorname{argmin}_{p \in \Omega_K} \left\{ \left\langle p, \sum_{\tau < t} \widehat{\ell}^\tau \right\rangle + \phi^t(p) \right\}$ where $\Omega_K$ is the simplex.

$\quad$ Draw arm $i^t \sim p^t$, suffer and observe loss $\ell_{i^t}^t$.

$\quad$ Construct $\widehat{\ell}^t$ as an unbiased estimator of $\ell_t$ with $\widehat{\ell}_i^t = \frac{\mathbb{I}\{i^t = i\} \ell_i^t}{p_i^t}$, $\forall i \in [K]$.

---

exists a unique optimal arm ($\mathring{i} = i^\star$). Under this uniqueness assumption, the Tsallis-INF algorithm of [Zimmert and Seldin, 2021] achieves $\text{Reg}^T = \mathcal{O} \left( \sum_{i \neq \mathring{i}} \frac{\log T}{\Delta_i} + \sqrt{C \sum_{i \neq \mathring{i}} \frac{\log T}{\Delta_i}} \right)$. The recent work [Ito, 2021] takes the first step to remove such an assumption for the Tsallis-INF algorithm and develops a refined analysis with regret bound $\mathcal{O} \left( \sum_{i : \Delta_i > 0} \frac{\log T}{\Delta_i} + \sqrt{C \sum_{i : \Delta_i > 0} \frac{\log T}{\Delta_i}} + D + K \right)$, where $D$ is such that $\mathbb{E} \left[ \sum_{t=1}^{T} \max_{i : \Delta_i = 0} \mathbb{E}^t \left[ \ell_i^t - \ell_{i^\star}^t \right] \right] \leq D$, and $\mathbb{E}^t[\cdot]$ is the conditional expectation with respect to the history before round $t$. Note that, in the i.i.d. setting, $D$ is simply 0, while in the corrupted setting, $D$ is at most $C$. We significantly generalize and improve this result to a borad family of algorithms, and all our results hold *without* the uniqueness assumption.

We denote by $U = \{i \in [K] : \Delta_i = 0\}$ the set of arms with a zero gap, and $V = [K] \backslash U$ the set of arms with a positive gap. In the i.i.d. setting, $U$ is simply the set of optimal arms and $V$ is the set of sub-optimal arms. We also define $\Delta_{\text{MIN}} = \min_{i \in V} \Delta_i$ to be the minimum nonzero gap.

## 3 Algorithms and Results

The pseudocode of our algorithm is presented in Algorithm 1. It is based on the general FTRL framework, which finds $p^t$, the distribution of selecting arms in around $t$, via solving the optimization problem $p^t = \operatorname{argmin}_{p \in \Omega_K} \left\langle p, \sum_{\tau < t} \widehat{\ell}^\tau \right\rangle + \phi^t(p)$. Here, $\Omega_K$ is the set of all possible distributions over $K$ arms, $\widehat{\ell}^\tau$ is an loss estimator for $\ell^\tau$, and $\phi^t$ is a regularizer. The learner then samples arm $i^t$ from the distribution $p^t$ and observes the suffered loss $\ell_{i^t}^t$. With this feedback, the algorithm constructs the standard unbiased importance-weighted loss estimator: $\widehat{\ell}_i^t = \frac{\mathbb{I}\{i^t = i\} \ell_i^t}{p_i^t}$, $\forall i \in [K]$, where $\mathbb{I}\{\cdot\}$ denotes the indicator function.

We consider three different reugularizers $\phi^t$: log-barrier, $\beta$-Tsallis entropy, and Shannon entropy; see Algorithm 1 for definitions. By now, they are standard reugularizers used extensively in the MAB literature, each with different useful properties, but there are some small tweaks in our definitions: 1) a linear term (from $p_i \log(p_i/e) = p_i \log p_i - p_i$) is added to the canonical form of Shannon entropy, which is critical to ensure a certain type of monotonicity of Bregman divergences (see Section 4); 2) for technical reasons, we also incorporate a small amount of extra log-barrier (with coefficient $C_{\text{LOG}}$), which ensures multiplicative stability of the algorithm.

More importantly, we propose the following unified arm-dependent learning rate:

$$\gamma_i^t = \theta \sqrt{1 + \sum_{\tau=1}^{t-1} \left( \max \{p_i^\tau, 1/T\} \right)^{1-2\alpha}}, \quad \forall i \in [K], \, t \in [T], \tag{2}$$

where $\alpha \in [0,1]$ and $\theta \in \mathbb{R}_{>0}$ are parameters (set differently for different regularizers). The clipping of $p_i^\tau$ to $1/T$ is because $\left(p_i^t\right)^{1-2\alpha}$ itself could be unbounded for $\alpha \in (1/2, 1]$ when $p_i^t$ is too small. This learning rate is not only conceptually simpler than those in [Ito, 2021, Ito et al., 2022] and important for removing the uniqueness assumption, but also leads to better bounds in some cases as we discuss below.

**Main results**  We now present our main results. Our regret bounds in the stochastic setting are expressed in terms of an instance complexity measure $\mathbb{G} = \frac{|U|}{\Delta_{\text{MIN}}} + \sum_{i \in V} \frac{1}{\Delta_i}$, which is of the same order as the standard complexity measure $\sum_{i \in V} \frac{1}{\Delta_i}$ when $|U| = \mathcal{O}(1)$ (in particular, this is the case when the uniqueness assumption holds). Similar to [Ito, 2021], our bounds are also in terms of the constant $D$ defined in Section 2. We start with the following result for the log-barrier regularizer.

**Theorem 3.1.** *When using the log-barrier regularizer with $C_{\text{LOG}} = 162$, $\alpha = 0$, and $\theta = \sqrt{1/\log T}$, Algorithm 1 ensures $\text{Reg}^T = \mathcal{O}\left(\sqrt{KT \log T}\right)$ always, and simultaneously the following regret bound when Condition (1) holds: $\text{Reg}^T = \mathcal{O}\left(\mathbb{G} \log T + \sqrt{C\mathbb{G} \log T} + \frac{K^2}{\sqrt{\log T}} + K \log T + D\right)$.*

Log-barrier was first used to achieve BOBW in [Wei and Luo, 2018] and later improved in [Ito, 2021], both of which require the uniqueness assumption. The $\mathcal{O}\left(\sqrt{KT \log T}\right)$ bound for the adversarial setting is almost minimax optimal except for the extra $\sqrt{\log T}$ factor (a common caveat for log-barrier). On the other hand, the bound under Condition (1) matches that of [Ito, 2021] when uniqueness holds and generalizes it otherwise.[2] It is worth noting that this bound (and the same for our other results) suffers an $\mathcal{O}(D)$ term, which unfortunately can be as large as $C$, making the bound weaker than those always with only $\sqrt{C}$ dependence under the uniqueness assumption. It is unclear to us whether such $\mathcal{O}(D)$ dependence is necessary when we do not make the uniqueness assumption.

Next, we present our results for Shannon entropy.

**Theorem 3.2.** *When using the Shannon entropy regularizer with $C_{\text{LOG}} = 162 \log K$, $\alpha = 1$, and $\theta = \sqrt{1/\log T}$, Algorithm 1 ensures $\text{Reg}^T = \mathcal{O}\left(\sqrt{KT} \log T\right)$ always, and simultaneously the following regret bound under Condition (1): $\text{Reg}^T = \mathcal{O}\left(\mathbb{G} \left(\log T\right)^2 + \sqrt{C\mathbb{G} \left(\log T\right)^2} + K^2 \log^{3/2} T + D\right)$.*

The recent work [Ito et al., 2022] is the first to discover that Shannon entropy, used in the very first adversarial MAB algorithm EXP3 [Auer et al., 2002b], in fact also achieves BOBW when configured and analyzed properly (assuming uniqueness). Their results are for the more general setting of MAB with a feedback graph, and when specified to standard MAB, their regret bound under Condition (1) is worse than ours with $\mathbb{G}$ defined as $K/\Delta_{\text{MIN}}$. The key of our improvement comes from our very different and arm-dependent learning rate schedule. Note that there are extra $\log T$ factors in the regret for both the adversarial setting and the stochastic setting, which is also the case in [Ito et al., 2022]. While this makes the bounds worse compared to other regularizers, in the more general setting with a feedback graph, Shannon entropy is known to be critical for achieving the right dependence on the independence number of the feedback graph, and we believe that our results shed light on how to remove the uniqueness requirement in this more general setting using Shannon entropy.

Finally, we present our results for Tsallis entropy.

**Theorem 3.3.** *For any $\beta \in (0,1)$, when using the $\beta$-Tsallis entropy regularizer with $C_{\text{LOG}} = \frac{162\beta}{1-\beta}$, $\alpha = \beta$, and $\theta = \sqrt{(1-\beta)/\beta}$, Algorithm 1 ensures $\text{Reg}^T = \mathcal{O}\left(\sqrt{\frac{1}{\beta(1-\beta)} KT \left(\log T\right)^{\mathbb{I}\left\{\beta \neq \frac{1}{2}\right\}}}\right)$ always, and simultaneously the following regret bound under Condition (1): $\text{Reg}^T = \mathcal{O}\left(\frac{\mathbb{G} \log T}{\beta(1-\beta)} + \sqrt{\frac{C\mathbb{G} \log T}{\beta(1-\beta)}} + D + \frac{K^2 \sqrt{\beta}}{(1-\beta)^{3/2}} + \frac{\beta K \log T}{1-\beta}\right)$.*

When $\beta = 1/2$, our learning rate $\gamma_i^t$ simply becomes $\sqrt{t}$ (which is arm-independent), and our algorithm exactly recovers Tsallis-INF [Zimmert and Seldin, 2021]. In this case, our result is essentially the same as what the improved analysis of [Ito, 2021] shows, which does not require

---

[2]Ito [2021] also provides other data-dependent bounds in the adversarial setting, which we do not consider here. Ignoring this part, his algorithm is also slightly different from ours, but we note in passing that our analysis technique also applies if one sets $\nu_i^t = p_i^t$ in his algorithm.

uniqueness. For $\beta \neq 1/2$, while such regularizers were also analyzed in [Zimmert and Seldin, 2021] (under uniqueness), their algorithm is infeasible since the learning rates are tuned based on the unknown $\Delta_i$'s. On the other hand, our algorithm not only uses a simple and feasible learning rate schedule, but also works without uniqueness. The bound for the adversarial setting has an extra $\sqrt{\log T}$ factor when $\beta \neq 1/2$ though, which we conjecture can be removed (as it is the case when using a fixed learning rate [Audibert and Bubeck, 2009, Abernethy et al., 2015]); see Remark C.2.2. We find it surprising that our learning rate exhibits totally different behavior when $\beta < 1/2$ versus when $\beta > 1/2$ (recall that $\alpha$ is set to $\beta$): in the former, $\gamma_i^t$ increases in the previous $p_i^\tau$ ($\tau < t$), while in the latter, it decreases in $p_i^\tau$.

It might not be clear at this point what the value is to consider $\beta \neq 1/2$ — after all, our bounds are minimized when $\beta = 1/2$. It turns out that, however, other values of $\beta$ play important roles in other problems, as for example demonstrated by Rouyer and Seldin [2020] in a decoupled exploration and exploitation setting. Below, we generalize our results to this setting, showcasing the broad applicability of our techniques.

**Decoupled exploration and exploitation** The Decoupled Exploration and Exploitation MAB (DEE-MAB) problem, first considered in [Avner et al., 2012], is a variant of MAB where in each round $t$, the learner picks an arm $i^t$ to exploit and an arm $j^t$ to explore, and then suffers the loss $\ell_{i^t}^t$ while observing the feedback $\ell_{j^t}^t$. The performance of the learner is still measured by the same regret definition in terms of the exploitation arms $i^1, \ldots, i^T$. The standard MAB can be seen as a special case where $i^t$ and $j^t$ must be the same. In DEE-MAB, it turns out that the adversarial setting is as difficult as standard MAB with a lower bound $\Omega(\sqrt{KT})$, but one can do much better in the stochastic setting with a $T$-*independent* regret bound. For example, [Rouyer and Seldin, 2020] uses FTRL with $2/3$-Tsallis entropy to achieve $\mathcal{O}(\sqrt{KT})$ in the adversarial setting and simultaneously $\mathcal{O}\left(\sqrt{\sum_{i \in V} \frac{K}{\Delta_i \cdot \Delta_{\mathrm{MIN}}}}\right)$ in the i.i.d. setting assuming a unique optimal arm.

Using our techniques, we not only remove the uniqueness requirement, but also improve their bounds. Specifically, we consider the exact same algorithm as theirs, which can be described using the framework of Algorithm 1: take $C_{\mathrm{LOG}} = 0$, $\theta = K^{1/6}$, $\alpha = 1/2$, and $\beta = 2/3$ for the Tsallis entropy regularizer; sample the exploitation arm $i^t$ according to $p^t$ as before and the exploration arm $j^t$ according to a different distribution $g^t$ with $g_i^t \propto \left(p_i^t\right)^{2/3}$; finally, construct the importance-weighted estimator using the exploration information: $\widehat{\ell}_i^t = \frac{\mathbb{I}\{j^t = i\}\ell_i^t}{g_i^t}$. Our results are as follows.

**Theorem 3.4.** *For the DEE-MAB problem, the algorithm described above ensures* $\mathrm{Reg}^T = \mathcal{O}(\sqrt{KT})$ *always, and simultaneously the following regret bound under Condition (1):* $\mathrm{Reg}^T = \mathcal{O}\left(\sqrt{\sum_{i \in V} \frac{K}{\Delta_i^2}} + \sqrt{C} \cdot \left(\sum_{i \in V} \frac{K}{\Delta_i^2}\right)^{1/4} + K + D\right)$.

Note that in the i.i.d. setting (where $C = D = 0$), we improve their bound from $\mathcal{O}\left(\sqrt{\sum_{i \in V} \frac{K}{\Delta_i \cdot \Delta_{\mathrm{MIN}}}}\right)$ to $\mathcal{O}\left(\sqrt{\sum_{i \in V} \frac{K}{\Delta_i^2}}\right)$ (in addition to removing the uniqueness assumption).

## 4 Analysis

In this section, we take $\beta$-Tsallis entropy as an example to illustrate the key ideas of our analysis in proving the MAB results under Condition (1). As in all prior work, the analysis relies on a so-called *self-bounding* technique. Specifically, our goal is to bound the regret as follows (ignoring all minor terms, including the dependence on $\beta$):

$$\mathrm{Reg}^T \lesssim \mathbb{E}\left[\sum_{i \in V} \sqrt{(\log T)\sum_{t=1}^T p_i^t} + \sqrt{|U|(\log T)\sum_{i \in V}\sum_{t=1}^T p_i^t}\right], \tag{3}$$

where the two terms above enjoy a self-bounding property since they can be related back to the regret under Condition (1). To see this, we apply AM-GM inequality followed by Condition (1) to show the

following for any $z \geq 0$:

$$\mathbb{E}\left[\sum_{i \in V} \sqrt{\log T \sum_{t=1}^{T} p_i^t}\right] \leq \mathbb{E}\left[\sum_{i \in V}\left(\frac{\log T}{4z\Delta_i} + z \sum_{t=1}^{T} p_i^t \Delta_i\right)\right] \leq z\left(\text{Reg}^T + C\right) + \sum_{i \in V} \frac{\log T}{4z\Delta_i},$$

$$\mathbb{E}\left[\sqrt{|U| \log T \sum_{i \in V} \sum_{t=1}^{T} p_i^t}\right] \leq \mathbb{E}\left[\frac{|U| \log T}{4z\Delta_{\text{MIN}}} + z \sum_{i \in V} \sum_{t=1}^{T} p_i^t \Delta_{\text{MIN}}\right] \leq z\left(\text{Reg}^T + C\right) + \frac{|U| \log T}{4z\Delta_{\text{MIN}}}.$$

Rearranging and picking the optimal $z$ yields the regret bound under Condition (1) in Theorem 3.3.

The key is thus to prove Eq. (3), which is not that difficult under uniqueness (when $|V| = K - 1$) but turns out to be much more complicated without uniqueness. To proceed, we start with some key concepts and ideas from [Ito, 2021] which we follow. First, define the *skewed Bregman divergence* for two time steps $s, t \in \mathbb{N}$ as

$$D^{s,t}(x, y) = \phi^s(x) - \phi^t(y) - \langle \nabla \phi^t(y), x - y \rangle, \tag{4}$$

and its variant restricted to any subset $\mathcal{I} \subseteq [K]$ as $D_{\mathcal{I}}^{s,t}(x, y) = \phi_{\mathcal{I}}^s(x) - \phi_{\mathcal{I}}^t(y) - \langle \nabla \phi_{\mathcal{I}}^t(y), x - y \rangle$, where $\phi_{\mathcal{I}}^t(x) = -C_{\text{LOG}} \sum_{i \in \mathcal{I}} \log x_i - \frac{1}{1-\beta} \sum_{i \in \mathcal{I}} \gamma_i^t x_i^\beta$ (that is, $\phi^t$ restricted to $\mathcal{I}$). The standard Bregman divergence associated with $\phi^t$, which we denote by $D^t(x, y)$, is then a shorthand for $D^{t,t}(x, y)$. One key idea of [Ito, 2021] is to carefully choose the right benchmark for the algorithm — when there is a unique optimal arm, the benchmark basically has to be this unique optimal arm, but when multiple optimal arms exist, the benchmark can now be any distribution over these arms, and it can even be varying over time. Indeed, for round $t$, the following benchmark was used in [Ito, 2021]:

$$q^t = \underset{p \in \Omega_U}{\text{argmin}}\left\{\left\langle p, \sum_{\tau < t} \widehat{\ell}^\tau\right\rangle + \phi^t(p)\right\} = \underset{p \in \Omega_U}{\text{argmin}}\, D^t(p, p^t), \tag{5}$$

which follows the same definition of $p^t$ but is restricted to $\Omega_U = \left\{p \in \Omega_K : \sum_{i \in U} p_i = 1\right\}$, the set of distributions over the zero-gap arms. As the second equality shows, $q^t$ is also the projection of $p^t$ onto $\Omega_U$ w.r.t. the Bregman divergence $D^t$. With these time-varying benchmarks, Ito [Ito, 2021] proves

$$\text{Reg}^T \lesssim \mathbb{E}\left[\sum_{t=1}^{T} D^{t,t+1}(p^t, p^{t+1}) - D_U^{t,t+1}(q^t, q^{t+1})\right] + D. \tag{6}$$

The rest of the analysis is where we start to deviate from that of [Ito, 2021] (thought still largely inspired by it), which is critical for our algorithms that use arm-dependent learning rates. First, we introduce an important intermediate point $\bar{p}^{t+1} = \bar{p}_U^{t+1} + \bar{p}_V^{t+1}$ where $\bar{p}_U^{t+1}$ and $\bar{p}_V^{t+1}$ are defined as

$$\begin{aligned}
\bar{p}_U^{t+1} &= \underset{\substack{x \in \mathbb{R}_{\geq 0}^K, \ \sum_{i \in V} x_i = 0, \\ \sum_{i \in U} x_i = \sum_{i \in U} p_i^t}}{\text{argmin}} \left\langle x, \sum_{\tau \leq t} \widehat{\ell}^\tau\right\rangle + \phi_U^{t+1}(x), \\
\bar{p}_V^{t+1} &= \underset{\substack{x \in \mathbb{R}_{\geq 0}^K, \ \sum_{i \in U} x_i = 0, \\ \sum_{i \in V} x_i = \sum_{i \in V} p_i^t}}{\text{argmin}} \left\langle x, \sum_{\tau \leq t} \widehat{\ell}^\tau\right\rangle + \phi_V^{t+1}(x).
\end{aligned} \tag{7}$$

By definition, $\bar{p}^{t+1}$ is obtained from $p^t$ by redistributing the weights among arms in $U$ and those in $V$, in a way that minimizes an FTRL objective similar to that of $p^{t+1}$. We note that Ito [2021] also uses the same $\bar{p}_U^{t+1}$ is his analysis, but we introduce $\bar{p}_V^{t+1}$ (and thus $\bar{p}^{t+1}$) as well since it importantly allows us to decompose each Bregman divergence difference term in Eq. (6) as follows.

**Lemma 4.1.** *For any $t$, $D^{t,t+1}(p^t, p^{t+1}) - D_U^{t,t+1}(q^t, q^{t+1})$ is bounded by*

$$\underbrace{D_V^{t,t+1}(p^t, \bar{p}^{t+1})}_{\text{regret on sub-optimal arms}} + \underbrace{D_U^{t,t+1}(p^t, \bar{p}^{t+1}) - D_U^{t,t+1}(q^t, q^{t+1})}_{\text{regret on optimal arms}} + \underbrace{D^{t+1}(\bar{p}^{t+1}, p^{t+1})}_{\text{residual regret}}. \tag{8}$$

In the rest of this section, we proceed to bound each of the three terms in Eq. (44) (see also Table 3 for a summary of bounds for each of these terms and each of the three regularizers).

**Regret on Sub-Optimal Arms**  The regret related to the sub-optimal arms (or more formally arms in $V$) is the most straightforward to deal with, since our objective is to arrive at the self-bounding terms in Eq. (3) which are exactly only in terms of arms in $V$. Indeed, we can write this term as (with $p_V^t$ being the vector with the same value as $p^t$ for coordinates in $V$ and 0 for coordinates in $U$)

$$D_V^{t,t+1}\left(p^t, \bar{p}^{t+1}\right) = \underbrace{\left\langle p_V^t - \bar{p}_V^{t+1}, \widehat{\ell}^t \right\rangle - D_V^t\left(\bar{p}^{t+1}, p^t\right)}_{\text{stability}} + \underbrace{\phi_V^t(\bar{p}^{t+1}) - \phi_V^{t+1}(\bar{p}^{t+1})}_{\text{penalty}}, \qquad (9)$$

and then apply standard arguments to show that the stability term is of order $\sum_{i \in V} (p_i^t)^{1-\beta}/\beta\gamma_i^{t+1}$ while the penalty term is of order $\sum_{i \in V} (\gamma_i^{t+1} - \gamma_i^t)(p_i^t)^\beta/1-\beta$. In the Tsallis-INF algorithm, we have $\beta = 1/2$ and $\gamma_i^t = \sqrt{t}$, and thus the stability term and the penalty term are of the same order. This inspires us to design a learning rate for general $\beta$ with the same objective. Indeed, it can be verified that our particular learning rate schedule makes sure that the penalty term is of the same order as the stability term, meaning $D_V^{t,t+1}(p^t, \bar{p}^{t+1}) = \mathcal{O}\left(\sum_{i \in V} (p_i^t)^{1-\beta}/\beta\gamma_i^{t+1}\right)$. Further plugging in the learning rate and summing over $t$, we arrive at the following with $z_i^t = \max\{p_i^t, 1/T\}$:

$$\sum_{t=1}^T \sum_{i \in V} \frac{(p_i^t)^{1-\beta}}{\beta\gamma_i^t} \le \sqrt{\frac{1}{\beta(1-\beta)}} \sum_{i \in V} \sum_{t=1}^T \frac{(z_i^t)^{1-\beta}}{\sqrt{1 + \sum_{k=1}^t (z_i^k)^{1-2\beta}}}. \qquad (10)$$

Finally, applying the following technical lemma shows that the above is of the same order as the first term in our objective Eq. (3). More details can be found in Section A.2.

**Lemma 4.2.** *Let $\{x_t\}_{t=1}^T$ be a sequence with $x_t > 0$ for all $t$. Then, for any $\alpha \in [0, 1]$, we have*

$$\sum_{t=1}^T \frac{x_t^{1-\alpha}}{\sqrt{1 + \sum_{s=1}^t x_s^{1-2\alpha}}} \le \mathcal{O}\left(\sqrt{\left(\sum_{t=1}^T x_t\right) \log\left(1 + \sum_{t=1}^T x_t^{1-2\alpha}\right)}\right). \qquad (11)$$

**Regret on Optimal Arms**  Next, we show that the regret on optimal arms (or more formally arms in $U$), $D_U^{t,t+1}(p^t, \bar{p}^{t+1}) - D_U^{t,t+1}(q^t, q^{t+1})$, is nonpositive, which corresponds to the intuition that pulling optimal arms incur no regret. Ito [2021] proves something similar for $1/2$-Tsallis entropy via a certain monotonicity property of Bregman divergence, but his proof is highly specific to $1/2$-Tsallis entropy. Instead, we develop the following general monotonicity theorem which applies to a broad spectrum of regularizers as long as they satisfy two mild conditions.

**Theorem 4.3** (Monotonicity of Bregman divergence). *For any $t \in \mathbb{N}$, let $f^t : \Omega \to \mathbb{R}$ be a continuously-differentiable and strictly-convex function defined on $\Omega \subseteq \mathbb{R}$. Suppose that the following two conditions hold for all $z \in \Omega$: (i) $(f^t)'(z)$ is differentiable and concave; (ii) $(f^{t+1})'(z) \le (f^t)'(z)$. Then, for any $x, m \in \mathbb{R}$ with $x \le m$, and $y, n \in \mathbb{R}$ such that $(f^{t+1})'(y) - (f^t)'(x) = (f^{t+1})'(n) - (f^t)'(m) = \xi$ for a fixed scalar $\xi$, we have $D^{t,t+1}(x, y) \le D^{t,t+1}(m, n)$, where $D^{t,t+1}(u, v) = f^t(u) - f^{t+1}(v) - (u - v) \cdot (f^{t+1})'(v)$ is the skewed Bregman divergence.*

While we state the theorem for the one-dimensional case, it trivially extends to multi-dimensional regularizers as long as they decompose over the coordinates (which is the case for all our regularizers). Take Tsallis entropy as an example: we only need to apply the theorem with $f^t(z) = -\frac{\gamma_i^t z^\beta}{1-\beta}$ for each $i$ and then sum up the conclusions. The two conditions stated in the theorem also hold for all regularizers we consider. In particular, Condition (ii) holds as long as the learning rate $\gamma_i^t$ is non-decreasing in $t$ and the regularizer itself is non-increasing (thus with nonpositive first derivative). This explains the additional linear term in our definition of Shannon entropy: this way it is strictly decreasing. Note that Condition (ii) also trivially holds if $f^t$ is independent of $t$, in which case the theorem states the monotonicity for the standard (non-skewed) Bregman divergence.

After verifying the conditions, we can now apply this theorem to show $D_U^{t,t+1}(p^t, \bar{p}^{t+1}) \le D_U^{t,t+1}(q^t, q^{t+1})$. For each $i \in U$, we take $x = p_i^t$ and $m = q_i^t$. Since by definition $q^t$ is obtained by projecting $p^t$ onto $\Omega_U$, it can be shown via KKT conditions that $p_i^t \le q_i^t$ indeed holds for all $i \in U$. Then, we define an intermediate point $z$ such that $\nabla\phi_U^{t+1}(z) - \nabla\phi_U^t(p^t) = \nabla\phi_U^{t+1}(q^{t+1}) - \nabla\phi_U^t(q^t)$ and show $D_U^{t,t+1}(p^t, \bar{p}^{t+1}) \le D_U^{t,t+1}(p^t, z)$. Finally, taking $y = z_i$ and $n = q_i^{t+1}$ and applying Theorem 4.3 finishes the proof; see Section A.3 for details.

**Residual Regret**  Finally, bounding the residual regret $D^{t+1}(\bar{p}^{t+1}, p^{t+1})$ by the self-bounding terms in Eq. (3) that are only in terms of arms in $V$ is another key challenge in our analysis, especially given the arm-dependent learning rates. We start by developing a new analysis that leads to tighter bounds compared to [Ito, 2021] on the Lagrangian multipliers associated with Eq. (7), which reveals that the key to analyze $D^{t+1}(\bar{p}^{t+1}, p^{t+1})$ is to bound the following term (or terms of a similar form)

$$\frac{\left(\sum_{i \in V} \frac{(p_i^t)^{2-\beta}}{\gamma_i^t}\right)\left(\sum_{i \in U} \frac{(p_i^t)^{3-2\beta}}{(\gamma_i^t)^2}\right)}{\left(\sum_{i \in [K]} \frac{(p_i^t)^{2-\beta}}{\gamma_i^t}\right)\left(\sum_{i \in U} \frac{(p_i^t)^{2-\beta}}{\gamma_i^t}\right)}. \tag{12}$$

Again, for the case of $1/2$-Tsallis entropy with an arm-independent learning rate $\gamma_i^t = \sqrt{t}$, removing all dependence on $i \in U$ in Eq. (12) is relatively straightforward as shown by Ito [2021]. Indeed, in this case, Eq. (12) simplifies to $\frac{1}{\sqrt{t}} \frac{\left(\sum_{i \in V} (p_i^t)^{3/2}\right)\left(\sum_{i \in U} (p_i^t)^2\right)}{\left(\sum_{i \in [K]} (p_i^t)^{3/2}\right)\left(\sum_{i \in U} (p_i^t)^{3/2}\right)}$. By splitting $\sum_{i \in [K]} (p_i^t)^{3/2}$ into two summations, one over $i \in V$ and another over $i \in U$, and further applying $x + y \geq x^{2/3} y^{1/3}$ for any $x, y > 0$, we have $\sum_{i \in [K]} (p_i^t)^{3/2} \geq \left(\sum_{i \in V} (p_i^t)^{3/2}\right)^{2/3}\left(\sum_{i \in U} (p_i^t)^{3/2}\right)^{1/3}$ and thus Eq. (12) is bounded by

$$\frac{1}{\sqrt{t}} \frac{\left(\sum_{i \in V} (p_i^t)^{3/2}\right)\left(\sum_{i \in U} (p_i^t)^2\right)}{\left(\sum_{i \in V} (p_i^t)^{3/2}\right)^{2/3}\left(\sum_{i \in U} (p_i^t)^{3/2}\right)^{4/3}} \leq \frac{1}{\sqrt{t}} \left(\sum_{i \in V} (p_i^t)^{3/2}\right)^{1/3} \leq \frac{\sum_{i \in V} \sqrt{p_i^t}}{\sqrt{t}},$$

where importantly, in the second step we use the fact $\|x\|_2 \leq \|x\|_{3/2}$ to drop all dependence on $i \in U$, eventually arriving at the self-bounding term of Eq. (3).

Unfortunately, with an arm-dependent learning rate, it is unclear to us how to analyze Eq. (12) in a similar way. Instead, we propose a different analysis with the following rough idea: we propose a condition under which Eq. (12) can be bounded by $\sum_{i \in V} (p_i^t)^{2-\beta}/\gamma_i^t$ and then further related to a self-bounding term similarly to the analysis of the regret on sub-optimal arms. If, on the other hand, the condition does not hold, then we show that the probability $p_i^t$ of selecting an optimal arm $i \in U$ must be no more than the total probability of selecting sub-optimal arms $\sum_{j \in V} p_j^t$. Using this fact again allows us to convert dependence on $i \in U$ to $i \in V$. Since we apply this technique to all $i \in U$, it leads to the extra $|U|$ factor in the second self-bounding term of Eq. (3), which eventually translates to $\frac{|U| \log T}{\Delta_{\min}}$ in our instance complexity $\mathbb{G}$. All details can be found in Section A.4.

## 5   Conclusions

In this work, we improve and generalize the analysis of [Ito, 2021], showing that many FTRL algorithms can achieve BOBW without the uniqueness assumption. Specifically, we propose a unified arm-dependent learning rate schedule and novel analytical techniques to remove the uniqueness assumption for a broad family of regularizers, including log-barrier, $\beta$-Tsallis entropy, and Shannon entropy. With these new techniques, our regret bounds improve upon prior results even when the uniqueness assumption holds. We further apply our results to the decoupled exploration and exploitation setting, showing that our techniques are broadly applicable.

There are many natural future directions, including (1) removing the $\frac{|U|}{\Delta_{\text{MIN}}}$ term in our regret bounds; (2) improving the dependence on $D$ (which as mentioned could be as large as the corruption level $C$); (3) understanding whether the extra the log-barrier regularizer is necessary or not; (4) and finally generalizing our results to other problems such as semi-bandits and Markov Decision Processes.

## Acknowledgments and Disclosure of Funding

TJ and HL are supported by NSF Award IIS-1943607.

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

# Appendix

# A Proofs for MAB Results in Table 1

In this section, we provide details of our novel analysis framework and take a closer look at our learning rate schedule. Importantly, our analysis framework is able to be applied to all configurations of Algorithm 1 for different regularizers in a unified way. As mentioned, the key term in the regret is $\sum_{t=1}^{T} D^{t,t+1}(p^t, p^{t+1}) - D_U^{t,t+1}(q^t, q^{t+1})$ with $q_t$ defined in Eq. (5). We thus define the preprocessing cost (PRECOST) as

$$\text{PRECOST} = \mathbb{E}\left[ \text{Reg}^T - \sum_{t=1}^{T} \left( D^{t,t+1}(p^t, p^{t+1}) - D_U^{t,t+1}(q^t, q^{t+1}) \right) \right], \tag{13}$$

which we later show is small. Then, we use Lemma 4.1 to decompose the key term into the regret related to the sub-optimal arms (REGSUB), the regret related to the optimal arms (REGOPT) and the residual regret (RESREG):

$$\mathbb{E}\left[ \sum_{t=1}^{T} D^{t,t+1}(p^t, p^{t+1}) - D_U^{t,t+1}(q^t, q^{t+1}) \right] = \underbrace{\mathbb{E}\left[ \sum_{t=1}^{T} D_V^{t,t+1}(p^t, \bar{p}^{t+1}) \right]}_{\text{REGSUB}}$$

$$+ \underbrace{\mathbb{E}\left[ \sum_{t=1}^{T} D_U^{t,t+1}(p^t, \bar{p}^{t+1}) - D_U^{t,t+1}(q^t, q^{t+1}) \right]}_{\text{REGOPT}} + \underbrace{\mathbb{E}\left[ \sum_{t=1}^{T} D^{t+1}(\bar{p}^{t+1}, p^{t+1}) \right]}_{\text{RESREG}}.$$

Our goal is to bound all these four terms in terms of two self-bounding quantities for some $x > 0$ (plus other minor terms):

$$\mathbb{S}_1(x) = \mathbb{E}\left[ \sqrt{x \cdot \sum_{t=1}^{T} \sum_{i \in V} p_i^t} \right], \quad \mathbb{S}_2(x) = \mathbb{E}\left[ \sum_{i \in V} \sqrt{x \cdot \sum_{t=1}^{T} p_i^t} \right]. \tag{14}$$

These two quantities enjoy a certain self-bounding property which is known to be critical to achieve the gap-dependent bound in the stochastic setting. Specifically, as we illustrate at the beginning of Section 4, under Condition (1), these self-bounding quantities can be related back to the regret itself when the set $V$ coincides with that of Condition (1).

**Lemma A.1** (Self-Bounding Quantities). *Under Condition (1), we have for any $z > 0$:*

$$\mathbb{S}_1(x) \leq z \left( \text{Reg}^T + C \right) + \frac{1}{z} \left( \frac{x}{4\Delta_{\text{MIN}}} \right),$$

$$\mathbb{S}_2(x) \leq z \left( \text{Reg}^T + C \right) + \frac{1}{z} \left( \sum_{i \in V} \frac{x}{4\Delta_i} \right).$$

*Proof.* For any $z > 0$, we have

$$\mathbb{S}_1(x) = \mathbb{E}\left[ \sqrt{\frac{x}{2z\Delta_{\text{MIN}}} \cdot \left( 2z\Delta_{\text{MIN}} \sum_{t=1}^{T} \sum_{i \in V} p_i^t \right)} \right]$$

$$\leq \mathbb{E}\left[ z\Delta_{\text{MIN}} \sum_{t=1}^{T} \sum_{i \in V} p_i^t \right] + \frac{x}{4z\Delta_{\text{MIN}}}$$

$$\leq z \left( \text{Reg}^T + C \right) + \frac{x}{4z\Delta_{\text{MIN}}},$$

where the second step follows from the AM-GM inequality: $2\sqrt{xy} \leq x + y$ for any $x, y > 0$, and the last step follows from Condition (1).

By similar arguments, we have $\mathbb{S}_2(x)$ bounded for any $z > 0$ as:

$$\mathbb{E}\left[ \sum_{i \in V} \sqrt{\frac{x}{2z\Delta_i} \cdot 2z\Delta_i \sum_{t=1}^{T} p_i^t} \right] \leq \sum_{i \in V} \frac{x}{4z\Delta_i} + z\mathbb{E}\left[ \sum_{i \in V} \sum_{t=1}^{T} p_i^t \Delta_i \right] = z \left( \text{Reg}^T + C \right) + \sum_{i \in V} \frac{x}{4z\Delta_i}.$$

Table 3: Bounds of PRECOST, REGSUB, REGOPT, and RESREG for Algorithm 1 with different regularizers. For simplicity, We omit the $\mathcal{O}(\cdot)$ notation in these bounds.

| Terms | Shannon entropy $(C_{\text{LOG}} = 162 \log K)$ | $\beta$-Tsallis entropy $\left(C_{\text{LOG}} = \frac{162\beta}{1-\beta}\right)$ | log-barrier $(C_{\text{LOG}} = 162)$ |
|---|---|---|---|
| PRECOST (Section A.1) | $K(\log T)(\log K) + D$ (Lemma A.1.4) | $\frac{K}{\sqrt{\beta(1-\beta)}} + \frac{\beta K \log T}{1-\beta} + D$ (Lemma A.1.6) | $\mathbb{S}_2(\log T) + K \log T + D$ (Lemma A.1.5) |
| REGSUB (Section A.2) | $\mathbb{S}_2\left(\log^2 T\right)$ (Lemma A.2.4) | $\mathbb{S}_2\left(\frac{\log T}{\beta(1-\beta)}\right)$ (Lemma A.2.3) | $\mathbb{S}_2(\log T)$ (Lemma A.2.2) |
| REGOPT (Section A.3) | $0$ (Lemma A.3.1) | $0$ (Lemma A.3.1) | $0$ (Lemma A.3.1) |
| RESREG (Section A.4) | $\mathbb{S}_1\left(\lvert U\rvert \log^2 T\right) + \mathbb{S}_2\left(\log^2 T\right)$ $+ K^2 \log^{3/2} T$ (Lemma A.4.5) | $\mathbb{S}_1\left(\frac{\lvert U\rvert \log T}{\beta(1-\beta)}\right) + \mathbb{S}_2\left(\frac{\log T}{\beta(1-\beta)}\right)$ $+ \frac{K^2\sqrt{\beta}}{(1-\beta)^{3/2}}$ (Lemma A.4.3) | $\mathbb{S}_1\left(\lvert U\rvert \log T\right) + \mathbb{S}_2\left(\log T\right)$ $+ \frac{K^2}{\sqrt{\log T}}$ (Lemma A.4.4) |
| $\text{Reg}^T$ | $\mathbb{S}_1\left(\lvert U\rvert \log^2 T\right) + \mathbb{S}_2\left(\log^2 T\right)$ $+ K^2 \log^{3/2} T + D$ (Lemma A.3) | $\mathbb{S}_1\left(\frac{\lvert U\rvert \log T}{\beta(1-\beta)}\right) + \mathbb{S}_2\left(\frac{\log T}{\beta(1-\beta)}\right)$ $+ \frac{K^2\sqrt{\beta}}{(1-\beta)^{3/2}} + \frac{\beta K \log T}{1-\beta} + D$ (Lemma A.2) | $\mathbb{S}_1\left(\lvert U\rvert \log T\right) + \mathbb{S}_2\left(\log T\right)$ $+ \frac{K^2}{\sqrt{\log T}} + K \log T + D$ (Lemma A.4) |

$\square$

In Table 3, we summarize the bounds of PRECOST, REGSUB, REGOPT and RESREG when using Algorithm 1 with Shannon entropy, $\beta$-Tsallis entropy, and log-barrier, respectively, together with the corresponding lemmas. Importantly, though the bounds are stated in separated lemmas for different regularizers, the proof ideas and techniques are almost the same with only slight modifications to adapt to the specific regularizer and the choice of parameters.

Using the bound on each of there four terms, we now show how to prove our main theorems, starting with the case for $\beta$-Tsallis entropy. First, we simply sum up all the bounds to arrive at the following adaptive regret bound.

**Lemma A.2** (Regret Bound for $\beta$-Tsallis Entropy). *For any $\beta \in (0,1)$, when using $\beta$-Tsallis entropy with $C_{\text{LOG}} = \frac{162\beta}{1-\beta}$, $\alpha = \beta$, and $\theta = \sqrt{\frac{1-\beta}{\beta}}$, Algorithm 1 guarantees:*

$$\text{Reg}^T = \mathcal{O}\left(D + \frac{K^2\sqrt{\beta}}{(1-\beta)^{3/2}} + \frac{\beta K \log T}{1-\beta} + \mathbb{S}_1\left(\frac{\lvert U\rvert \log T}{\beta(1-\beta)}\right) + \mathbb{S}_2\left(\frac{\log T}{\beta(1-\beta)}\right)\right),$$

*for any subset $U \subseteq [K]$, $V = [K]\backslash U$, and $D = \mathbb{E}\left[\sum_{t=1}^{T} \max_{i \in U} \mathbb{E}^t\left[\ell_i^t - \ell_{i^\star}^t\right]\right]$.*

We emphasize that this bound holds for *any* subset $U \subseteq [K]$, not just the $U$ defined with respect to Condition (1) (so a slight abuse of notations here). This is important for proving the regret bound in the adversarial case, as shown in the following proof for Theorem 3.3.

*Proof of Theorem 3.3.* First, for the adversarial setting, we set $U = \{i^\star\}$ so that $D = 0$ by its definition. Therefore, we have the self-bounding quantity $\mathbb{S}_1\left(\frac{\lvert U\rvert \log T}{\beta(1-\beta)}\right)$ bounded as:

$$\mathbb{S}_1\left(\frac{\lvert U\rvert \log T}{\beta(1-\beta)}\right) = \mathbb{E}\left[\sqrt{\frac{\lvert U\rvert \log T}{\beta(1-\beta)} \cdot \sum_{t=1}^{T}\sum_{i \in V} p_i^t}\right] = \mathcal{O}\left(\sqrt{\frac{T \log T}{\beta(1-\beta)}}\right),$$

where the second step follows from the fact $|U| = 1$ and $\sum_{t=1}^{T} \sum_{i \in [K]} p_i^t = T$. Similarly, we bound the other self-bounding quantity $\mathbb{S}_2\left(\frac{\log T}{\beta(1-\beta)}\right)$ as:

$$\mathbb{S}_2\left(\frac{\log T}{\beta(1-\beta)}\right) = \mathbb{E}\left[\sum_{i \in V}\sqrt{\frac{\log T}{\beta(1-\beta)}\sum_{t=1}^{T} p_i^t}\right] \leq \mathbb{E}\left[\sqrt{\frac{|V|\log T}{\beta(1-\beta)}\sum_{i \in V}\sum_{t=1}^{T} p_i^t}\right] = \mathcal{O}\left(\sqrt{\frac{KT\log T}{\beta(1-\beta)}}\right),$$

where the second step uses the Cauchy-Schwarz inequality. Combining the bounds together concludes that $\mathrm{Reg}^T = \mathcal{O}\left(\sqrt{\frac{KT\log T}{\beta(1-\beta)}}\right)$ in the adversarial setting.

Next, suppose that Condition (1) holds. We now set $U = \{i \in [K] : \Delta_i = 0\}$ as in Section 2, which makes $D$ the same as that defined in Section 2 as well. We further denote $W = D + \frac{K^2\sqrt{\beta}}{(1-\beta)^{3/2}} + \frac{\beta K \log T}{1-\beta}$ to simplify notations. Then, we have $\mathrm{Reg}^T$ bounded for any $z > 0$ by

$$\mathrm{Reg}^T \leq \kappa \cdot W + \kappa \cdot \mathbb{S}_1\left(\frac{|U|\log T}{\beta(1-\beta)}\right) + \kappa \cdot \mathbb{S}_2\left(\frac{\log T}{\beta(1-\beta)}\right)$$

$$\leq z\left(\mathrm{Reg}^T + C\right) + \frac{\kappa^2}{z} \cdot \frac{\mathbb{G}\log T}{\beta(1-\beta)} + \kappa \cdot W,$$

where $\kappa > 0$ is an absolute constant (based on Lemma A.2), and the second step applies Lemma A.1 (with the arbirary $z$ there set to $\frac{z}{2\kappa}$) together with our complexity measure $\mathbb{G} = \frac{|U|}{\Delta_{\mathrm{MIN}}} + \sum_{i \in V}\frac{1}{\Delta_i}$.

For any $z \in (0, 1)$, we can further rearrange and arrive at

$$\mathrm{Reg}^T \leq \frac{zC}{1-z} + \frac{1}{z(1-z)} \cdot \frac{\kappa^2\mathbb{G}\log T}{\beta(1-\beta)} + \frac{1}{1-z} \cdot \kappa W$$

$$= \frac{C}{x} + \frac{(x+1)^2}{x} \cdot \frac{\kappa^2\mathbb{G}\log T}{\beta(1-\beta)} + \frac{x+1}{x} \cdot \kappa W$$

$$= \frac{1}{x} \cdot \left(C + \frac{\kappa^2\mathbb{G}\log T}{\beta(1-\beta)} + \kappa W\right) + x \cdot \frac{\kappa^2\mathbb{G}\log T}{\beta(1-\beta)} + \left(\frac{2\kappa^2\mathbb{G}\log T}{\beta(1-\beta)} + \kappa W\right),$$

where we define $x = \frac{1-z}{z} > 0$ in the second step. Picking up the optimal $x$ to minimize the right hand side gives

$$\mathrm{Reg}^T \leq 2\sqrt{\left(C + \frac{\kappa^2\mathbb{G}\log T}{\beta(1-\beta)} + \kappa W\right) \cdot \frac{\kappa^2\mathbb{G}\log T}{\beta(1-\beta)}} + \frac{2\kappa^2\mathbb{G}\log T}{\beta(1-\beta)} + \kappa W$$

$$\leq 6\sqrt{\kappa^2 \cdot \frac{C\mathbb{G}\log T}{\beta(1-\beta)}} + \frac{8\kappa^2\mathbb{G}\log T}{\beta(1-\beta)} + \kappa W + 6\sqrt{\kappa W \cdot \frac{\kappa^2\mathbb{G}\log T}{\beta(1-\beta)}}$$

$$\leq 6\sqrt{\kappa^2 \cdot \frac{C\mathbb{G}\log T}{\beta(1-\beta)}} + \frac{11\kappa^2\mathbb{G}\log T}{\beta(1-\beta)} + 4\kappa W$$

$$= \mathcal{O}\left(\sqrt{\frac{C\mathbb{G}\log T}{\beta(1-\beta)}} + \frac{\mathbb{G}\log T}{\beta(1-\beta)} + W\right),$$

where the second step follows from the fact that $\sqrt{x+y+z} \leq 3\left(\sqrt{x} + \sqrt{y} + \sqrt{z}\right)$ for any $x, y, z \geq 0$, and the third step uses the AM-GM inequality $2\sqrt{xy} \leq x + y$ for any $x, y \geq 0$. $\square$

For the Shannon entropy regularizer and the log-barrier regularizer, we again summarize the adaptive regret bound in the following two lemmas. Theorem 3.2 and Theorem 3.1 then follow immediately. Since the arguements are exactly the same as above, we omit them for simplicity.

**Lemma A.3** (Regret Bound for Shannon Entropy). *When using the Shannon entropy regularizer with* $C_{\mathrm{LOG}} = 162\log K$, $\alpha = 1$, *and* $\theta = \sqrt{1/\log T}$, *Algorithm 1 ensures*

$$\mathrm{Reg}^T = \mathcal{O}\left(D + K^2\log^{3/2} T + \mathbb{S}_1\left(|U|\log^2 T\right) + \mathbb{S}_2\left(\log^2 T\right)\right),$$

*for any subset* $U \subseteq [K]$, $V = [K]\backslash U$, *and* $D = \mathbb{E}\left[\sum_{t=1}^{T}\max_{i \in U}\mathbb{E}^t\left[\ell_i^t - \ell_{i^\star}^t\right]\right]$.

**Lemma A.4** (Regret Bound for Log-barrier). *When using the log-barrier regularizer with $C_{\text{LOG}} = 162$, $\alpha = 0$, and $\theta = \sqrt{1/\log T}$, Algorithm 1 ensures*

$$\text{Reg}^T = \mathcal{O}\left( D + \frac{K^2}{\sqrt{\log T}} + K \log T + \frac{\beta K \log T}{1 - \beta} + \mathbb{S}_1 \left( |U| \log T \right) + \mathbb{S}_2 \left( \log T \right) \right),$$

*for any subset $U \subseteq [K]$, $V = [K]\backslash U$, and $D = \mathbb{E}\left[ \sum_{t=1}^{T} \max_{i \in U} \mathbb{E}^t \left[ \ell_i^t - \ell_{i^\star}^t \right] \right]$.*

In the following four subsections, we discuss how to bound PRECOST, REGSUB, REGOPT, and RESREG respectively. We recall the following definitions discussed in Section 4: the skewed Bregman divergence for two time steps $s, t \in \mathbb{N}$ is

$$D^{s,t}(x, y) = \phi^s(x) - \phi^t(y) - \left\langle \nabla \phi^t(y), x - y \right\rangle,$$

and its variant restricted to any subset $\mathcal{I} \subseteq [K]$ is

$$D_{\mathcal{I}}^{s,t}(x, y) = \phi_{\mathcal{I}}^s(x) - \phi_{\mathcal{I}}^t(y) - \left\langle \nabla \phi_{\mathcal{I}}^t(y), x - y \right\rangle,$$

where $\phi_{\mathcal{I}}^t(x) = -C_{\text{LOG}} \sum_{i \in \mathcal{I}} \log x_i - \frac{1}{1-\beta} \sum_{i \in \mathcal{I}} \gamma_i^t x_i^\beta$ (that is, $\phi^t$ restricted to $\mathcal{I}$). The standard Bregman divergence associated with $\phi^t$ is denoted by $D^t(x, y)$, a shorthand for $D^{t,t}(x, y)$. For any subset $\mathcal{I} \subseteq [K]$ and any vector $x \in \mathbb{R}^K$, we define $x_{\mathcal{I}} \in \mathbb{R}^K$ such that its entries in $\mathcal{I}$ are the same as $x$ while those in $[K]\backslash \mathcal{I}$ are 0. Finally, we use $\mathbf{1} \in \mathbb{R}^K$ to denote the all-one vector.

### A.1 Preprocessing Cost

Due to the extra log-barrier added to stabilize the algorithm, we consider distribution $\widetilde{q}^t$ defined as

$$\widetilde{q}^t = (1 - \epsilon) \cdot q^t + \frac{\epsilon}{|V|} \cdot \mathbf{1}_V, \tag{15}$$

where $\mathbf{1}_V \in \mathbb{R}^K$, according to our earlier notations, is the vector with all entries in $V$ being 1 and the remaining being 0. This specific distribution moves a small amount of weights of $q^t$ from $U$ to $V$, which guarantees that our framework can also work for the log-barrier and $\log(1/\widetilde{q}_i^t)$ is bounded for $\forall i \in V$ with a proper $\epsilon$.

**Lemma A.1.1.** *For any $\epsilon \in (0, 1)$, any subset $U \subseteq [K]$ and $V = [K]\backslash U$, $q^t$ defined in Eq. (5), and $\widetilde{q}^t$ defined in Eq. (15), Algorithm 1 ensures*

$$\text{Reg}^T \leq \mathbb{E}\left[ \sum_{t=1}^{T} \left\langle \widehat{\ell}^t, p^t - \widetilde{q}^t \right\rangle \right] + D + \epsilon T,$$

*where $D = \mathbb{E}\left[ \sum_{t=1}^{T} \max_{i \in U} \mathbb{E}^t \left[ \ell_i^t - \ell_{i^\star}^t \right] \right]$.*

*Proof.* By the definition of $\widetilde{q}^t$ in Eq. (15), one can show

$$\sum_{t=1}^{T} \mathbb{E}\left[ \langle \ell^t, \widetilde{q}^t \rangle - \ell_{i^\star}^t \right] = \sum_{t=1}^{T} \mathbb{E}\left[ \langle \ell^t, \widetilde{q}_U^t \rangle - \ell_{i^\star}^t \right] + \sum_{t=1}^{T} \mathbb{E}\left[ \langle \ell^t, \widetilde{q}_V^t \rangle \right]$$

$$= \sum_{t=1}^{T} \mathbb{E}\left[ \mathbb{E}^t \left[ \langle \ell^t, \widetilde{q}_U^t \rangle - \ell_{i^\star}^t \right] \right] + \sum_{t=1}^{T} \mathbb{E}\left[ \frac{\epsilon \sum_{i \in V} \ell_i^t}{|V|} \right]$$

$$\leq \sum_{t=1}^{T} \mathbb{E}\left[ (1 - \epsilon) \max_{i \in U} \mathbb{E}^t \left[ \ell_i^t \right] - \mathbb{E}^t \left[ \ell_{i^\star}^t \right] \right] + \epsilon T \leq D + \epsilon T. \tag{16}$$

Therefore, we have

$$\text{Reg}^T = \sum_{t=1}^{T} \mathbb{E}\left[ \ell_{i^t}^t - \langle \ell^t, \widetilde{q}^t \rangle + \langle \ell^t, \widetilde{q}^t \rangle - \ell_{i^\star}^t \right]$$

$$\leq \sum_{t=1}^{T} \mathbb{E}\left[ \ell_{i^t}^t - \langle \ell^t, \widetilde{q}^t \rangle \right] + D + \epsilon T = \mathbb{E}\left[ \sum_{t=1}^{T} \left\langle \widehat{\ell}^t, p^t - \widetilde{q}^t \right\rangle \right] + D + \epsilon T,$$

where the second step uses Eq. (16), and the last step follows from the law of total expectation. $\square$

According to Lemma A.1.1, our main goal here is to bound $\left\langle \widehat{\ell}^t, p^t - \widetilde{q}^t \right\rangle$. To this end, we present the following decomposition lemma which is helpful for our further analysis.

**Lemma A.1.2** (Lemma 16, Ito [2021]). *For $p^t$ in Algorithm 1 and $\widetilde{q}^t$ in Eq.* (15), *we have*

$$\left\langle \widehat{\ell}^t, p^t - \widetilde{q}^t \right\rangle = D^{t,t+1}(p^t, p^{t+1}) + D^t(\widetilde{q}^t, p^t) - D^{t,t+1}(\widetilde{q}^t, p^{t+1}).$$

Armed with Lemma A.1.2, we are now ready to introduce our main result of the regret preprocessing. It is worth noting that the following lemma does not rely on the specific form of regularizers, and thus, we can use it for all the regularizers we consider.

**Lemma A.1.3.** *Suppose that there exist $B_1, B_2 \in \mathbb{R}$ such that*

$$\mathbb{E}\left[\sum_{t=1}^{T} \phi_U^{t+1}(\widetilde{q}^{t+1}) - \phi_U^t(\widetilde{q}^t)\right] \leq B_1 + \mathbb{E}\left[\sum_{t=1}^{T} \phi_U^{t+1}(q^{t+1}) - \phi_U^t(q^t)\right], \tag{17}$$

$$\mathbb{E}\left[\sum_{t=1}^{T} \phi_V^{t+1}(\widetilde{q}^{t+1}) - \phi_V^t(\widetilde{q}^t)\right] \leq B_2. \tag{18}$$

*Let $B = B_1 + B_2$, and we have*

$$\mathbb{E}\left[\sum_{t=1}^{T} \left\langle \widehat{\ell}^t, p^t - \widetilde{q}^t \right\rangle\right] \leq \mathbb{E}\left[D^1(\widetilde{q}^1, p^1) + \sum_{t=1}^{T} D^{t,t+1}(p^t, p^{t+1}) - D_U^{t,t+1}(q^t, q^{t+1})\right] + \mathcal{O}\left(\epsilon KT\right) + B.$$

*In other words, we have* PRECOST *bounded by* $\mathbb{E}\left[D^1(\widetilde{q}^1, p^1) + \mathcal{O}\left(\epsilon KT\right) + B + D\right].$

*Proof.* Adding and subtracting $D^{t+1}(\widetilde{q}^{t+1}, p^{t+1})$ from the bound in Lemma A.1.2 , we have

$$\left\langle \widehat{\ell}^t, p^t - \widetilde{q}^t \right\rangle = D^{t,t+1}(p^t, p^{t+1}) + D^{t+1}(\widetilde{q}^{t+1}, p^{t+1}) - D^{t,t+1}(\widetilde{q}^t, p^{t+1})$$
$$+ D^t(\widetilde{q}^t, p^t) - D^{t+1}(\widetilde{q}^{t+1}, p^{t+1}).$$

First, we consider the term $D^{t+1}(\widetilde{q}^{t+1}, p^{t+1}) - D^{t,t+1}(\widetilde{q}^t, p^{t+1})$ as follows:

$$D^{t+1}(\widetilde{q}^{t+1}, p^{t+1}) - D^{t,t+1}(\widetilde{q}^t, p^{t+1})$$
$$= \phi^{t+1}(\widetilde{q}^{t+1}) - \phi^t(\widetilde{q}^t) - \left\langle \nabla\phi^{t+1}(p^{t+1}), \widetilde{q}^{t+1} - \widetilde{q}^t \right\rangle$$
$$= \phi^{t+1}(\widetilde{q}^{t+1}) - \phi^t(\widetilde{q}^t) - (1-\epsilon) \cdot \left\langle \nabla\phi^{t+1}(p^{t+1}), q^{t+1} - q^t \right\rangle$$
$$= \phi^{t+1}(\widetilde{q}^{t+1}) - \phi^t(\widetilde{q}^t) - (1-\epsilon) \cdot \left\langle \nabla\phi_U^{t+1}(p^{t+1}), q^{t+1} - q^t \right\rangle$$
$$= \phi^{t+1}(\widetilde{q}^{t+1}) - \phi^t(\widetilde{q}^t) - \left\langle \nabla\phi_U^{t+1}(p^{t+1}), q^{t+1} - q^t \right\rangle + \epsilon \left\langle \nabla\phi_U^{t+1}(p^{t+1}), q^{t+1} - q^t \right\rangle$$
$$= \phi^{t+1}(\widetilde{q}^{t+1}) - \phi^t(\widetilde{q}^t) - \left\langle \nabla\phi_U^{t+1}(q^{t+1}), q^{t+1} - q^t \right\rangle + \epsilon \left\langle \nabla\phi_U^{t+1}(p^{t+1}), q^{t+1} - q^t \right\rangle$$
$$= \phi^{t+1}(\widetilde{q}^{t+1}) - \phi^t(\widetilde{q}^t) - \left\langle \nabla\phi_U^{t+1}(q^{t+1}), q^{t+1} - q^t \right\rangle + \epsilon \left\langle \widehat{L}_U^{t+1}, q^{t+1} - q^t \right\rangle,$$

where the second step follows from the fact that $\widetilde{q}^{t+1} - \widetilde{q}^t = (1-\epsilon)\left(q^{t+1} - q^t\right)$ according to Eq. (15); the fifth step follows from facts that $\nabla\phi_U^{t+1}(q^{t+1}) = \nabla\phi_U^{t+1}(p^{t+1}) + c \cdot \mathbf{1}_U$ for a Lagrange multiplier $c \in \mathbb{R}$ and $\left\langle c \cdot \mathbf{1}_K, q^{t+1} - q^t \right\rangle = 0$; the last step uses the fact that $\nabla\phi^{t+1}(p^{t+1}) = \widehat{L}^{t+1} + c' \cdot \mathbf{1}_K$ where $\widehat{L}^t \triangleq \sum_{\tau < t} \widehat{\ell}^\tau$ is the cumulative loss vector prior to round $t$ and $c' \in \mathbb{R}$ is another Lagrange multiplier.

By Eq. (17), Eq. (18), and the definition $B = B_1 + B_2$, we can further show

$$\mathbb{E}\left[\sum_{t=1}^{T} D^{t+1}(\widetilde{q}^{t+1}, p^{t+1}) - D^{t,t+1}(\widetilde{q}^t, p^{t+1})\right]$$
$$\leq B + \mathbb{E}\left[\sum_{t=1}^{T} \phi_U^{t+1}(q^{t+1}) - \phi_U^t(q^t) - \left\langle \nabla\phi_U^{t+1}(q^{t+1}), q^{t+1} - q^t \right\rangle + \epsilon \left\langle \widehat{L}_U^{t+1}, q^{t+1} - q^t \right\rangle\right]$$

$$= B - \mathbb{E}\left[\sum_{t=1}^{T} D_U^{t,t+1}(q^t, q^{t+1})\right] + \mathbb{E}\left[\sum_{t=1}^{T} \epsilon\left\langle \widehat{L}_U^{t+1}, q^{t+1} - q^t \right\rangle\right].$$

Then, we can show the following by reorganizing the summation:

$$\mathbb{E}\left[\epsilon\sum_{t=1}^{T}\left\langle \widehat{L}_U^{t+1}, q^{t+1} - q^t \right\rangle\right] = \epsilon \cdot \mathbb{E}\left[\left\langle \widehat{L}_U^{T+1}, q^{T+1} \right\rangle - \left\langle \widehat{L}_U^2, q^1 \right\rangle + \sum_{t=2}^{T}\left\langle q^t, \widehat{L}_U^t - \widehat{L}_U^{t+1} \right\rangle\right]$$

$$\leq \epsilon \cdot \mathbb{E}\left[\left\langle \widehat{L}_U^{T+1}, \mathbf{1}_U \right\rangle\right] \leq \mathcal{O}\left(\epsilon K T\right),$$

where the second step follows $\widehat{L}_U^t - \widehat{L}_U^{t+1} = -\widehat{\ell}_U^t$ and the last step uses $q_i^t \leq 1$ for $\forall t, i$.

Finally, by telescoping, we have

$$\mathbb{E}\left[\sum_{t=1}^{T} D^t(\widetilde{q}^t, p^t) - D^{t+1}(\widetilde{q}^{t+1}, p^{t+1})\right] = \mathbb{E}\left[D^1(\widetilde{q}^1, p^1) - D^{t+1}(\widetilde{q}^{T+1}, p^{T+1})\right] \leq \mathbb{E}\left[D^1(\widetilde{q}^1, p^1)\right],$$

where the second step holds since the Bregman divergence is non-negative. Combining all the inequalities above concludes the proof. □

**Lemma A.1.4** (PRECOST for Shannon Entropy). *When using the Shannon entropy regularizer with $\alpha = 0$ and $\theta = \sqrt{1/\log T}$, Algorithm 1 ensures*

- *Eq. (17) with $B_1 = 4\epsilon T \log K + 1$;*

- *Eq. (18) with $B_2 = 0$;*

- $D^1(\widetilde{q}^1, p^1) \leq \frac{\log K}{\sqrt{\log T}} + C_{\text{LOG}}|V|\log\left(\frac{|V|}{K\epsilon}\right) + \frac{\epsilon C_{\text{LOG}}|U|}{1-\epsilon}.$

*Therefore, according to Lemma A.1.3, by picking $\epsilon = \frac{1}{T}$, our algorithm ensures*

$$\text{PRECOST} = \mathcal{O}\left(C_{\text{LOG}} K \log T + D\right).$$

*Proof.* We first show Eq. (18). For the entries in $V$, we have

$$\phi_V^{t+1}(\widetilde{q}^{t+1}) - \phi_V^t(\widetilde{q}^t) = \sum_{i \in V}\left(\gamma_i^{t+1}\widetilde{q}_i^{t+1}\log\left(\frac{\widetilde{q}_i^{t+1}}{e}\right) - \gamma_i^t\widetilde{q}_i^t\log\left(\frac{\widetilde{q}_i^t}{e}\right)\right)$$

$$- C_{\text{LOG}} \cdot \sum_{i \in V}\left(\log\widetilde{q}_i^{t+1} - \log\widetilde{q}_i^t\right)$$

$$= \sum_{i \in V}\left(\gamma_i^{t+1} - \gamma_i^t\right) \cdot \frac{\epsilon}{|V|}\log\left(\frac{\epsilon}{e|V|}\right) \leq 0,$$

where the second step follows from the definition of $\widetilde{q}$ in Eq. (15) which states $\widetilde{q}_i^t = \frac{\epsilon}{|V|}$ for any $i \in V$.

For the entries in $U$, by direct calculation, we have

$$\phi_U^{t+1}(\widetilde{q}^{t+1}) - \phi_U^t(\widetilde{q}^t)$$

$$= \sum_{i \in U}\left(\gamma_i^{t+1}\widetilde{q}_i^{t+1}\log\left(\frac{\widetilde{q}_i^{t+1}}{e}\right) - \gamma_i^t\widetilde{q}_i^t\log\left(\frac{\widetilde{q}_i^t}{e}\right)\right) - C_{\text{LOG}}\sum_{i \in U}\left(\log\widetilde{q}_i^{t+1} - \log\widetilde{q}_i^t\right)$$

$$= (1-\epsilon)\sum_{i \in U}\left(\gamma_i^{t+1}q_i^{t+1}\log\left(\frac{\widetilde{q}_i^{t+1}}{e}\right) - \gamma_i^t q_i^t\log\left(\frac{\widetilde{q}_i^t}{e}\right)\right) - C_{\text{LOG}}\sum_{i \in U}\left(\log q_i^{t+1} - \log q_i^t\right)$$

$$= (1-\epsilon)\sum_{i \in U}\left(\gamma_i^{t+1}q_i^{t+1}\log\left(\frac{q_i^{t+1}}{e}\right) - \gamma_i^t q_i^t\log\left(\frac{q_i^t}{e}\right) + \log(1-\epsilon)\left(\gamma_i^{t+1}q_i^{t+1} - \gamma_i^t q_i^t\right)\right)$$

$$- C_{\text{LOG}}\sum_{i \in U}\left(\log q_i^{t+1} - \log q_i^t\right)$$

$$= \phi_U^{t+1}(q^{t+1}) - \phi_U^t(q^t)$$

$$+ (1-\epsilon)\log(1-\epsilon)\sum_{i \in U}\left(\gamma_i^{t+1}q_i^{t+1} - \gamma_i^t q_i^t\right) \tag{19}$$

$$- \epsilon\sum_{i \in U}\left(\gamma_i^{t+1}q_i^{t+1}\log\left(\frac{q_i^{t+1}}{e}\right) - \gamma_i^t q_i^t\log\left(\frac{q_i^t}{e}\right)\right). \tag{20}$$

Note that, taking the summation of Eq. (19) over all rounds yields

$$\sum_{t=1}^T (1-\epsilon)\log(1-\epsilon)\sum_{i \in U}\left(\gamma_i^{t+1}q_i^{t+1} - \gamma_i^t q_i^t\right)$$

$$= (1-\epsilon)\log(1-\epsilon)\sum_{i \in U}\left(\gamma_i^{T+1}q_i^{T+1} - \gamma_i^1 q_i^1\right)$$

$$\leq -(1-\epsilon)\log(1-\epsilon)\sum_{i \in U}\gamma_i^1 q_i^1 \leq \sum_{i \in U}\gamma_i^1 q_i^1 \leq 1,$$

where the second and third step follow from the fact $-1 \leq (1-\epsilon)\log(1-\epsilon) \leq 0$, and the last step uses the definition of the learning rates.

On the other hand, by summing Eq. (20) over all rounds, we have

$$-\epsilon\sum_{t=1}^T\sum_{i \in U}\left(\gamma_i^{t+1}q_i^{t+1}\left(\log\left(q_i^{t+1}\right) - 1\right) - \gamma_i^t q_i^t\left(\log\left(q_i^t\right) - 1\right)\right)$$

$$= -\epsilon\sum_{i \in U}\left(\gamma_i^{T+1}q_i^{T+1}\left(\log\left(q_i^{T+1}\right) - 1\right) - \gamma_i^1 q_i^1\left(\log\left(q_i^1\right) - 1\right)\right)$$

$$\leq \epsilon\sum_{i \in U}\gamma_i^{T+1}q_i^{T+1}\left(\log\left(\frac{1}{q_i^{T+1}}\right) + 1\right)$$

$$\leq 2\epsilon T\left(1 + \sum_{i \in U}q_i^{T+1}\log\left(\frac{1}{q_i^{T+1}}\right)\right)$$

$$\leq 2\epsilon T\left(1 + \log|U|\right),$$

where the second step uses the fact that $x\log x - x \leq 0$ for any $x \in [0,1]$; the third step follows from the definition of $\gamma_i^t$ which ensures $\gamma_i^{T+1} \leq \sqrt{1 + T \cdot T} \leq 2T$; the last step follows from Jensen's inequality.

Finally, we bound $D^1(\widetilde{q}^1, p^1)$ as:

$$D^1(\widetilde{q}^1, p^1) = \sum_{i \in [K]}\gamma_i^1\left(\widetilde{q}_i^1\log\widetilde{q}_i^1 - p_i^1\log p_i^1\right) + \sum_{i \in [K]}\gamma_i^1\left(p_i^1 - \widetilde{q}_i^1\right) - C_{\text{LOG}}\sum_{i \in [K]}\left(\log\left(\widetilde{q}_i^1\right) - \log\left(p_i^1\right)\right)$$

$$\leq \sum_{i \in [K]}\gamma_i^1\left(-p_i^1\log p_i^1\right) + C_{\text{LOG}}|V|\log\left(\frac{|V|}{K\epsilon}\right) + C_{\text{LOG}}|U|\log\left(\frac{1}{1-\epsilon}\right)$$

$$\leq \frac{\log K}{\sqrt{\log T}} + C_{\text{LOG}}|V|\log\left(\frac{|V|}{K\epsilon}\right) + C_{\text{LOG}}|U|\log\left(\frac{1}{1-\epsilon}\right)$$

$$\leq \frac{\log K}{\sqrt{\log T}} + C_{\text{LOG}}|V|\log\left(\frac{|V|}{K\epsilon}\right) + \frac{\epsilon C_{\text{LOG}}|U|}{1-\epsilon},$$

where the second step follows from the fact that $x\log x \leq 0$ for $x \in [0,1]$; the third step follows from the facts that $p_i^1 = \frac{1}{K}$ and $\gamma_i^1 = \sqrt{1/\log T}$ for any $i \in [K]$; the forth step uses the fact that $\log\left(\frac{1}{1-\epsilon}\right) = \log\left(1 + \frac{\epsilon}{1-\epsilon}\right) \leq \frac{\epsilon}{1-\epsilon}$ for any $\epsilon \in (0,1)$. $\qquad\square$

**Lemma A.1.5** (PRECOST for Log-barrier). *When using the log-barrier regularizer with $\alpha = 1$ and $\theta = \sqrt{1/\log T}$, Algorithm 1 ensures*

- *Eq. (17) with $B_1 = \frac{\epsilon|U|T}{1-\epsilon}$;*

- *Eq. (18) with* $B_2 = \mathbb{S}_2\left(\frac{\log^2\left(\frac{|V|}{\epsilon}\right)}{\log T}\right)$;

- $D^1(\widetilde{q}^1, p^1) \leq \frac{|V|}{\sqrt{\log T}} \log\left(\frac{|V|}{K\epsilon}\right) + \frac{|U|}{\sqrt{\log T}} \log\left(\frac{|U|}{(1-\epsilon)K}\right) + C_{\text{LOG}}|V| \log\left(\frac{|V|}{K\epsilon}\right) + \frac{\epsilon C_{\text{LOG}}|U|}{1-\epsilon}$.

*Therefore, according to Lemma A.1.3, by picking $\epsilon = \frac{1}{T}$, our algorithm ensures*

$$\text{PRECOST} = \mathcal{O}\left(\mathbb{S}_2\left(\log T\right) + C_{\text{LOG}}K\log T + D\right).$$

*Proof.* We first show that Eq. (18) holds. For the entries in $V$, we have

$$\phi_V^{t+1}(\widetilde{q}^{t+1}) - \phi_V^t(\widetilde{q}^t) = -\sum_{i\in V}\left(\gamma_i^{t+1}\log\left(\widetilde{q}_i^{t+1}\right) - \gamma_i^t\log\left(\widetilde{q}_i^t\right)\right) - C_{\text{LOG}}\cdot\sum_{i\in V}\left(\log\widetilde{q}_i^{t+1} - \log\widetilde{q}_i^t\right)$$

$$= \sum_{i\in V}\left(\gamma_i^{t+1} - \gamma_i^t\right)\log\left(\frac{|V|}{\epsilon}\right),$$

where the second step follows from the definition of $\widetilde{q}^t$ in Eq. (15) that $\widetilde{q}_i^t = \frac{\epsilon}{|V|}$ for all $i \in V$. Taking the summation over all rounds, we have

$$\sum_{t=1}^{T}\sum_{i\in V}\left(\gamma_i^{t+1} - \gamma_i^t\right)\log\left(\frac{|V|}{\epsilon}\right) \leq \sum_{i\in V}\gamma_i^{T+1}\log\left(\frac{|V|}{\epsilon}\right),$$

which could be bounded with $\mathbb{S}_2(\cdot)$ with our specific learning rate schedule as:

$$\mathbb{E}\left[\sum_{i\in V}\gamma_i^{T+1}\log\left(\frac{|V|}{\epsilon}\right)\right] = \mathcal{O}\left(\sum_{i\in V}\sqrt{\frac{\log^2\left(\frac{|V|}{\epsilon}\right)}{\log T}\cdot\left(\sum_{t=1}^{T}p_i^t\right)}\right) = \mathcal{O}\left(\mathbb{S}_2\left(\frac{\log^2\left(\frac{|V|}{\epsilon}\right)}{\log T}\right)\right).$$

For the entries in $U$, we have

$$\phi_U^{t+1}(\widetilde{q}^{t+1}) - \phi_U^t(\widetilde{q}^t)$$
$$= -\sum_{i\in U}\left(\gamma_i^{t+1}\log\left(\widetilde{q}_i^{t+1}\right) - \gamma_i^t\log\left(\widetilde{q}_i^t\right)\right) - C_{\text{LOG}}\cdot\sum_{i\in U}\left(\log\widetilde{q}_i^{t+1} - \log\widetilde{q}_i^t\right)$$
$$= -\sum_{i\in U}\left(\gamma_i^{t+1}\log\left(q_i^{t+1}\right) - \gamma_i^t\log\left(q_i^t\right)\right) - C_{\text{LOG}}\cdot\sum_{i\in U}\left(\log q_i^{t+1} - \log q_i^t\right)$$
$$\quad + \log\left(\frac{1}{1-\epsilon}\right)\sum_{i\in U}\left(\gamma_i^{t+1} - \gamma_i^t\right)$$
$$\leq \phi_U^{t+1}(q^{t+1}) - \phi_U^t(q^t) + \frac{\epsilon}{1-\epsilon}\sum_{i\in U}\left(\gamma_i^{t+1} - \gamma_i^t\right).$$

By summing over all rounds and direct calculation, we have

$$\frac{\epsilon}{1-\epsilon}\sum_{t=1}^{T}\sum_{i\in U}\left(\gamma_i^{t+1} - \gamma_i^t\right) \leq \frac{\epsilon}{1-\epsilon}\sum_{i\in U}\gamma_i^{T+1} \leq \frac{\epsilon}{1-\epsilon}|U|T.$$

Finally, we bound $D^1(\widetilde{q}^1, p^1) = \phi^1(\widetilde{q}^1) - \phi^1(p^1) - \langle\nabla\phi^1(p^1), \widetilde{q}^1 - p^1\rangle = \phi^1(\widetilde{q}^1) - \phi^1(p^1)$, because $\sum_{i\in[K]}\widetilde{q}_i^1 = \sum_{i\in[K]}p_i^1$ and $p^1 = \frac{1}{K}\cdot\mathbf{1}_K$ which implies that all entries of $\nabla\phi^1(p^1)$ are equal. Note that $p^1 = \frac{1}{K}\cdot\mathbf{1}_K$, $\widetilde{q}_U^1 = \frac{1-\epsilon}{|U|}\cdot\mathbf{1}_U$, and $\widetilde{q}_V^1 = \frac{\epsilon}{|V|}\cdot\mathbf{1}_V$. For $\phi_V^1(\widetilde{q}^1) - \phi_V^1(p^1)$, we have

$$\phi_V^1(\widetilde{q}^1) - \phi_V^1(p^1) = \sum_{i\in V}\gamma_i^1\log\left(\frac{p_i^1}{\widetilde{q}_i^1}\right) - C_{\text{LOG}}\sum_{i\in V}\left(\log\left(\widetilde{q}_i^1\right) - \log\left(p_i^1\right)\right)$$

$$= \sum_{i\in V}\gamma_i^1\log\left(\frac{1/K}{\epsilon/|V|}\right) + C_{\text{LOG}}|V|\log\left(\frac{|V|}{K\epsilon}\right)$$

$$\leq \frac{|V|}{\sqrt{\log T}} \log\left(\frac{|V|}{K\epsilon}\right) + C_{\text{LOG}} |V| \log\left(\frac{|V|}{K\epsilon}\right),$$

where $C_{\text{LOG}} \sum_{i \in V} \left(\log\left(\widetilde{q}_i^1\right) - \log\left(p_i^1\right)\right)$ is bounded via a similar way used to bound the Shannon entropy. Similarly, for $\phi_U^1(\widetilde{q}^1) - \phi_U^1(p^1)$, we have

$$\phi_U^1(\widetilde{q}^1) - \phi_U^1(p^1) = \sum_{i \in U} \gamma_i^1 \log\left(\frac{p_i^1}{\widetilde{q}_i^1}\right) - C_{\text{LOG}} \sum_{i \in U} \left(\log\left(\widetilde{q}_i^1\right) - \log\left(p_i^1\right)\right)$$

$$\leq \sum_{i \in U} \gamma_i^1 \log\left(\frac{1/K}{(1-\epsilon)/|U|}\right) + \frac{\epsilon C_{\text{LOG}} |U|}{1 - \epsilon}$$

$$\leq \frac{|U|}{\sqrt{\log T}} \log\left(\frac{|U|}{(1-\epsilon)K}\right) + \frac{\epsilon C_{\text{LOG}} |U|}{1 - \epsilon}.$$

Combining bounds for $\phi_V^1(\widetilde{q}^1) - \phi_V^1(p^1)$ and $\phi_U^1(\widetilde{q}^1) - \phi_U^1(p^1)$, we have $D^1(\widetilde{q}^1, p^1)$ bounded by

$$\frac{|V|}{\sqrt{\log T}} \log\left(\frac{|V|}{K\epsilon}\right) + \frac{|U|}{\sqrt{\log T}} \log\left(\frac{|U|}{(1-\epsilon)K}\right) + C_{\text{LOG}}|V| \log\left(\frac{|V|}{K\epsilon}\right) + \frac{\epsilon C_{\text{LOG}} |U|}{1 - \epsilon}.$$

$\square$

**Lemma A.1.6** (PRECOST for $\beta$-Tsallis Entropy). *For any $\beta \in (0,1)$, when using the $\beta$-Tsallis entropy regularizer with $\alpha = \beta$ and $\theta = \sqrt{\frac{1-\beta}{\beta}}$, [Algorithm 1] ensures*

- *[Eq. (17)] with $B_1 = \frac{2\epsilon KT}{\sqrt{\beta(1-\beta)}}$;*

- *[Eq. (18)] with $B_2 = 0$;*

- $D^1(\widetilde{q}^1, p^1) \leq \frac{K}{\sqrt{\beta(1-\beta)}} + C_{\text{LOG}}|V| \log\left(\frac{|V|}{K\epsilon}\right) + \frac{\epsilon C_{\text{LOG}}|U|}{1-\epsilon}.$

*Therefore, according to [Lemma A.1.3], by picking $\epsilon = \frac{1}{T}$, our algorithm further ensures*

$$\text{PRECOST} = \mathcal{O}\left(\frac{K}{\sqrt{\beta(1-\beta)}} + C_{\text{LOG}} K \log T + D\right).$$

*Proof.* Let us first show that [Eq. (18)] holds. For the entries in $V$, we have

$$\phi_V^{t+1}(\widetilde{q}^{t+1}) - \phi_V^t(\widetilde{q}^t) = -\frac{1}{1-\beta} \sum_{i \in V} \left(\gamma_i^{t+1} \left(\widetilde{q}_i^{t+1}\right)^\beta - \gamma_i^t \left(\widetilde{q}_i^t\right)^\beta\right) - C_{\text{LOG}} \cdot \sum_{i \in V} \left(\log \widetilde{q}_i^{t+1} - \log \widetilde{q}_i^t\right)$$

$$= -\frac{1}{1-\beta} \sum_{i \in V} \left(\gamma_i^{t+1} - \gamma_i^t\right) \cdot \left(\frac{\epsilon}{|V|}\right)^\beta \leq 0,$$

where the second step follows from the definition of $\widetilde{q}^t$ in [Eq. (15)] that $\widetilde{q}_i^t = \frac{\epsilon}{|V|}$ for any $i \in V$. Since $\phi_V^{t+1}(\widetilde{q}^{t+1}) - \phi_V^t(\widetilde{q}^t) \leq 0$, we have $\sum_{t=1}^T \phi_V^{t+1}(\widetilde{q}^{t+1}) - \phi_V^t(\widetilde{q}^t) \leq 0$, which gives $B_2 = 0$ for [Eq. (18)].

For the entries in $U$, we have

$$\phi_U^{t+1}(\widetilde{q}^{t+1}) - \phi_U^t(\widetilde{q}^t)$$

$$= -\frac{1}{1-\beta} \sum_{i \in U} \left(\gamma_i^{t+1} \left(\widetilde{q}_i^{t+1}\right)^\beta - \gamma_i^t \left(\widetilde{q}_i^t\right)^\beta\right) - C_{\text{LOG}} \cdot \sum_{i \in U} \left(\log \widetilde{q}_i^{t+1} - \log \widetilde{q}_i^t\right)$$

$$= -\frac{(1-\epsilon)^\beta}{1-\beta} \sum_{i \in U} \left(\gamma_i^{t+1} \left(q_i^{t+1}\right)^\beta - \gamma_i^t \left(q_i^t\right)^\beta\right) - C_{\text{LOG}} \cdot \sum_{i \in U} \left(\log(1-\epsilon) + \log q_i^{t+1} - \log(1-\epsilon) - \log q_i^t\right)$$

$$= -\frac{(1-\epsilon)^\beta}{1-\beta} \sum_{i \in U} \left(\gamma_i^{t+1} \left(q_i^{t+1}\right)^\beta - \gamma_i^t \left(q_i^t\right)^\beta\right) - C_{\text{LOG}} \cdot \sum_{i \in U} \left(\log q_i^{t+1} - \log q_i^t\right)$$

$$= \phi_U^{t+1}(q^{t+1}) - \phi_U^t(q^t) + \frac{1 - (1-\epsilon)^\beta}{1-\beta} \sum_{i \in U} \left( \gamma_i^{t+1} \left( q_i^{t+1} \right)^\beta - \gamma_i^t \left( q_i^t \right)^\beta \right).$$

By summing over all rounds and direct calculation, we have

$$\mathbb{E}\left[ \sum_{t=1}^T \frac{1 - (1-\epsilon)^\beta}{1-\beta} \sum_{i \in U} \left( \gamma_i^{t+1} \left( q_i^{t+1} \right)^\beta - \gamma_i^t \left( q_i^t \right)^\beta \right) \right]$$

$$= \mathbb{E}\left[ \frac{1 - (1-\epsilon)^\beta}{1-\beta} \sum_{i \in U} \left( \gamma_i^{T+1} \left( q_i^{T+1} \right)^\beta - \gamma_i^1 \left( q_i^1 \right)^\beta \right) \right]$$

$$\leq \mathbb{E}\left[ \frac{1 - (1-\epsilon)^\beta}{1-\beta} \sum_{i \in U} \gamma_i^{T+1} \right]$$

$$\leq \frac{1 - (1-\epsilon)^\beta}{1-\beta} \cdot 2KT \sqrt{\frac{1-\beta}{\beta}}$$

$$\leq \frac{2\epsilon KT}{\sqrt{\beta(1-\beta)}},$$

where the third step uses the fact that $\left( \max\{p_i^t, 1/T\} \right)^{1-2\beta} \leq \left( \max\{p_i^t, 1/T\} \right)^{-1} \leq T$,

$$\gamma_i^{T+1} = \sqrt{\frac{1-\beta}{\beta}} \cdot \sqrt{1 + \sum_{t=1}^T \left( \max\{p_i^t, 1/T\} \right)^{1-2\beta}} \leq \sqrt{\frac{1-\beta}{\beta}} \cdot \sqrt{1 + \sum_{t=1}^T T} \leq 2T \sqrt{\frac{1-\beta}{\beta}},$$

and the last step uses the fact that $(1-\epsilon)^\beta \geq 1 - \epsilon$ for any $\beta \in (0,1)$.

Finally, we have by direct calculation:

$$D^1(\widetilde{q}^1, p^1) = -\frac{1}{1-\beta} \sum_{i \in [K]} \gamma_i^1 \left( \left( \widetilde{q}_i^1 \right)^\beta - \left( p_i^1 \right)^\beta \right) - C_{\text{LOG}} \sum_{i \in [K]} \left( \log\left( \widetilde{q}_i^1 \right) - \log\left( p_i^1 \right) \right)$$

$$\leq \frac{1}{1-\beta} \sum_{i \in [K]} \gamma_i^1 + C_{\text{LOG}} \sum_{i \in [K]} \log\left( \frac{1}{K\widetilde{q}_i^1} \right) + \frac{\epsilon C_{\text{LOG}} |U|}{1-\epsilon}$$

$$\leq \frac{1}{1-\beta} \sum_{i \in [K]} \gamma_i^1 + C_{\text{LOG}}|V| \log\left( \frac{|V|}{K\epsilon} \right) + \frac{\epsilon C_{\text{LOG}} |U|}{1-\epsilon}$$

$$= \frac{K}{\sqrt{\beta(1-\beta)}} + C_{\text{LOG}}|V| \log\left( \frac{|V|}{K\epsilon} \right) + \frac{\epsilon C_{\text{LOG}} |U|}{1-\epsilon},$$

where the first step uses the fact that $\langle \phi^1(p^1), \widetilde{q}^1 - p^1 \rangle = 0$, since $\nabla \phi^1(p^1) = c \cdot \mathbf{1}_K$ for some constant $c \in \mathbb{R}$ and $\widetilde{q}^1, p^1 \in \Omega_K$; the third and forth steps follow the definitions that $p^1 = \frac{1}{K} \cdot \mathbf{1}_K$, $\widetilde{q}_i^1 = \frac{\epsilon}{|V|}$ for any $i \in V$, and $\gamma_i^1 = \sqrt{\frac{1-\beta}{\beta}} \cdot \mathbf{1}_K$. $\qquad\square$

## A.2 Regret on Sub-Optimal Arms

One important property of our learning rate schedule is that:

$$\gamma_i^{t+1} - \gamma_i^t = \theta \cdot \frac{\left( \max\{p_i^t, 1/T\} \right)^{1-2\beta}}{\sqrt{1 + \sum_{k=1}^{t-1} \left( \max\{p_i^k, 1/T\} \right)^{1-2\beta}} + \sqrt{1 + \sum_{k=1}^{t} \left( \max\{p_i^k, 1/T\} \right)^{1-2\beta}}}$$

$$\leq \theta^2 \cdot \frac{\left( \max\{p_i^t, 1/T\} \right)^{1-2\beta}}{\theta \cdot \sqrt{1 + \sum_{k=1}^{t} \left( \max\{p_i^k, 1/T\} \right)^{1-2\beta}}}$$

$$= \theta^2 \cdot \frac{\left( \max\{p_i^t, 1/T\} \right)^{1-2\beta}}{\gamma_i^{t+1}}, \tag{21}$$

holds for any arm $i \in [K]$ and $t \in [T]$, which controls the increase of learning rates from round $t$ to $t + 1$.

On the other hand, our algorithm is somehow stabilized by the extra log-barrier, i.e., it ensures the following *multiplicative relation* between $p^t$, $\bar{p}^{t+1}$, and $p^{t+1}$:

$$\frac{1}{2} p_i^t \leq \bar{p}_i^{t+1} \leq 2p_i^t, \quad \frac{1}{2} p_i^t \leq p_i^{t+1} \leq 2p_i^t, \quad \forall t \in [T], \forall i \in [K], \tag{22}$$

according to Lemma C.3.2, Lemma C.3.3, and Lemma C.3.4.

Then, we are ready to analyze the regret related to the sub-optimal arms (REGSUB). For the skewed Bregman divergence $D_V^{t,t+1}\left(p^t, \bar{p}^{t+1}\right)$, we can decompose it into the stability and penalty terms as

$$
\begin{aligned}
D_V^{t,t+1}\left(p^t, \bar{p}^{t+1}\right) &= \left\langle p_V^t - \bar{p}_V^{t+1}, \nabla \phi_V^{t+1}(\bar{p}^{t+1}) - \nabla \phi_V^t(p^t) \right\rangle - D_V^{t+1,t}\left(\bar{p}^{t+1}, p^t\right) \\
&= \underbrace{\left\langle p_V^t - \bar{p}_V^{t+1}, \widehat{\ell}^t \right\rangle - D_V^t\left(\bar{p}^{t+1}, p^t\right)}_{\text{stability}} + \underbrace{\phi_V^t(\bar{p}^{t+1}) - \phi_V^{t+1}(\bar{p}^{t+1})}_{\text{penalty}}, \tag{23}
\end{aligned}
$$

where the first step follows from Lemma C.5.1, and the second step follows from the definition of skewed Bregman divergence.

**Lemma A.2.1** (Bound on Stability). *With $C_{\text{LOG}} \geq 162$, Algorithm 1 ensures*

$$\left\langle p_V^t - \bar{p}_V^{t+1}, \widehat{\ell}^t \right\rangle - D_V^t\left(\bar{p}^{t+1}, p^t\right) \leq \mathcal{O}\left(\left\|\widehat{\ell}^t\right\|_{\nabla^{-2}\phi_V^t(p^t)}^2\right),$$

*where $\|x\|_M \triangleq \sqrt{x^\top M x}$ is the quadratic norm of $x \in \mathbb{R}^K$ with respect to some positive semi-definite matrix $M \in \mathbb{R}^{K \times K}$, and $\nabla^{-2}\phi_V^t(p^t)$ denotes the Moore–Penrose inverse of the Hessian $\nabla^2\phi_V^t(\cdot)$.*

Specifically, the Hessian $\nabla^2\phi_V^t(x)$ is a diagonal matrix:

$$\nabla^2\phi_V^t(x) = \text{diag}\left\{\frac{\partial^2 \phi_V^t(x)}{\partial (x_i)^2} : \forall i \in [K]\right\},$$

where $\frac{\partial^2 \phi_V^t(x)}{\partial (x_i)^2}$ is the second order derivative of $\phi_V^t(x)$ with respect to the variable $x_i$. According to the definition of $\phi_V^t(\cdot)$, $\frac{\partial^2 \phi_V^t(x)}{\partial (x_i)^2} = 0$ for any arm $i \in U$. Therefore, the Moore–Penrose inverse of $\nabla^2\phi_V^t(x)$ is also a diagonal matrix, and it holds that

$$\left(\nabla^{-2}\phi_V^t(x)\right)_{i,i} = \begin{cases} \left(\frac{\partial^2 \phi_V^t(x)}{\partial (x_i)^2}\right)^{-1}, & i \in V, \\ 0, & i \in U, \end{cases}$$

where $\left(\nabla^{-2}\phi_V^t(x)\right)_{i,i}$ denotes the $i$-th diagonal element of the matrix $\nabla^{-2}\phi_V^t(x)$.

*Proof.* To prove this result, we introduce the minimizer $z \in \mathbb{R}_{\geq 0}^K$ defined as:

$$z = \underset{\substack{x_i = 0, \forall i \in U \\ \sum_{i \in V} x_i = \sum_{i \in V} p_i^t}}{\text{argmin}} \left\langle \sum_{\tau \leq t} \widehat{\ell}_V^\tau, x \right\rangle + \phi_V^t(x). \tag{24}$$

By Lemma C.3.5, the extra log-barrier also guarantees the multiplicative relation between $p^t$ and $z$, that is, $z_i/2 \leq p_i^t \leq 2z_i$ holds for any arm $i \in V$.

According to Lemma C.5.2, we then have

$$\left\langle p_V^t - \bar{p}_V^{t+1}, \widehat{\ell}^t \right\rangle - D_V^t\left(\bar{p}^{t+1}, p^t\right) \leq \left\langle p_V^t - z, \widehat{\ell}^t \right\rangle - D_V^t\left(z, p^t\right) = \mathcal{O}\left(\left\|\widehat{\ell}_V^t\right\|_{\nabla^{-2}\phi_V^t(\xi)}^2\right),$$

where $\xi$ is some intermediate point between $z$ and $p^t$, that is, $\xi = \rho z + (1 - \rho) p^t$ for some $\rho \in [0, 1]$. With the multiplicative relation between $z$ and $p^t$, we have $\frac{1}{2} p_i^t \leq \xi_i \leq 2p_i^t$ for any arm $i \in V$.

Therefore, one can verify

$$\frac{1}{4} \cdot \frac{\partial^2 \phi_V^t(p_i^t)}{\partial (p_i^t)^2} \leq \frac{\partial^2 \phi_V^t(\xi)}{\partial (\xi_i)^2} \leq 4 \cdot \frac{\partial^2 \phi_V^t(p_i^t)}{\partial (p_i^t)^2}, \quad \forall i \in V, \tag{25}$$

for all the regularizers of Algorithm 1. Thus, we can further bound $\left\|\widehat{\ell}_V^t\right\|_{\nabla^{-2}\phi_V^t(\xi)}^2$ as:

$$\left\|\widehat{\ell}^t\right\|_{\nabla^{-2}\phi_V^t(\xi)}^2 = \sum_{i \in V} \left(\widehat{\ell}_i^t\right)^2 \left(\frac{\partial^2 \phi_V^t(\xi)}{\partial (\xi_i)^2}\right)^{-1} \leq 4 \sum_{i \in V} \left(\widehat{\ell}_i^t\right)^2 \left(\frac{\partial^2 \phi_V^t(p_i^t)}{\partial (p_i^t)^2}\right)^{-1} = 4 \left\|\widehat{\ell}_V^t\right\|_{\nabla^{-2}\phi_V^t(p_V^t)}^2,$$

where the first step follows from the definition of quadratic norm, and the second step follows from Eq. (25). □

**Lemma A.2.2** (REGSUB for Log-barrier). *When using the log-barrier regularizer with $C_{\text{LOG}} = 162$, $\alpha = 0$, and $\theta = \sqrt{1/\log T}$, Algorithm 1 ensures $\text{REGSUB} = \mathcal{O}\left(\mathbb{S}_2\left(\log T\right)\right)$.*

*Proof.* For the log-barrier regularizer, the multiplicative relation between $p^t, \bar{p}^{t+1}$, and $p^{t+1}$ in Eq. (22) is guaranteed by Lemma C.3.2.

We first consider the stability term. By Lemma A.2.1, we have

$$\mathbb{E}^t\left[\left\langle p_V^t - \bar{p}_V^{t+1}, \widehat{\ell}^t\right\rangle - D_V^t\left(\bar{p}^{t+1}, p^t\right)\right] \leq \mathcal{O}\left(\mathbb{E}^t\left[\sum_{i \in V} \frac{(p_i^t)^2 \left(\widehat{\ell}_i^t\right)^2}{\gamma_i^t}\right]\right) = \mathcal{O}\left(\sum_{i \in V} \frac{p_i^t}{\gamma_i^t}\right),$$

where the first step follows from the facts that the Hessian $\nabla^2 \phi_V^t(p^t)$ is a diagonal matrix and its diagonal element for arm $i \in V$ is $\frac{C_{\text{LOG}} + \gamma_i^t}{(p_i^t)^2}$.

On the other hand, we bound the penalty term $\phi_V^t(\bar{p}^{t+1}) - \phi_V^{t+1}(\bar{p}_V^{t+1})$ as

$$\phi_V^t(\bar{p}^{t+1}) - \phi_V^{t+1}(\bar{p}^{t+1}) = \sum_{i \in V} \gamma_i^t \log\left(\frac{1}{\bar{p}_i^{t+1}}\right) - \sum_{i \in V} \gamma_i^{t+1} \log\left(\frac{1}{\bar{p}_i^{t+1}}\right)$$

$$= \sum_{i \in V} \left(\gamma_i^t - \gamma_i^{t+1}\right) \log\left(\frac{1}{\bar{p}_i^{t+1}}\right)$$

$$\leq 0,$$

where the third step follows from the fact that $\gamma_i^t \leq \gamma_i^{t+1}$ for any arm $i$. Therefore, we have REGSUB bounded as

$$\text{REGSUB} = \mathbb{E}\left[\sum_{t=1}^{T} \left\langle p_V^t - \bar{p}_V^{t+1}, \widehat{\ell}^t\right\rangle - D_V^t\left(\bar{p}^{t+1}, p^t\right) + \phi_V^t(\bar{p}^{t+1}) - \phi_V^{t+1}(\bar{p}^{t+1})\right]$$

$$\leq \mathcal{O}\left(\mathbb{E}\left[\sum_{t=1}^{T} \sum_{i \in V} \frac{p_i^t}{\gamma_i^t}\right]\right) = \mathcal{O}\left(\mathbb{E}\left[\sum_{t=1}^{T} \sum_{i \in V} \frac{p_i^t}{\gamma_i^{t+1}}\right]\right),$$

where the last step follows from the multiplicative relation $p_i^{t+1} \leq 2p_i^t$ for any arm $i$.

By our learning rate schedule, we further have

$$\mathbb{E}\left[\sum_{t=1}^{T} \sum_{i \in V} \frac{p_i^t}{\gamma_i^{t+1}}\right] \leq \mathbb{E}\left[\sqrt{\log T} \sum_{t=1}^{T} \sum_{i \in V} \frac{p_i^t}{\sqrt{1 + \sum_{s=1}^{t} p_i^s}}\right]$$

$$\leq \mathbb{E}\left[\sqrt{\log T} \sum_{i \in V} \sum_{t=1}^{T} \int_{\sum_{s=1}^{t-1} p_i^s}^{\sum_{s=1}^{t} p_i^s} \frac{du}{\sqrt{1+u}}\right]$$

$$\leq 2\mathbb{E}\left[\sqrt{\log T}\sum_{i\in V}\sqrt{1+\sum_{t=1}^{T}p_i^t}\right]$$

$$= \mathcal{O}\left(\mathbb{S}_2\left(\log T\right)\right),$$

where the first step follows from the fact $p_i^t \leq \max\{p_i^t, 1/T\}$; the third step uses the Newton-Leibniz formula $\int_a^b \frac{du}{\sqrt{1+u}} = 2\sqrt{1+u}\big|_a^b$. $\qquad\square$

**Lemma A.2.3** (REGSUB for $\beta$-Tsallis Entropy). *For any $\beta \in (0,1)$, when using the $\beta$-Tsallis entropy regularizer with $C_{\text{LOG}} = \frac{162\beta}{1-\beta}$, $\alpha = \beta$, and $\theta = \sqrt{\frac{1-\beta}{\beta}}$, Algorithm 1 ensures*

$$\text{REGSUB} = \mathcal{O}\left(\mathbb{S}_2\left(\frac{\log T}{\beta\left(1-\beta\right)}\right)\right).$$

*Proof.* First, the multiplicative relation between $p^t, \bar{p}^{t+1}$, and $p^{t+1}$ in Eq. (22) is ensured by Lemma C.3.2. According to Lemma A.2.1, we have

$$\mathbb{E}^t\left[\left\langle p_V^t - \bar{p}_V^{t+1}, \widehat{\ell}^t\right\rangle - D_V^t\left(\bar{p}^{t+1}, p^t\right)\right] \leq \mathcal{O}\left(\mathbb{E}\left[\sum_{i\in V}\frac{(p_i^t)^{2-\beta}\left(\widehat{\ell}_i^t\right)^2}{\beta\gamma_i^t}\right]\right) = \mathcal{O}\left(\sum_{i\in V}\frac{(p_i^t)^{1-\beta}}{\beta\gamma_i^t}\right),$$

since the diagonal of $\nabla^2\phi_V^t(p^t)$ is $\frac{\beta\gamma_i^t}{(p_i^t)^{2-\beta}} + \frac{C_{\text{LOG}}}{(p_i^t)^2}$ for any arm $i \in V$.

Therefore, we have REGSUB bounded as

$$\text{REGSUB} = \mathbb{E}\left[\sum_{t=1}^{T}\mathbb{E}^t\left[\frac{(p_i^t)^{1-\beta}}{\beta\gamma_i^t} + \phi_V^t(\bar{p}^{t+1}) - \phi_V^{t+1}(\bar{p}^{t+1})\right]\right]$$

$$\leq \mathcal{O}\left(\mathbb{E}\left[\sum_{t=1}^{T}\sum_{i\in V}\left(\frac{(p_i^t)^{1-\beta}}{\beta\gamma_i^t} + \frac{(\gamma_i^{t+1} - \gamma_i^t)(\bar{p}_i^{t+1})^{\beta}}{1-\beta}\right)\right]\right)$$

$$\leq \mathcal{O}\left(\mathbb{E}\left[\sum_{t=1}^{T}\sum_{i\in V}\left(\frac{(p_i^t)^{1-\beta}}{\beta\gamma_i^t} + \frac{(\gamma_i^{t+1} - \gamma_i^t)(p_i^t)^{\beta}}{1-\beta}\right)\right]\right),$$

where the last step follows from the multiplicative relation between $p^t$ and $\bar{p}^{t+1}$ in Eq. (22). Let $p_i^0 = \frac{1}{K}$ for any arm $i$ for notational convenience. For the stability term, we have

$$\sum_{i\in V}\frac{(p_i^t)^{1-\beta}}{\beta\gamma_i^t} \leq \sum_{i\in V}\frac{(2p_i^{t-1})^{1-\beta}}{\beta\gamma_i^t} \leq 2\sum_{i\in V}\frac{\max\{p_i^{t-1}, 1/T\}^{1-\beta}}{\beta\gamma_i^t}$$

where the first step follows from the multiplicative relation between $p_i^{t+1}$ and $p_i^t$. On the other hand, we can bound the penalty term as:

$$\sum_{i\in V}\frac{(\gamma_i^{t+1} - \gamma_i^t)(p_i^t)^{\beta}}{1-\beta} \leq \sum_{i\in V}\frac{(p_i^t)^{\beta}}{1-\beta}\cdot\frac{1-\beta}{\beta}\cdot\frac{(\max\{p_i^t, 1/T\})^{1-2\beta}}{\gamma_i^{t+1}}$$

$$= \sum_{i\in V}\frac{(p_i^t)^{\beta}(\max\{p_i^t, 1/T\})^{1-2\beta}}{\beta\gamma_i^{t+1}} \leq \sum_{i\in V}\frac{(\max\{p_i^t, 1/T\})^{1-\beta}}{\beta\gamma_i^{t+1}},$$

where the first step uses Eq. (21). This shows that the stability term and the penalty term are of the same order, and the rest of the proof boils down to the following final calculation:

$$\sum_{t=1}^{T}\sum_{i\in V}\frac{(\max\{p_i^t, 1/T\})^{1-\beta}}{\beta\gamma_i^{t+1}} \leq \frac{1}{\beta}\cdot\sqrt{\frac{\beta}{1-\beta}}\sum_{t=1}^{T}\sum_{i\in V}\frac{(\max\{p_i^t, 1/T\})^{1-\beta}}{\sqrt{1+\sum_{k=1}^{t}\left(\max\{p_i^k, 1/T\}\right)^{1-2\beta}}}$$

$$= \mathcal{O}\left(\sqrt{\frac{1}{\beta\left(1-\beta\right)}}\sum_{i\in V}\sqrt{\log T\sum_{t=1}^{T}\max\left\{p_i^t, 1/T\right\}}\right)$$

$$= \mathcal{O}\left(\sqrt{\frac{\log T}{\beta\left(1-\beta\right)}}\sum_{i\in V}\sqrt{\sum_{t=1}^{T}p_i^t}\right),$$

where the first step follows from the definition of $\gamma_i^t$; the second step applies Lemma 4.2 ; the last step follows from the fact that $\max\left\{p_i^t, 1/T\right\} \le p_i^t + \frac{1}{T}$. $\qquad\square$

**Lemma A.2.4** (REGSUB for Shannon Entropy). *When using the Shannon entropy regularizer with* $C_{\text{LOG}} = 162\log K$, $\alpha = 1$, *and* $\theta = \sqrt{1/\log T}$, *Algorithm 1 ensures* $\text{REGSUB} = \mathcal{O}\left(\mathbb{S}_2\left(\log^2 T\right)\right)$.

*Proof.* Similarly, we apply Lemma A.2.1 and have

$$\mathbb{E}^t\left[\left\langle p_V^t - \bar{p}_V^{t+1}, \widehat{\ell}^t\right\rangle - D_V^t\left(\bar{p}^{t+1}, p^t\right)\right] \le \mathcal{O}\left(\mathbb{E}^t\left[\sum_{i\in V}\frac{p_i^t\left(\widehat{\ell}_i^t\right)^2}{\gamma_i^t}\right]\right) = \mathcal{O}\left(\sum_{i\in V}\frac{1}{\gamma_i^t}\right),$$

since the diagonal of $\nabla^2\phi_V^t(p^t)$ is $\frac{\gamma_i^t}{p_i^t} + \frac{C_{\text{LOG}}}{\left(p_i^t\right)^2}$ for arm $i$.

For the penalty term, we have $\phi_V^t(\bar{p}^{t+1}) - \phi_V^{t+1}(\bar{p}_V^{t+1})$ bounded by

$$\sum_{i\in V}\left(\gamma_i^t - \gamma_i^{t+1}\right)\bar{p}_i^{t+1}\log\left(\frac{\bar{p}_i^{t+1}}{e}\right)$$

$$= \sum_{i\in V}\left(\gamma_i^{t+1} - \gamma_i^t\right)\bar{p}_i^{t+1}\log\left(\frac{e}{\bar{p}_i^{t+1}}\right)$$

$$\le \sum_{i\in V}\left(\gamma_i^{t+1} - \gamma_i^t\right)\max\left\{\bar{p}_i^{t+1}, 1/T\right\}\log\left(\frac{e}{\max\left\{\bar{p}_i^{t+1}, 1/T\right\}}\right)$$

$$\le \sum_{i\in V}\frac{\left(\max\left\{p_i^t, 1/T\right\}\right)^{-1}\max\left\{\bar{p}^{t+1}, 1/T\right\}}{(\log T)\gamma_i^{t+1}}\log\left(\frac{e}{\max\left\{\bar{p}_i^{t+1}, 1/T\right\}}\right)$$

$$\le \mathcal{O}\left(\sum_{i\in V}\frac{1}{\gamma_i^{t+1}}\right),$$

where the second step uses the fact that $x\log\left(\frac{e}{x}\right)$ is monotonically increasing for $x \in [0, 1/T]$; the third step follows from Eq. (21); the last step follows from the multiplicative relation between $p^t$ and $\bar{p}^{t+1}$ so that $\max\left\{p_i^t, 1/T\right\} \ge \frac{1}{2}\max\left\{\bar{p}_i^{t+1}, 1/T\right\}$.

Therefore, we have REGSUB bounded as

$$\text{REGSUB} = \mathbb{E}\left[\sum_{t=1}^{T}\mathbb{E}^t\left[\left\langle p_V^t - \bar{p}_V^{t+1}, \widehat{\ell}^t\right\rangle - D_V^t\left(\bar{p}^{t+1}, p^t\right) + \phi_V^t(\bar{p}^{t+1}) - \phi_V^{t+1}(\bar{p}^{t+1})\right]\right]$$

$$\le \mathcal{O}\left(\mathbb{E}\left[\sum_{t=1}^{T}\sum_{i\in V}\frac{1}{\gamma_i^t} + \frac{1}{\gamma_i^{t+1}}\right]\right) = \mathcal{O}\left(\mathbb{E}\left[\sum_{t=1}^{T}\sum_{i\in V}\frac{1}{\gamma_i^{t+1}}\right]\right),$$

by combining the bounds for the stability and penalty terms.

According to our learning rate schedule, we have

$$\mathbb{E}\left[\sum_{t=1}^{T}\sum_{i\in V}\frac{1}{\gamma_i^{t+1}}\right] = \mathbb{E}\left[\sqrt{\log T}\sum_{i\in V}\sum_{t=1}^{T}\frac{1}{\sqrt{1 + \sum_{k=1}^{t}\max\left\{p_i^k, 1/T\right\}^{-1}}}\right]$$

$$\leq \mathcal{O}\left(\mathbb{E}\left[\sqrt{\log T}\sum_{i\in V}\sqrt{\log\left(1+\sum_{t=1}^{T}\frac{1}{\max\{p_i^t, 1/T\}}\right)\sum_{t=1}^{T}\max\{p_i^t, 1/T\}}\right]\right)$$

$$\leq \mathcal{O}\left(\mathbb{E}\left[\sqrt{\log T}\sum_{i\in V}\sqrt{\log\left(1+T\cdot T\right)\left(1+\sum_{t=1}^{T}p_i^t\right)}\right]\right)$$

$$= \mathcal{O}\left(\mathbb{S}_2\left(\log^2 T\right)\right),$$

where the second step applies Lemma 4.2; the third step uses the fact that $\max\{p_i^t, 1/T\} \geq 1/T$. $\quad\square$

## A.3 Regret on Optimal Arms

We show that REGOPT $\leq 0$ holds for all regularizers used in this paper with a specific technique.

**Lemma A.3.1** (REGOPT for All Regularizers)**.** *For any of the three regularizers, Algorithm 1 ensures* REGOPT $\leq 0$.

*Proof.* According to the KKT conditions of the optimization problem for $q^t$, we have

$$\nabla\phi_U^{t+1}(q^{t+1}) = \nabla\phi_U^t(q^t) - \widehat{\ell}_U^t + \xi'\cdot\mathbf{1}_U,$$

where $\xi' \in \mathbb{R}$ is a Lagrange multiplier for $q^{t+1}$. Let $z \in \mathbb{R}_{>0}^K$ be a vector that satisfies

$$\nabla\phi_U^{t+1}(z) = \nabla\phi_U^t(p^t) - \widehat{\ell}_U^t + \xi'\cdot\mathbf{1}_U.$$

By definition, we have

$$\nabla\phi_U^{t+1}(z) - \nabla\phi_U^t(p^t) = \nabla\phi_U^{t+1}(q^{t+1}) - \nabla\phi_U^t(q^t) = -\widehat{\ell}_U^t + \xi'\cdot\mathbf{1}_U. \qquad (26)$$

Also, by [Ito, 2021, Lemma 18], we have

$$D_U^{t,t+1}(p^t, \bar{p}^{t+1}) = D_U^{t,t+1}(p^t, z) - D_U^{t+1}(\bar{p}^{t+1}, z) \leq D_U^{t,t+1}(p^t, z),$$

where the last step uses the fact that the Bregman divergence is non-negative. Hence, to show $D_U^{t,t+1}(p^t, \bar{p}^{t+1}) \leq D_U^{t,t+1}(q^t, q^{t+1})$, it suffices to show $D_U^{t,t+1}(p^t, z) \leq D_U^{t,t+1}(q^t, q^{t+1})$, which is done below by applying $x = p_i^t, y = z, m = q_i^t, n = q_i^{t+1}$, and $\xi = -\widehat{\ell}_i^t + \xi'$ in Theorem 4.3.

First, we show that $p_i^t \leq q_i^t$ for any arm $i \in U$. By the KKT conditions for $q^t$ and $p^t$, we know that $\nabla\phi_U^t(q^t) = \nabla\phi_U^t(p_U^t) + \eta\cdot\mathbf{1}_U$ for a Lagrange multiplier $\eta \in \mathbb{R}$. Due to the facts that $\sum_{i\in U}q_i^t = 1 \geq \sum_{i\in U}p_i^t$ and $\nabla\phi^t(\cdot)$ is monotone-increasing, $\eta$ is non-negative and $q_i^t \geq p_i^t$ holds for any arm $i \in U$.

It is clear that all our regularizers can be written in the form of $\phi^t(p) = \sum_{i\in[K]}f^t(p_i)$, and we will apply Theorem 4.3 with such $f^t : [0, 1] \to \mathbb{R}$. It suffices to check the two required conditions on $f^t$. First, all regularizers used in Algorithm 1 satisfy condition (i): $(f^t)'(z)$ is differentiable and concave. Second, as mentioned in Section 4, we only need to show that the learning rate $\gamma_i^t$ is non-decreasing in $t$ (which is true in our case) and the regularizer itself is non-increasing, the latter of which can be verifed for all our regularizers as long as we have $p^t, z, q^t, q^{t+1} \in (0, 1]^K$. As $z \in \mathbb{R}_{>0}^K$, we still need to check $z_i \leq 1$. Indeed, from Eq. (26), we have $\nabla\phi^{t+1}(z) \leq \nabla\phi^{t+1}(q^{t+1})$ (entry-wise inequality), and since $q_i^{t+1} \leq 1$ for $\forall i \in U$ and $\nabla\phi^{t+1}(\cdot)$ is monotone-increasing entry-wise, we have $z_i \leq 1$. To sum up, Theorem 4.3 indeed implies

$$D_U^{t,t+1}(p^t, z) \leq D_U^{t,t+1}(q^t, q^{t+1}).$$

which further implies $D_U^{t,t+1}(p^t, \bar{p}^{t+1}) \leq D_U^{t,t+1}(q^t, q^{t+1})$. Taking the summation over all the rounds concludes the proof. $\quad\square$

**Theorem A.3.2** (Restatement of Theorem 4.3)**.** *For any $t \in \mathbb{N}$, let $f^t : \Omega \to \mathbb{R}$ be a continuously-differentiable and strictly-convex function defined on $\Omega \subseteq \mathbb{R}$. Suppose that the following two conditions hold for all $z \in \Omega$: (i) $(f^t)'(z)$ is differentiable and concave; (ii) $(f^{t+1})'(z) \leq (f^t)'(z)$. Then, for any $x, m \in \mathbb{R}$ with $x \leq m$, and $y, n \in \mathbb{R}$ such that $(f^{t+1})'(y) - (f^t)'(x) = (f^{t+1})'(n) - (f^t)'(m) = \xi$ for a fixed scalar $\xi$, we have $D^{t,t+1}(x, y) \leq D^{t,t+1}(m, n)$, where $D^{t,t+1}(u, v) = f^t(u) - f^{t+1}(v) - (u - v)\cdot\left(f^{t+1}\right)'(v)$ is the skewed Bregman divergence.*

*Proof of Theorem 4.3.* For shorthand, we denote $F^t(z) = (f^t)'(z)$ and use $(F^t)'(z)$ to denote the first-order derivative of $F^t$ with respect to $z$. Taking the derivative with respect to $x$ on both sides of $(f^{t+1})'(y) - (f^t)'(x) = \xi$, we have

$$(F^t)'(x) = (F^{t+1})'(y)\frac{dy}{dx}. \tag{27}$$

We now see $y$ as a function of $x$ and $D^{t,t+1}(x,y) = f^t(x) - f^{t+1}(y) - (x - y)F^{t+1}(y)$ as a function of $x$ as well. Taking the derivative with respect to $x$, we have

$$
\begin{aligned}
&(D^{t,t+1})'(x,y) \\
&= F^t(x) - F^{t+1}(y)\frac{dy}{dx} - F^{t+1}(y) - x(F^{t+1})'(y)\frac{dy}{dx} + F^{t+1}(y)\frac{dy}{dx} + y(F^{t+1})'(y)\frac{dy}{dx} \\
&= F^t(x) - F^{t+1}(y) - x(F^{t+1})'(y)\frac{dy}{dx} + y(F^{t+1})'(y)\frac{dy}{dx} \\
&\geq F^t(x) - F^t(y) - x(F^{t+1})'(y)\frac{dy}{dx} + y(F^{t+1})'(y)\frac{dy}{dx} && (F^{t+1}(y) \leq F^t(y)) \\
&= F^t(x) - F^t(y) + (y - x)(F^t)'(x) && \text{(by Eq. (27))} \\
&\geq 0. && \text{(concavity of } F^t(\cdot))
\end{aligned}
$$

The implies $D^{t,t+1}(x,y) \leq D^{t,t+1}(m,n)$ given the condition $x \leq m$. $\qquad\square$

## A.4   Residual Regret

Throughout this section, we write all our regularizers in the form $\phi^t(x) = -C_{\text{LOG}}\sum_{i\in[K]}\log x_i + \sum_{i\in[K]}\gamma_i^t\psi(x_i)$ for a proper choice of $\psi(\cdot)$ (e.g., $\psi(x_i) = x_i\log\left(\frac{x_i}{e}\right)$ for the Shannon entropy regularizer). Also, we will present the Hessian $\nabla^2\phi^t(p^t)$ as a diagonal matrix whose diagonal element for arm $i$ is denoted as $w_i^t$, that is, $\nabla^2\phi^t(p^t) = \text{diag}\{w_i^t : i \in [K]\}$. Clearly, it always holds that $w_i^t = \frac{C_{\text{LOG}}}{(p_i^t)^2} + \gamma_i^t\psi''(p_i^t)$.

With these notations, we are now ready to introduce our decomposition of $\mathbb{E}^t\left[D^{t+1}(\bar{p}^{t+1}, p^{t+1})\right]$ into the term we discussed in Section 4 plus other terms.

**Lemma A.4.1.** *Under Eq. (22), Algorithm 1 ensures that $\mathbb{E}^t\left[D^{t+1}(\bar{p}^{t+1}, p^{t+1})\right]$ is bounded by*

$$
\mathcal{O}\left(\frac{\left(\sum_{i\in V}\frac{1}{w_i^t}\right)\left(\sum_{i\in U}\frac{1}{(w_i^t)^2 p_i^t}\right)}{\left(\sum_{i\in[K]}\frac{1}{w_i^t}\right)\left(\sum_{i\in U}\frac{1}{w_i^t}\right)} + \sum_{i\in V}\frac{1}{p_i^t w_i^t} + K\sum_{i\in[K]}\frac{\left(\gamma_i^t - \gamma_i^{t+1}\right)^2\left(\psi'(p_i^t)\right)^2}{w_i^t}\right).
$$

*Proof.* By Lemma C.5.2, for any $c \in \mathbb{R}$, we have $D^{t+1}(\bar{p}^{t+1}, p^{t+1})$ bounded as

$$
\begin{aligned}
&\left\langle\bar{p}^{t+1} - p^{t+1}, \nabla\phi^{t+1}(\bar{p}^{t+1}) - \nabla\phi^{t+1}(p^{t+1})\right\rangle - D^{t+1}(p^{t+1}, \bar{p}^{t+1}) \\
&= \left\langle\bar{p}^{t+1} - p^{t+1}, \nabla\phi^{t+1}(\bar{p}^{t+1}) - \nabla\phi^{t+1}(p^{t+1}) - c\cdot\mathbf{1}_K\right\rangle - D^{t+1}(p^{t+1}, \bar{p}^{t+1}) \\
&= 2\left\|\nabla\phi^{t+1}(\bar{p}^{t+1}) - \nabla\phi^{t+1}(p^{t+1}) - c\cdot\mathbf{1}_K\right\|_{\nabla^{-2}\phi^{t+1}(\xi)} \\
&= \mathcal{O}\left(\left\|\nabla\phi^{t+1}(\bar{p}^{t+1}) - \nabla\phi^{t+1}(p^{t+1}) - c\cdot\mathbf{1}_K\right\|_{\nabla^{-2}\phi^t(p^t)}\right),
\end{aligned}
$$

where the first step follows from the fact that $\bar{p}^{t+1}, p^{t+1} \in \Omega_K$, thus $\left\langle\bar{p}^{t+1} - p^{t+1}, c\cdot\mathbf{1}_K\right\rangle = 0$ holds for any $c \in \mathbb{R}$; the second step follows from Lemma C.5.2, where $\xi$ is some intermediate point between $\bar{p}^{t+1}$ and $p^{t+1}$; the thirds step follows from the multiplicative relation between $\bar{p}^{t+1}$ and $p^{t+1}$, and the fact that $\gamma_i^{t+1} \geq \gamma_i^t$ for all $t, i$.

Then, we analyze the difference $\nabla\phi^{t+1}(\bar{p}^{t+1}) - \nabla\phi^{t+1}(p^{t+1})$. According to the KKT conditions of the optimization problems in the FTRL framework, we have

$$
\begin{aligned}
\nabla\phi_U^{t+1}(\bar{p}^{t+1}) &= \nabla\phi_U^t(p^t) - \widehat{\ell}_U^t + \lambda_U\cdot\mathbf{1}_U, \\
\nabla\phi_V^{t+1}(\bar{p}^{t+1}) &= \nabla\phi_V^t(p^t) - \widehat{\ell}_V^t + \lambda_V\cdot\mathbf{1}_V,
\end{aligned}
$$

$$\nabla \phi_U^{t+1}(p^{t+1}) = \nabla \phi_U^t(p^t) - \widehat{\ell}_U^t + \lambda_K \cdot \mathbf{1}_U,$$
$$\nabla \phi_V^{t+1}(p^{t+1}) = \nabla \phi_V^t(p^t) - \widehat{\ell}_V^t + \lambda_K \cdot \mathbf{1}_V,$$

where $\lambda_U, \lambda_V, \lambda_K$ are corresponding Lagrange multipliers for the $\bar{p}^{t+1}$ and $p^{t+1}$.

To give a tighter bound of the Lagrange multipliers $\lambda_U$ and $\lambda_V$, we greatly extend the approach of Ito [2021] in Lemma C.4.3. Together with the multiplicative relation between $p^t, \bar{p}^{t+1}$ and $p^{t+1}$, we obtain $\lambda_U$'s upper and lower bounds:

$$\lambda_U \le \mathcal{O}\left( \left( \sum_{i \in U} \frac{1}{w_i^t} \right)^{-1} \left( \sum_{i \in U} \frac{\widehat{\ell}_i^t}{w_i^t} \right) \right),$$

$$\lambda_U \ge \mathcal{O}\left( -\left( \sum_{i \in U} \frac{1}{w_i^t} \right)^{-1} \left( \sum_{i \in U} \frac{\left( \gamma_i^t - \gamma_i^{t+1} \right) \psi'(p_i^t)}{w_i^t} \right) \right).$$

Similarly, we have the following upper and lower bounds of $\lambda_V$:

$$\lambda_V \le \mathcal{O}\left( \left( \sum_{i \in V} \frac{1}{w_i^t} \right)^{-1} \left( \sum_{i \in V} \frac{\widehat{\ell}_i^t}{w_i^t} \right) \right)$$

$$\lambda_V \ge \mathcal{O}\left( -\left( \sum_{i \in V} \frac{1}{w_i^t} \right)^{-1} \left( \sum_{i \in V} \frac{\left( \gamma_i^t - \gamma_i^{t+1} \right) \psi'(p_i^t)}{w_i^t} \right) \right).$$

With the help of these notations, for any $c \in \mathbb{R}$, we have

$$\left\| \nabla \phi^{t+1}(\bar{p}^{t+1}) - \nabla \phi^{t+1}(p^{t+1}) - c \cdot \mathbf{1}_K \right\|_{\nabla^{-2}\phi^t(p^t)}$$
$$= \left\| \lambda_U \cdot \mathbf{1}_U + \lambda_V \cdot \mathbf{1}_V - \lambda_K \cdot \mathbf{1}_K - c \cdot \mathbf{1}_K \right\|_{\nabla^{-2}\phi^t(p^t)}^2$$
$$= \left\| \lambda_U \cdot \mathbf{1}_U + \lambda_V \cdot \mathbf{1}_V - c' \cdot \mathbf{1}_K \right\|_{\nabla^{-2}\phi^t(p^t)}^2$$
$$= \sum_{i \in U} \frac{1}{w_i^t} \left( \lambda_U - c' \right)^2 + \sum_{i \in V} \frac{1}{w_i^t} \left( \lambda_V - c' \right)^2,$$

where in the second step we define $c' = c + \lambda_K$. Picking $c'$ to minimize this term yields

$$\mathbb{E}^t \left[ D^{t+1}(\bar{p}^{t+1}, p^{t+1}) \right]$$
$$\le \mathcal{O}\left( \mathbb{E}^t \left[ \frac{\left( \sum_{i \in V} \frac{1}{w_i^t} \right) \left( \sum_{i \in U} \frac{1}{w_i^t} \right)}{\sum_{i \in [K]} \frac{1}{w_i^t}} \left( \lambda_U - \lambda_V \right)^2 \right] \right)$$
$$= \mathcal{O}\left( \mathbb{E}^t \left[ \frac{\left( \sum_{i \in V} \frac{1}{w_i^t} \right) \left( \sum_{i \in U} \frac{1}{w_i^t} \right)}{\sum_{i \in [K]} \frac{1}{w_i^t}} \left( \lambda_U^2 + \lambda_V^2 \right) \right] \right)$$
$$= \mathcal{O}\left( \mathbb{E}^t \left[ \frac{\left( \sum_{i \in V} \frac{1}{w_i^t} \right) \left( \sum_{i \in U} \frac{1}{w_i^t} \right)}{\sum_{i \in [K]} \frac{1}{w_i^t}} \lambda_U^2 \right] + \mathbb{E}^t \left[ \frac{\left( \sum_{i \in V} \frac{1}{w_i^t} \right) \left( \sum_{i \in U} \frac{1}{w_i^t} \right)}{\sum_{i \in [K]} \frac{1}{w_i^t}} \lambda_V^2 \right] \right), \quad (28)$$

where the second step uses $(x - y)^2 \le 2(x^2 + y^2)$ for any $x, y \in \mathbb{R}$. In Eq. (28), the first term is only related to $\lambda_U$, and the second term depends on $\lambda_V$. By upper and lower bounds of $\lambda_U$ and $\lambda_V$,

we have Eq. (28) bounded as

$$
\mathcal{O}\left( \mathbb{E}^t \left[ \frac{\left(\sum_{i\in V}\frac{1}{w_i^t}\right)\left(\sum_{i\in U}\frac{\widehat{\ell}_i^t}{w_i^t}\right)^2}{\left(\sum_{i\in[K]}\frac{1}{w_i^t}\right)\left(\sum_{i\in U}\frac{1}{w_i^t}\right)} \right] + \frac{\left(\sum_{i\in U}\frac{(\gamma_i^t-\gamma_i^{t+1})\psi'(p_i^t)}{w_i^t}\right)^2}{\left(\sum_{i\in U}\frac{1}{w_i^t}\right)} \right)
$$
$$
+ \mathcal{O}\left( \mathbb{E}^t \left[ \frac{\left(\sum_{i\in U}\frac{1}{w_i^t}\right)\left(\sum_{i\in V}\frac{\widehat{\ell}_i^t}{w_i^t}\right)^2}{\left(\sum_{i\in[K]}\frac{1}{w_i^t}\right)\left(\sum_{i\in V}\frac{1}{w_i^t}\right)} \right] + \frac{\left(\sum_{i\in V}\frac{(\gamma_i^t-\gamma_i^{t+1})\psi'(p_i^t)}{w_i^t}\right)^2}{\left(\sum_{i\in V}\frac{1}{w_i^t}\right)} \right).
$$
(29)

Then, we further bound the first and the third term in Eq. (29). For the first term, we have

$$
\mathbb{E}^t \left[ \frac{\left(\sum_{i\in V}\frac{1}{w_i^t}\right)\left(\sum_{i\in U}\frac{\widehat{\ell}_i^t}{w_i^t}\right)^2}{\left(\sum_{i\in[K]}\frac{1}{w_i^t}\right)\left(\sum_{i\in U}\frac{1}{w_i^t}\right)} \right] = \mathbb{E}^t \left[ \frac{\left(\sum_{i\in V}\frac{1}{w_i^t}\right)\left(\sum_{i\in U}\left(\frac{\widehat{\ell}_i^t}{w_i^t}\right)^2\right)}{\left(\sum_{i\in[K]}\frac{1}{w_i^t}\right)\left(\sum_{i\in U}\frac{1}{w_i^t}\right)} \right]
$$
$$
\leq \frac{\left(\sum_{i\in V}\frac{1}{w_i^t}\right)\left(\sum_{i\in U}\frac{1}{(w_i^t)^2 p_i^t}\right)}{\left(\sum_{i\in[K]}\frac{1}{w_i^t}\right)\left(\sum_{i\in U}\frac{1}{w_i^t}\right)},
$$
(30)

where the first step uses the fact that $\widehat{\ell}_i^t \cdot \widehat{\ell}_j^t = 0$ for $i \neq j$, and the second step follows from the definition of importance-weighted loss estimator. Similarly, we have the third term bounded as

$$
\mathbb{E}^t \left[ \frac{\left(\sum_{i\in U}\frac{1}{w_i^t}\right)\left(\sum_{i\in V}\frac{\widehat{\ell}_i^t}{w_i^t}\right)^2}{\left(\sum_{i\in[K]}\frac{1}{w_i^t}\right)\left(\sum_{i\in V}\frac{1}{w_i^t}\right)} \right] \leq \frac{\left(\sum_{i\in V}\frac{1}{p_i^t(w_i^t)^2}\right)\left(\sum_{i\in U}\frac{1}{w_i^t}\right)}{\left(\sum_{i\in[K]}\frac{1}{w_i^t}\right)\left(\sum_{i\in V}\frac{1}{w_i^t}\right)} \leq \sum_{i\in V}\frac{1}{p_i^t w_i^t}.
$$
(31)

For the second term in Eq. (29), we have

$$
\frac{\left(\sum_{i\in U}\frac{(\gamma_i^t-\gamma_i^{t+1})\psi'(p_i^t)}{w_i^t}\right)^2}{\left(\sum_{i\in U}\frac{1}{w_i^t}\right)} \leq \frac{|U|\sum_{i\in U}\left(\frac{(\gamma_i^t-\gamma_i^{t+1})\psi'(p_i^t)}{w_i^t}\right)^2}{\left(\sum_{i\in U}\frac{1}{w_i^t}\right)} \leq K\sum_{i\in U}\frac{\left(\gamma_i^t-\gamma_i^{t+1}\right)^2\left(\psi'(p_i^t)\right)^2}{w_i^t},
$$
(32)

where the first step follows from the Cauchy-Schwarz inequality. Similarly, we have the following bound for the forth term

$$
\frac{\left(\sum_{i\in V}\frac{(\gamma_i^t-\gamma_i^{t+1})\psi'(p_i^t)}{w_i^t}\right)^2}{\left(\sum_{i\in V}\frac{1}{w_i^t}\right)} \leq K\sum_{i\in V}\frac{\left(\gamma_i^t-\gamma_i^{t+1}\right)^2\left(\psi'(p_i^t)\right)^2}{w_i^t}.
$$
(33)

Plugging Eq. (30), Eq. (31), Eq. (32), and Eq. (33) into Eq. (29) finishes the proof. $\square$

As mentioned earlier in Section 4, the first term in Lemma A.4.1 is the leading term in the stochastic setting which will yield a term of the order $\mathcal{O}\left(\frac{|U|\log T}{\Delta_{\text{MIN}}}\right)$ in the regret bound. In the following lemma, we further decompose this term into two parts where the first part is related to $\mathbb{S}_2$ and the second part is related to $\mathbb{S}_1$.

**Lemma A.4.2.** *The following holds:*

$$
\frac{\left(\sum_{i\in V}\frac{1}{w_i^t}\right)\left(\sum_{i\in U}\frac{1}{(w_i^t)^2 p_i^t}\right)}{\left(\sum_{i\in[K]}\frac{1}{w_i^t}\right)\left(\sum_{i\in U}\frac{1}{w_i^t}\right)} \leq \sum_{i\in V}\frac{1}{p_i^t w_i^t} + \sum_{i\in U}\frac{\mathbb{I}\left\{p_i^t \leq \sum_{j\in V}p_j^t\right\}}{p_i^t w_i^t}\left(\frac{\frac{1}{w_i^t}}{\sum_{i\in U}\frac{1}{w_i^t}}\right).
$$

*Proof.* We first rewrite the fraction on left hand side as:

$$\frac{\left(\sum_{i\in V}\frac{1}{w_i^t}\right)\left(\sum_{i\in U}\frac{1}{(w_i^t)^2 p_i^t}\right)}{\left(\sum_{i\in[K]}\frac{1}{w_i^t}\right)\left(\sum_{i\in U}\frac{1}{w_i^t}\right)} = \sum_{i\in U}\frac{1}{p_i^t w_i^t}\frac{\sum_{i\in V}\frac{1}{w_i^t}}{\sum_{i\in[K]}\frac{1}{w_i^t}}\left(\frac{\frac{1}{w_i^t}}{\sum_{i\in U}\frac{1}{w_i^t}}\right). \tag{34}$$

To bound the right-hand side of Eq. (34), we introduce an indicator function for any $t$ and any $i\in U$:

$$A_i^t = \mathbb{I}\left\{\frac{1}{p_i^t w_i^t}\frac{\sum_{j\in V}\frac{1}{w_j^t}}{\sum_{j\in[K]}\frac{1}{w_j^t}} \le \sum_{j\in V}\frac{1}{p_j^t w_j^t}\right\},$$

to separate the term into two parts as:

$$\underbrace{\sum_{i\in U}\frac{A_i^t}{p_i^t w_i^t}\frac{\sum_{i\in V}\frac{1}{w_i^t}}{\sum_{i\in[K]}\frac{1}{w_i^t}}\left(\frac{\frac{1}{w_i^t}}{\sum_{i\in U}\frac{1}{w_i^t}}\right)}_{\text{Case 1}}, \quad \underbrace{\sum_{i\in U}\frac{1-A_i^t}{p_i^t w_i^t}\frac{\sum_{i\in V}\frac{1}{w_i^t}}{\sum_{i\in[K]}\frac{1}{w_i^t}}\left(\frac{\frac{1}{w_i^t}}{\sum_{i\in U}\frac{1}{w_i^t}}\right)}_{\text{Case 2}}.$$

**Case 1.** By direct calculation, we have

$$\sum_{i\in U}\frac{A_i^t}{p_i^t w_i^t}\frac{\sum_{i\in V}\frac{1}{w_i^t}}{\sum_{i\in[K]}\frac{1}{w_i^t}}\left(\frac{\frac{1}{w_i^t}}{\sum_{i\in U}\frac{1}{w_i^t}}\right) \le \sum_{i\in U}\left(\sum_{j\in V}\frac{1}{p_j^t w_j^t}\right)\left(\frac{\frac{1}{w_i^t}}{\sum_{i\in U}\frac{1}{w_i^t}}\right) = \sum_{j\in V}\frac{1}{p_j^t w_j^t}.$$

**Case 2.** For arm $i$ that $A_i^t = 0$, we have

$$\frac{1}{p_i^t w_i^t}\frac{\sum_{j\in V}\frac{1}{w_j^t}}{\sum_{j\in[K]}\frac{1}{w_j^t}} > \sum_{j\in V}\frac{1}{p_j^t w_j^t} \Rightarrow \frac{1}{p_i^t w_i^t}\left(\frac{\sum_{j\in V}\frac{1}{w_j^t}}{\sum_{j\in V}\frac{1}{p_j^t w_j^t}}\right) > \sum_{j\in[K]}\frac{1}{w_j^t}$$

$$\Rightarrow \frac{1}{p_i^t w_i^t}\left(\sum_{j\in V}p_j^t\right) > \sum_{j\in[K]}\frac{1}{w_j^t}$$

$$\Rightarrow \frac{1}{p_i^t w_i^t}\left(\sum_{j\in V}p_j^t\right) > \frac{1}{w_i^t}$$

$$\Rightarrow \sum_{j\in V}p_j^t > p_i^t,$$

where the second step follows from the fact that $\left(\sum_{j\in V}p_j^t\right)\left(\sum_{j\in V}\frac{1}{p_j^t w_j^t}\right) \ge \sum_{j\in V}\frac{1}{w_j^t}$.

Therefore, we have

$$\sum_{i\in U}\frac{1-A_i^t}{p_i^t w_i^t}\frac{\sum_{i\in V}\frac{1}{w_i^t}}{\sum_{i\in[K]}\frac{1}{w_i^t}}\left(\frac{\frac{1}{w_i^t}}{\sum_{i\in U}\frac{1}{w_i^t}}\right) \le \sum_{i\in U}\frac{\mathbb{I}\left\{p_i^t \le \sum_{j\in V}p_j^t\right\}}{p_i^t w_i^t}\left(\frac{\frac{1}{w_i^t}}{\sum_{i\in U}\frac{1}{w_i^t}}\right),$$

where $1 - A_i^t = \mathbb{I}\{A_i^t = 0\} \le \mathbb{I}\left\{p_i^t \le \sum_{j\in V}p_j^t\right\}$ holds due to the fact that $\mathbb{I}\{A\} \le \mathbb{I}\{B\}$ for any events $A, B$ with $A \subseteq B$. Finally, combining bounds for these two cases finishes the proof. $\square$

Based on Lemma A.4.1 and the careful decomposition in Lemma A.4.2, we now bound RESREG by

$$\mathcal{O}\left(\mathbb{E}\left[\sum_{t=1}^{T}\sum_{i\in V}\frac{1}{p_i^t w_i^t} + \sum_{t=1}^{T}\sum_{i\in U}\frac{\mathbb{I}\left\{p_i^t \le \sum_{j\in V}p_j^t\right\}}{p_i^t w_i^t}\left(\frac{\frac{1}{w_i^t}}{\sum_{i\in U}\frac{1}{w_i^t}}\right)\right]\right)$$

$$+ \mathcal{O}\left(\mathbb{E}\left[K\sum_{t=1}^{T}\sum_{i\in[K]}\frac{\left(\gamma_i^t - \gamma_i^{t+1}\right)^2\left(\psi'(p_i^t)\right)^2}{w_i^t}\right]\right).$$

Then, we are ready to bound these three terms with different configurations of Algorithm 1 in Lemma A.4.3, Lemma A.4.4, and Lemma A.4.5. The corresponding $\psi'(x)$'s are $-\frac{\beta x^{\beta-1}}{1-\beta}$ for the $\beta$-Tsallis entropy regularizer, $-\frac{1}{x}$ for the log-barrier regularizer, and $\log(x)$ for the Shannon entropy regularizer.

**Lemma A.4.3** (RESREG for $\beta$-Tsallis Entropy). *For any $\beta \in (0,1)$, when using the $\beta$-Tsallis entropy regularizer with $C_{\text{LOG}} = \frac{162\beta}{1-\beta}$, $\alpha = \beta$, and $\theta = \sqrt{\frac{1-\beta}{\beta}}$, Algorithm 1 ensures:*

$$\mathbb{E}\left[\sum_{t=1}^{T}\sum_{i\in V}\frac{1}{p_i^t w_i^t}\right] = \mathcal{O}\left(\mathbb{S}_2\left(\frac{\log T}{\beta(1-\beta)}\right)\right),$$

$$\mathbb{E}\left[\sum_{t=1}^{T}\sum_{i\in U}\frac{\mathbb{I}\left\{p_i^t \leq \sum_{j\in V}p_j^t\right\}}{p_i^t w_i^t}\left(\frac{\frac{1}{w_i^t}}{\sum_{i\in U}\frac{1}{w_i^t}}\right)\right] = \mathcal{O}\left(\mathbb{S}_1\left(\frac{|U|\log T}{\beta(1-\beta)}\right)\right),$$

$$K\sum_{t=1}^{T}\sum_{i\in[K]}\frac{\left(\gamma_i^t - \gamma_i^{t+1}\right)^2\left(\frac{\beta\left(p_i^t\right)^{\beta-1}}{1-\beta}\right)^2}{w_i^t} = \mathcal{O}\left(\frac{\sqrt{\beta}K^2}{(1-\beta)^{3/2}}\right),$$

*which leads to the following bound of* RESREG*:*

$$\text{RESREG} = \mathcal{O}\left(\mathbb{S}_2\left(\frac{\log T}{\beta(1-\beta)}\right) + \mathbb{S}_1\left(\frac{|U|\log T}{\beta(1-\beta)}\right) + \frac{\sqrt{\beta}K^2}{(1-\beta)^{3/2}}\right).$$

*Proof.* Clearly, the first inequality relates to that of the stability term since

$$\mathbb{E}\left[\sum_{t=1}^{T}\sum_{i\in V}\frac{1}{p_i^t w_i^t}\right] \leq \mathbb{E}\left[\sum_{t=1}^{T}\sum_{i\in V}\frac{\left(p_i^t\right)^{1-\beta}}{\beta\gamma_i^t}\right] = \mathcal{O}\left(\mathbb{S}_2\left(\frac{\log T}{\beta(1-\beta)}\right)\right),$$

where the first step uses the fact that $w_i^t \geq \frac{\gamma_i^t \beta}{\left(p_i^t\right)^{2-\beta}}$, and the second step follows from the similar argument of the analysis in Lemma A.2.

Next, by direct calculation, we have

$$\sum_{t=1}^{T}\sum_{i\in U}\frac{\mathbb{I}\left\{p_i^t \leq \sum_{j\in V}p_j^t\right\}}{p_i^t w_i^t}\left(\frac{\frac{1}{w_i^t}}{\sum_{i\in U}\frac{1}{w_i^t}}\right)$$

$$\leq \sum_{t=1}^{T}\sum_{i\in U}\frac{\left(p_i^t\right)^{1-\beta}}{\beta\gamma_i^t}\left(\mathbb{I}\left\{p_i^t \leq \sum_{j\in V}p_j^t\right\}\cdot\frac{\frac{1}{w_i^t}}{\sum_{i\in U}\frac{1}{w_i^t}}\right)$$

$$\leq \frac{2}{\sqrt{\beta(1-\beta)}}\sum_{i\in U}\sum_{t=1}^{T}\frac{\left(\max\left\{p_i^{t-1},1/T\right\}\right)^{1-\beta}}{\sqrt{1+\sum_{s<t}\max\left\{p_i^s,1/T\right\}^{1-2\beta}}}\left(\mathbb{I}\left\{p_i^t \leq \sum_{j\in V}p_j^t\right\}\cdot\frac{\frac{1}{w_i^t}}{\sum_{i\in U}\frac{1}{w_i^t}}\right)$$

$$\leq \mathcal{O}\left(\sum_{i\in U}\sqrt{\frac{\log\left(1+\sum_{t=1}^{T}\max\left\{p_i^t,1/T\right\}^{1-2\beta}\right)}{\beta(1-\beta)}}\sum_{t=1}^{T}\max\left\{p_i^t,1/T\right\}\cdot\mathbb{I}\left\{p_i^t \leq \sum_{j\in V}p_j^t\right\}\cdot\frac{\frac{1}{w_i^t}}{\sum_{i\in U}\frac{1}{w_i^t}}\right)$$

$$= \mathcal{O}\left(\sum_{i\in U}\sqrt{\frac{\log T}{\beta(1-\beta)}\sum_{t=1}^{T}\left(p_i^t\mathbb{I}\left\{p_i^t \leq \sum_{j\in V}p_j^t\right\}\frac{\frac{1}{w_i^t}}{\sum_{i\in U}\frac{1}{w_i^t}}\right)}\right),$$

where the first step uses the fact that $w_i^t \geq \frac{\beta\gamma_i^t}{\left(p_i^t\right)^{2-\beta}}$; the second step follows from the multiplicative relation between $p_i^t$ and $p_i^{t-1}$ (we set $p_i^0 = 1/K = p_i^1$ for $\forall i \in [K]$ for convenience); the third step

applies Lemma C.2.1, a weighted variant of Lemma 4.2, and again the multiplicative relation; the last step follows from $\max\left\{p_i^t, 1/T\right\}^{1-2\beta} \le T$ for $\beta \in (0,1)$ and $\max\left\{p_i^t, 1/T\right\} \le p_i^t + 1/T$.

To further bound this term, we here have to use the Cauchy-Schwarz inequality which leads to the extra $\frac{|U|}{\Delta_{\text{MIN}}}$ dependency in the final regret bound:

$$\mathcal{O}\left(\sqrt{\frac{|U|\log T}{\beta(1-\beta)}\sum_{t=1}^{T}\sum_{i\in U}\left(p_i^t\mathbb{I}\left\{p_i^t \le \sum_{j\in V}p_j^t\right\} + \frac{1}{T}\right)\frac{\frac{1}{w_i^t}}{\sum_{i\in U}\frac{1}{w_i^t}}}\right)$$

$$\le \mathcal{O}\left(\sqrt{\frac{|U|\log T}{\beta(1-\beta)}\sum_{t=1}^{T}\left(\left(\sum_{j\in V}p_j^t\right) + \frac{1}{T}\right)\sum_{i\in U}\frac{\frac{1}{w_i^t}}{\sum_{i\in U}\frac{1}{w_i^t}}}\right)$$

$$= \mathcal{O}\left(\sqrt{\frac{|U|\log T}{\beta(1-\beta)}\sum_{t=1}^{T}\left(\frac{1}{T} + \sum_{j\in V}p_j^t\right)}\right),$$

where the first step uses the property of indicator that $p_i^t\mathbb{I}\left\{p_i^t \le \sum_{j\in V}p_j^t\right\} \le \sum_{j\in V}p_j^t$. Taking the expectation finishes the proof for the second inequality.

For the last inequality in the statement, for any round $t$, we have

$$K\sum_{i\in[K]}\frac{\left(\gamma_i^t - \gamma_i^{t+1}\right)^2\left(\frac{\beta\left(p_i^t\right)^{\beta-1}}{1-\beta}\right)^2}{w_i^t}$$

$$\le \frac{K\beta^2}{(1-\beta)^2}\sum_{i\in[K]}\frac{\left(\gamma_i^{t+1} - \gamma_i^t\right)^2\left(p_i^t\right)^{2-\beta}}{\beta\gamma_i^t\left(p_i^t\right)^{2-2\beta}}$$

$$= \frac{K\beta}{(1-\beta)^2}\sum_{i\in[K]}\frac{\left(\gamma_i^{t+1} - \gamma_i^t\right)^2\left(p_i^t\right)^{\beta}}{\gamma_i^t}$$

$$\le \mathcal{O}\left(\frac{K}{(1-\beta)}\sum_{i\in[K]}\frac{\left(\gamma_i^{t+1} - \gamma_i^t\right)\left(\max\left\{p_i^t, 1/T\right\}\right)^{1-2\beta}\left(p_i^t\right)^{\beta}}{\gamma_i^t\gamma_i^{t+1}}\right)$$

$$\le \mathcal{O}\left(\frac{K}{(1-\beta)}\sum_{i\in[K]}\frac{\left(\gamma_i^{t+1} - \gamma_i^t\right)\left(\max\left\{p_i^t, 1/T\right\}\right)^{1-\beta}}{\gamma_i^t\gamma_i^{t+1}}\right) \qquad \left(p_i^t \le \max\left\{p_i^t, 1/T\right\}\right)$$

$$\le \mathcal{O}\left(\frac{K}{(1-\beta)}\sum_{i\in[K]}\left(\frac{1}{\gamma_i^t} - \frac{1}{\gamma_i^{t+1}}\right)\right), \qquad \left(\max\left\{p_i^t, 1/T\right\} \le 1\right)$$

where the first step applies $w_i^t \ge \frac{\beta\gamma_i^t}{\left(p_i^t\right)^{2-\beta}}$ and Eq. (21).

Taking the summation over all the rounds, we have

$$\sum_{t=1}^{T}\frac{K}{(1-\beta)}\sum_{i\in[K]}\left(\frac{1}{\gamma_i^t} - \frac{1}{\gamma_i^{t+1}}\right) = \frac{K}{(1-\beta)}\sum_{i\in[K]}\sum_{t=1}^{T}\left(\frac{1}{\gamma_i^t} - \frac{1}{\gamma_i^{t+1}}\right) = \mathcal{O}\left(\frac{K^2\beta^{1/2}}{(1-\beta)^{3/2}}\right),$$

which finishes the proof. $\qquad\square$

**Lemma A.4.4** (RESREG for Log-barrier). *When using the log-barrier regularizer with $C_{\text{LOG}} = 162$, $\alpha = 0$, and $\theta = \sqrt{1/\log T}$, Algorithm 1 ensures:*

$$\mathbb{E}\left[\sum_{t=1}^{T}\sum_{i\in V}\frac{1}{p_i^t w_i^t}\right] = \mathcal{O}\left(\mathbb{S}_2\left(\log T\right)\right),$$

$$\mathbb{E}\left[\sum_{t=1}^{T}\sum_{i\in U}\frac{\mathbb{I}\left\{p_i^t\le\sum_{j\in V}p_j^t\right\}}{p_i^t w_i^t}\left(\frac{\frac{1}{w_i^t}}{\sum_{i\in U}\frac{1}{w_i^t}}\right)\right]=\mathcal{O}\left(\mathbb{S}_1\left(|U|\log T\right)\right),$$

$$K\sum_{t=1}^{T}\sum_{i\in[K]}\frac{\left(\gamma_i^t-\gamma_i^{t+1}\right)^2\left(\frac{1}{p_i^t}\right)^2}{w_i^t}=\mathcal{O}\left(\frac{K^2}{\sqrt{\log T}}\right),$$

*which leads to the following bound of* RESREG*:*

$$\text{RESREG}=\mathcal{O}\left(\mathbb{S}_2\left(\log T\right)+\mathbb{S}_1\left(|U|\log T\right)+\frac{K^2}{\sqrt{\log T}}\right).$$

*Proof.* By direct calculation, we have

$$\mathbb{E}\left[\sum_{t=1}^{T}\sum_{i\in V}\frac{1}{p_i^t w_i^t}\right]\le\mathbb{E}\left[\sum_{i\in V}\sum_{t=1}^{T}\frac{p_i^t}{\gamma_i^t}\right]=\mathcal{O}\left(\mathbb{E}\left[\sqrt{\log T}\sum_{i\in V}\sum_{t=1}^{T}\frac{p_i^t}{\gamma_i^{t+1}}\right]\right)=\mathcal{O}\left(\mathbb{S}_2\left(\log T\right)\right),$$

where the second step use the same arguments used in [Lemma A.2.2](#).

Following the same idea used for the $\beta$-Tsallis entropy regularizer, we have

$$\mathbb{E}\left[\sum_{t=1}^{T}\sum_{i\in U}\frac{\mathbb{I}\left\{p_i^t\le\sum_{j\in V}p_j^t\right\}}{p_i^t w_i^t}\left(\frac{\frac{1}{w_i^t}}{\sum_{i\in U}\frac{1}{w_i^t}}\right)\right]$$

$$=\mathcal{O}\left(\mathbb{E}\left[\sum_{i\in U}\sqrt{\log T\cdot\sum_{t=1}^{T}p_i^t\mathbb{I}\left\{p_i^t\le\sum_{j\in V}p_j^t\right\}\left(\frac{\frac{1}{w_i^t}}{\sum_{i\in U}\frac{1}{w_i^t}}\right)}\right]\right)$$

$$=\mathcal{O}\left(\mathbb{E}\left[\sqrt{|U|\log T\cdot\sum_{t=1}^{T}\sum_{i\in U}\left(\sum_{i\in V}p_i^t\right)\left(\frac{\frac{1}{w_i^t}}{\sum_{i\in U}\frac{1}{w_i^t}}\right)}\right]\right)$$

$$=\mathcal{O}\left(\mathbb{S}_1\left(|U|\log T\right)\right),$$

where the second step follows from [Lemma 4.2](#) and $w_i^t\ge\gamma_i^t/(p_i^t)^2$; the second step uses the Cauchy-Schwarz inequality; the last follows from the definition of the self-bounding quantity $\mathbb{S}_1$ in [Eq. (14)](#).

Finally, by direct calculation, for any round $t$, we have

$$K\sum_{i\in[K]}\frac{\left(\gamma_i^t-\gamma_i^{t+1}\right)^2\left(\frac{1}{p_i^t}\right)^2}{w_i^t}\le K\sum_{i\in[K]}\frac{\left(\gamma_i^{t+1}-\gamma_i^t\right)^2\left(p_i^t\right)^2}{\gamma_i^t\left(p_i^t\right)^2}\qquad (w_i^t\ge\gamma_i^t/(p_i^t)^2)$$

$$\le\frac{K}{\log T}\sum_{i\in[K]}\frac{\left(\gamma_i^{t+1}-\gamma_i^t\right)\max\left\{p_i^t,1/T\right\}}{\gamma_i^t\gamma_i^{t+1}}\qquad (\text{by Eq. (21)})$$

$$\le\frac{K}{\log T}\sum_{i\in[K]}\left(\frac{1}{\gamma_i^t}-\frac{1}{\gamma_i^{t+1}}\right).\qquad (\max\left\{p_i^t,1/T\right\}\le 1)$$

Taking the summation over all rounds yields that

$$\frac{K}{\log T}\sum_{i\in[K]}\sum_{t=1}^{T}\left(\frac{1}{\gamma_i^t}-\frac{1}{\gamma_i^{t+1}}\right)=\mathcal{O}\left(\frac{K}{\log T}\sum_{i\in[K]}\frac{1}{\gamma_i^1}\right)=\mathcal{O}\left(\frac{K^2}{\sqrt{\log T}}\right),$$

which finishes the proof. $\qquad\square$

**Lemma A.4.5** (RESREG for Shannon Entropy). *When using the Shannon entropy regularizer with* $C_{\text{LOG}}=162\log K$, $\alpha=1$, *and* $\theta=\sqrt{1/\log T}$, [Algorithm 1](#) *ensures:*

$$\mathbb{E}\left[\sum_{t=1}^{T}\sum_{i\in V}\frac{1}{p_i^t w_i^t}\right]=\mathcal{O}\left(\mathbb{S}_2\left(\log^2 T\right)\right),$$

$$\mathbb{E}\left[\sum_{t=1}^{T}\sum_{i\in U}\frac{\mathbb{I}\left\{p_i^t \le \sum_{j\in V}p_j^t\right\}}{p_i^t w_i^t}\left(\frac{\frac{1}{w_i^t}}{\sum_{i\in U}\frac{1}{w_i^t}}\right)\right] = \mathcal{O}\left(\mathbb{S}_1\left(|U|\log^2 T\right)\right),$$

$$K\sum_{t=1}^{T}\sum_{i\in[K]}\frac{\left(\gamma_i^t - \gamma_i^{t+1}\right)^2\left(\log\left(p_i^t\right)\right)^2}{w_i^t} = \mathcal{O}\left(K^2\log^{3/2}T\right),$$

*which leads to the following bound of* RESREG*:*

$$\text{RESREG} = \mathcal{O}\left(\mathbb{S}_2\left(\log^2 T\right) + \mathbb{S}_1\left(|U|\log^2 T\right) + |K|^2\log^{3/2}T\right).$$

*Proof.* Similar to the analysis for $\beta$-Tsallis entropy, we also have the first inequality relates to the bound of the stability term as:

$$\mathbb{E}\left[\sum_{t=1}^{T}\sum_{i\in V}\frac{1}{p_i^t w_i^t}\right] \le \mathbb{E}\left[\sum_{i\in V}\sum_{t=1}^{T}\frac{1}{\gamma_i^t}\right] = \mathcal{O}\left(\mathbb{E}\left[\sum_{i\in V}\sqrt{\log^2 T \cdot \sum_{t=1}^{T}p_i^t}\right]\right) = \mathcal{O}\left(\mathbb{S}_2\left(\log^2 T\right)\right),$$

where the second step follows form similar analysis of the stability term in Lemma A.2.4.

By the same arguments in the proof of Lemma A.4.3, we have

$$\mathbb{E}\left[\sum_{t=1}^{T}\sum_{i\in U}\frac{\mathbb{I}\left\{p_i^t \le \sum_{j\in V}p_j^t\right\}}{p_i^t w_i^t}\left(\frac{\frac{1}{w_i^t}}{\sum_{i\in U}\frac{1}{w_i^t}}\right)\right] = \mathcal{O}\left(\mathbb{E}\left[\sqrt{|U|\left(\log T\right)^2\cdot\sum_{t=1}^{T}\sum_{i\in V}p_i^t}\right]\right)$$
$$= \mathcal{O}\left(\mathbb{S}_1\left(|U|\log^2 T\right)\right),$$

according to our learning rate schedule, where the extra $\sqrt{\log T}$ factor is casued by $\theta$.

By direct calculation, for any round $t$, we have

$$K\sum_{i\in[K]}\frac{\left(\gamma_i^t - \gamma_i^{t+1}\right)^2\left(\log\left(p_i^t\right)\right)^2}{w_i^t}$$

$$\le K\cdot\sum_{i\in[K]}\frac{\left(\gamma_i^{t+1} - \gamma_i^t\right)^2\log^2\left(p_i^t\right)p_i^t}{\gamma_i^t} \qquad (w_i^t \ge \gamma_i^t/p_i^t)$$

$$\le \frac{K}{\log T}\cdot\sum_{i\in[K]}\frac{\left(\gamma_i^{t+1} - \gamma_i^t\right)}{\gamma_i^t\gamma_i^{t+1}}\cdot\frac{\log^2\left(p_i^t\right)p_i^t}{\max\left\{p_i^t, 1/T\right\}} \qquad \text{(by Eq. (21))}$$

$$= \frac{K}{\log T}\sum_{i\in[K]}\left(\frac{1}{\gamma_i^t} - \frac{1}{\gamma_i^{t+1}}\right)\frac{\log^2\left(p_i^t\right)p_i^t}{\max\left\{p_i^t, 1/T\right\}}.$$

Note that, for any arm $i$, the term $\frac{\log^2(p_i^t)p_i^t}{\max\{p_i^t, 1/T\}}$ can be bounded as :

$$\frac{\log^2\left(p_i^t\right)p_i^t}{\max\left\{p_i^t, 1/T\right\}}$$

$$= \frac{\left(\log\left(z_i^t\right) + \log\left(\frac{p_i^t}{z_i^t}\right)\right)^2 p_i^t}{z_i^t} \qquad \text{(let } z_i^t = \max\left\{p_i^t, 1/T\right\} \text{ for simplicity)}$$

$$\le 2\left(\left(\log T\right)^2 + \frac{p_i^t}{z_i^t}\cdot\log^2\left(\frac{p_i^t}{z_i^t}\right)\right) \qquad ((x+y)^2 \le 2\left(x^2 + y^2\right) \text{ and } z_i^t \in [1/T, 1])$$

$$\le 2\left(\left(\log T\right)^2 + 1\right). \qquad \text{(since } x\left(\log x\right)^2 \le 1 \text{ for } x \in [0, 1])$$

Finally, taking the summation over all rounds yields

$$\frac{K}{\log T}\sum_{t=1}^{T}\sum_{i\in[K]}\left(\frac{1}{\gamma_i^t} - \frac{1}{\gamma_i^{t+1}}\right)\frac{\log^2\left(p_i^t\right)p_i^t}{\max\left\{p_i^t, 1/T\right\}} \le \frac{2K\left(\left(\log T\right)^2 + 1\right)}{\log T}\sum_{t=1}^{T}\sum_{i\in[K]}\left(\frac{1}{\gamma_i^t} - \frac{1}{\gamma_i^{t+1}}\right)$$

$$= \mathcal{O}\left(K \log T \sum_{i \in [K]} \frac{1}{\gamma_i^1}\right) = \mathcal{O}\left(K^2 \log^{3/2} T\right),$$

which finishes the proof. $\qquad\square$

# B  Proof of Theorem 3.4: Decoupled-Tsallis-INF

For the Decoupled-Tsallis-INF algorithm [Rouyer and Seldin, 2020], the learning rate schedule is $\gamma_i^t = K^{1/6}\sqrt{t}$ for all $i \in [K]$ at round $t$ (henceforth denoted by $\gamma^t$ for conciseness), and the loss estimator is

$$\widehat{\ell}_i^t = \frac{\mathbb{I}\{j^t = i\}\ell_i^t}{g_i^t}, \quad \forall i \in [K], \quad \text{where} \quad g_i^t = \frac{(p_i^t)^{2/3}}{\sum_{j \in [K]} (p_j^t)^{2/3}} \text{ and } j^t \sim g^t. \tag{35}$$

Although our analysis of Decoupled-Tsallis-INF follows similar ideas as the case for MAB, we need a finer decomposition of $D^{t,t+1}(p^t, p^{t+1})$ since we do not add an extra log-barrier regularizer to ensure the multiplicative relation between $\bar{p}^{t+1}$ and $p^t$ (which is important to avoid any $T$-dependence). To present this new decomposition, we first introduce some definitions. For $\bar{p}^{t+1}$, we maintain the same definition of $\bar{p}_U^{t+1}$ in Eq. (7), but slightly adjust the definition of $\bar{p}_V^{t+1}$ as:

$$\bar{p}_V^{t+1} = \operatorname*{argmin}_{\substack{x \in \mathbb{R}_{\geq 0}^K, \ \sum_{i \in U} x_i = 0, \\ \sum_{i \in V} x_i = \sum_{i \in V} p_i^t}} \left\langle x, \sum_{\tau \leq t} \widehat{\ell}^\tau \right\rangle + \phi_V^t(x). \tag{36}$$

Compared to the definition in Eq. (7) used for the analysis of MAB, the new definition here changes $\phi^{t+1}$ to $\phi^t$. Such a skewness of the regularizer eventually allows us to get $\sum_{i \in V}(p^t)^{2/3}$ rather than $\sum_{i \in V}(\bar{p}^{t+1})^{2/3}$ in the regret bound, which is important since we cannot easily convert between these two anymore as we did for the MAB analysis due to the aforementioned lack of multiplicative relation. Furthermore, as $\phi^{t+1}$ is used for $U$ and $\phi^t$ is used for $V$, we introduce another intermediate point $\widetilde{p}^{t+1}$ to bridge this skewness, which is defined as:

$$\widetilde{p}^{t+1} = \operatorname*{argmin}_{x \in \Omega_K} \left\langle x, \sum_{\tau \leq t} \widehat{\ell}^\tau \right\rangle + \widetilde{\phi}^t(x), \tag{37}$$

where $\widetilde{\phi}^t(x) = \phi_V^t(x) + \phi_U^{t+1}(x)$ can be regarded as a specific intermediate regularizer between $\phi_V^t$ and $\phi_U^{t+1}$. To distinguish the Bregman divergence $D^t(x, y)$ defined on regularizer $\phi^t$, we define $D^{\widetilde{t}}(x, y)$ on regularizer $\widetilde{\phi}^t$ as:

$$D^{\widetilde{t}}(x, y) = \widetilde{\phi}^t(x) - \widetilde{\phi}^t(y) - \left\langle \nabla\widetilde{\phi}^t(y), x - y \right\rangle. \tag{38}$$

Similarly, we define the following two notions of skewed Bregman divergence for $s, t \in \mathbb{N}$:

$$D^{t,\widetilde{s}}(x, y) = \phi^t(x) - \widetilde{\phi}^s(y) - \left\langle \nabla\widetilde{\phi}^s(y), x - y \right\rangle,$$

$$D^{\widetilde{t},s}(x, y) = \widetilde{\phi}^t(y) - \phi^s(x) - \left\langle \nabla\phi^s(y), x - y \right\rangle.$$

With the help of definitions above, our new decomposition for $D^{t,t+1}(p^t, p^{t+1}) - D_U^{t,t+1}(q^t, q^{t+1})$ is as follows (see Lemma B.1.1)

$$\underbrace{D_V^t(p^t, \bar{p}^{t+1}) + \phi_V^t(p^{t+1}) - \phi_V^{t+1}(p^{t+1})}_{\text{regret on sub-optimal arms}}$$
$$+ \underbrace{D_U^{t,t+1}(p^t, \bar{p}^{t+1}) - D_U^{t,t+1}(q^t, q^{t+1})}_{\text{regret on optimal arms}} + \underbrace{D^{\widetilde{t}}(\bar{p}^{t+1}, \widetilde{p}^{t+1})}_{\text{residual regret}}. \tag{39}$$

Now, we are ready to use this decomposition to analyze the regret bound for Decoupled-Tsallis-INF. As we do not add an extra log-barrier, we just set $\epsilon = 0$, i.e., $q^t = \widetilde{q}^t$ where $\widetilde{q}^t$ is given in Eq. (15). As a result, the only preprocessing cost for Decoupled-Tsallis-INF is $D$. To show Theorem 3.4, we present the following theorem.

**Theorem B.1.** *Decoupled-Tsallis-INF ensures*

$$\text{Reg}^T \leq \mathcal{O}\left(\sqrt{KT_0} + \mathbb{E}\left[\sum_{t=T_0+1}^{T}\left(\frac{K^{1/6}}{\sqrt{t}}\sum_{i\in V}\left(p_i^t\right)^{2/3}\right)\right] + \sqrt{K}\right) + D,$$

*for any $T_0 \geq 16K$, any subset $U \subseteq [K]$, $V = [K]\backslash U$, and $D = \mathbb{E}\left[\sum_{t=1}^{T}\max_{i\in U}\mathbb{E}^t\left[\ell_i^t - \ell_{i^\star}^t\right]\right]$.*

*Proof.* With the decomposition in Lemma B.1.1, we have the following upper bound for $\text{Reg}^T$:

$$\mathbb{E}\left[D^1(q^1,p^1) + \sum_{t=1}^{T}D^{t,t+1}(p^t,p^{t+1}) - D_U^{t,t+1}(q^t,q^{t+1})\right] + D$$

$$\leq \mathbb{E}\left[\sum_{t=T_0+1}^{T}D_V^t(p^t,\bar{p}^{t+1}) + \phi_V^t(p^{t+1}) - \phi_V^{t+1}(p^{t+1})\right] + \mathbb{E}\left[\sum_{t=T_0+1}^{T}D_U^{t,t+1}(p^t,\bar{p}^{t+1}) - D_U^{t,t+1}(q^t,q^{t+1})\right]$$

$$+ \mathbb{E}\left[\sum_{t=T_0+1}^{T}D^{\widetilde{t}}(\bar{p}^{t+1},\widetilde{p}^{t+1})\right] + \mathcal{O}\left(\sqrt{KT_0} + \sqrt{K}\right) + D,$$

where we bound the regret of the first $T_0$ rounds by $\mathcal{O}\left(\sqrt{KT_0}\right)$ via Lemma B.5.3, and $D^1(q^1,p^1)$ by

$$D^1(q^1,p^1) = -3\gamma^1\sum_{i\in[K]}\left(\left(q_i^1\right)^{2/3} - \left(p_i^1\right)^{2/3}\right) \leq 3K^{1/6}\sum_{i\in[K]}\left(p_i^1\right)^{2/3} = 3\sqrt{K}.$$

First, we bound the regret on sub-optimal arms using Lemma B.2.1:

$$\mathbb{E}\left[\sum_{t=T_0+1}^{T}D_V^t(p^t,\bar{p}^{t+1}) + \phi_V^t(p^{t+1}) - \phi_V^{t+1}(p^{t+1})\right] \leq \mathcal{O}\left(\mathbb{E}\left[\sum_{t=T_0+1}^{T}\sum_{i\in V}\frac{K^{1/6}\left(p_i^t\right)^{2/3}}{\sqrt{t}}\right]\right). \tag{40}$$

The regret related to the optimal arms for any given $t$ is again non-positive (see Appendix B.3):

$$D_U^{t,t+1}(p^t,\bar{p}^{t+1}) - D_U^{t,t+1}(q^t,q^{t+1}) \leq 0.$$

Finally, we turn to bound the residual regret (see Appendix B.4). By Lemma B.4.1, the residual regret is bounded as

$$\mathbb{E}\left[\sum_{t=T_0+1}^{T}D^{\widetilde{t}}\left(\bar{p}^{t+1},\widetilde{p}^{t+1}\right)\right] \leq \mathcal{O}\left(\mathbb{E}\left[\sum_{t=T_0+1}^{T}\sum_{i\in V}\frac{K^{1/6}\left(p_i^t\right)^{2/3}}{\sqrt{t}}\right]\right). \tag{41}$$

Combining all the above, we complete the proof. $\square$

Armed with the result in Theorem B.1, we are now ready to prove Theorem 3.4.

*Proof of Theorem 3.4.* Theorem B.1 shows that there exists a constant $z \in \mathbb{R}_{>0}$ such that

$$\text{Reg}^T \leq z\left(\sqrt{KT_0} + \mathbb{E}\left[\sum_{t=T_0+1}^{T}\frac{K^{1/6}}{\sqrt{t}}\sum_{i\in V}\left(p_i^t\right)^{2/3}\right] + \sqrt{K}\right) + D$$

holds for any $T_0 \geq 16K$. For the adversarial setting, we simply pick $T_0 = T$ and $V = [K]\backslash\{i^\star\}$ (so that $D = 0$), and then the claimed bound $\mathcal{O}(\sqrt{KT})$ follows.

For the stochastic setting, let $V = \{i : \Delta_i \neq 0\}$ as in Section 2 and define

$$\widetilde{\mathbb{G}} = \sum_{i\in V}\frac{1}{\Delta_i^2}, \tag{42}$$

as an instance complexity measure for DEE-MAB. We have the following bound of $\mathrm{Reg}^T$ for any $\eta > 0$ and $T_0 \geq 16K$:

$$\mathrm{Reg}^T = (1+\eta)\,\mathrm{Reg}^T - \eta\mathrm{Reg}^T$$

$$\leq (1+\eta)\left(z\sqrt{KT_0} + \mathbb{E}\left[\sum_{t=T_0+1}^{T}\frac{zK^{1/6}}{\sqrt{t}}\sum_{i\in V}\left(p_i^t\right)^{2/3}\right] + D + z\sqrt{K}\right)$$

$$\quad - \eta\left(\mathbb{E}\left[\sum_{t=T_0+1}^{T}\sum_{i\in V}\Delta_i p_i^t - C\right]\right)$$

$$= \mathbb{E}\left[\sum_{t=T_0+1}^{T}\sum_{i\in V}\left(\frac{(1+\eta)\,zK^{1/6}}{\sqrt{t}}\left(p_i^t\right)^{2/3} - \eta\Delta_i p_i^t\right)\right]$$

$$\quad + (1+\eta)\left(z\sqrt{KT_0} + D + z\sqrt{K}\right) + \eta C$$

$$\leq \sum_{t=T_0+1}^{T}\sum_{i\in V}\frac{(1+\eta)^3\,z^3\sqrt{K}}{\eta^2\Delta_i^2 t^{3/2}} + (1+\eta)\left(z\sqrt{KT_0} + D + z\sqrt{K}\right) + \eta C$$

$$\leq \frac{2z^3(1+\eta)^3\,\widetilde{\mathbb{G}}}{\eta^2}\sqrt{\frac{K}{T_0}} + (1+\eta)\left(z\sqrt{KT_0} + D + z\sqrt{K}\right) + \eta C$$

$$= \mathcal{O}\left(\frac{(1+\eta^3)\,\widetilde{\mathbb{G}}}{\eta^2}\sqrt{\frac{K}{T_0}} + \eta\left(C + \sqrt{KT_0}\right) + \sqrt{KT_0} + (1+\eta)\left(D + \sqrt{K}\right)\right)$$

$$= \mathcal{O}\left(\frac{\widetilde{\mathbb{G}}}{\eta^2}\sqrt{\frac{K}{T_0}} + \eta\left(\widetilde{\mathbb{G}}\sqrt{\frac{K}{T_0}} + C + \sqrt{KT_0} + D\right) + \left(\sqrt{KT_0} + D\right)\right),$$

where the second step uses Condition (1); the forth step uses [Rouyer and Seldin, 2020, Lemma 8]; the fifth step follows from the fact $\sum_{t=T_0+1}^{T} t^{-3/2} \leq \int_{T_0+1}^{T} t^{-3/2}dt \leq \frac{2}{\sqrt{T_0+1}} \leq \frac{2}{\sqrt{T_0}}$; the sixth step uses $(1+\eta)^3 \leq 4(1+\eta^3)$ for any $\eta > 0$; the last step uses $\sqrt{K} \leq \sqrt{KT_0}$ as $T_0 \geq 16K$.

Picking the optimal $\eta$ yields

$$\mathrm{Reg}^T \leq \mathcal{O}\left(\left(\widetilde{\mathbb{G}}\sqrt{\frac{K}{T_0}}\right)^{1/3}\left(\widetilde{\mathbb{G}}\sqrt{\frac{K}{T_0}} + C + \sqrt{KT_0} + D\right)^{2/3} + \sqrt{KT_0} + D\right)$$

$$\leq \mathcal{O}\left(\left(\widetilde{\mathbb{G}}\sqrt{\frac{K}{T_0}}\right)^{1/3}\left(\widetilde{\mathbb{G}}\sqrt{\frac{K}{T_0}}\right)^{2/3} + \left(\widetilde{\mathbb{G}}\sqrt{\frac{K}{T_0}}\right)^{1/3}C^{2/3} + \left(\widetilde{\mathbb{G}}\sqrt{\frac{K}{T_0}}\right)^{1/3}\left(\sqrt{KT_0}\right)^{2/3}\right)$$

$$\quad + \mathcal{O}\left(\left(\widetilde{\mathbb{G}}\sqrt{\frac{K}{T_0}}\right)^{1/3}D^{2/3} + \sqrt{KT_0} + D\right)$$

$$\leq \mathcal{O}\left(\frac{\widetilde{\mathbb{G}}K^{1/2}}{T_0^{1/2}} + \frac{\widetilde{\mathbb{G}}^{1/3}K^{1/6}C^{2/3}}{T_0^{1/6}} + \widetilde{\mathbb{G}}^{1/3}K^{1/2}T_0^{1/6} + K^{1/2}T_0^{1/2} + D\right), \tag{43}$$

where the second step uses the fact $(x+y)^{2/3} \leq 3\left(x^{2/3} + y^{2/3}\right)$ for any $x, y \geq 0$, and the third step follows from the fact $x^{1/3}y^{2/3} \leq x + y$ for any $x, y \geq 0$.

For $C \leq \widetilde{\mathbb{G}}^{1/2}K^{1/2}$, by picking $T_0 = \max\left\{\widetilde{\mathbb{G}}, 16K\right\}$, we have

$$\mathrm{Reg}^T \leq \mathcal{O}\left(\frac{\widetilde{\mathbb{G}}K^{1/2}}{T_0^{1/2}} + \frac{\widetilde{\mathbb{G}}^{2/3}K^{1/2}}{T_0^{1/6}} + \widetilde{\mathbb{G}}^{1/3}K^{1/2}T_0^{1/6} + K^{1/2}T_0^{1/2} + D\right)$$

$$\leq \mathcal{O}\left(\frac{\widetilde{\mathbb{G}}K^{1/2}}{\widetilde{\mathbb{G}}^{1/2}} + \frac{\widetilde{\mathbb{G}}^{2/3}K^{1/2}}{\widetilde{\mathbb{G}}^{1/6}} + K^{1/2}\left(\widetilde{\mathbb{G}} + K\right)^{1/2} + D\right)$$

$$\leq \mathcal{O}\left(\widetilde{\mathbb{G}}^{1/2}K^{1/2} + K + D\right),$$

where the first step uses $C \leq \widetilde{\mathbb{G}}^{1/2}K^{1/2}$; the second step follows from $\widetilde{\mathbb{G}} \leq T_0 \leq \widetilde{\mathbb{G}} + 16K$; the last step follows from the fact $\sqrt{x+y} \leq \sqrt{x} + \sqrt{y}$ for $x, y \geq 0$.

For $C \geq \widetilde{\mathbb{G}}^{1/2}K^{1/2}$, we set $T_0 = \max\left\{\frac{\widetilde{\mathbb{G}}^{1/2}C}{K^{1/2}}, 16K\right\}$ and have Eq. (43) further bounded by

$$\mathcal{O}\left(\frac{\mathbb{G}K^{1/2}}{T_0^{1/2}} + \frac{\widetilde{\mathbb{G}}^{1/3}K^{1/6}C^{2/3}}{T_0^{1/6}} + \widetilde{\mathbb{G}}^{1/3}K^{1/2}T_0^{1/6} + K^{1/2}T_0^{1/2} + D\right)$$

$$\leq \mathcal{O}\left(\widetilde{\mathbb{G}}K^{1/2}\left(\frac{K^{1/2}}{\widetilde{\mathbb{G}}^{1/2}C}\right)^{1/2} + \widetilde{\mathbb{G}}^{1/3}K^{1/6}C^{2/3}\left(\frac{K^{1/2}}{\widetilde{\mathbb{G}}^{1/2}C}\right)^{1/6} + \widetilde{\mathbb{G}}^{1/3}K^{1/2}\left(\frac{\widetilde{\mathbb{G}}^{1/2}C}{K^{1/2}}\right)^{1/6}\right)$$

$$+ \mathcal{O}\left(\widetilde{\mathbb{G}}^{1/3}K^{1/2} \cdot K^{1/6} + K^{1/2}\left(\frac{\widetilde{\mathbb{G}}^{1/2}C}{K^{1/2}}\right)^{1/2} + K^{1/2} \cdot K^{1/2} + D\right)$$

$$= \mathcal{O}\left(\widetilde{\mathbb{G}}^{3/4}K^{3/4}C^{-1/2} + \widetilde{\mathbb{G}}^{1/4}K^{1/4}C^{1/2} + \widetilde{\mathbb{G}}^{5/12}K^{5/12}C^{1/6} + \widetilde{\mathbb{G}}^{1/4}K^{1/4}C^{1/2}\right)$$

$$+ \mathcal{O}\left(\widetilde{\mathbb{G}}^{1/2}K^{1/2} + K + D\right)$$

$$\leq \mathcal{O}\left(\widetilde{\mathbb{G}}^{1/4}K^{1/4}C^{1/2} + \widetilde{\mathbb{G}}^{1/2}K^{1/2} + K + D\right),$$

where the first step uses $\frac{\widetilde{\mathbb{G}}^{1/2}C}{K^{1/2}} \leq T_0 \leq \frac{\widetilde{\mathbb{G}}^{1/2}C}{K^{1/2}} + 16K$; the second step follows from the facts that $x^{1/3}y^{2/3} \leq x + y$ for any $x, y \geq 0$, and that $\widetilde{\mathbb{G}}^{1/3}K^{2/3} = \left(\widetilde{\mathbb{G}}^{1/2}K^{1/2}\right)^{2/3} \cdot (K)^{1/3}$; the third step uses $C \geq \widetilde{\mathbb{G}}^{1/2}K^{1/2}$.

Combining these two cases together yields the claimed bound:

$$\mathrm{Reg}^T = \mathcal{O}\left(\widetilde{\mathbb{G}}^{1/4}K^{1/4}C^{1/2} + \widetilde{\mathbb{G}}^{1/2}K^{1/2} + K + D\right)$$

$$= \mathcal{O}\left(\sqrt{\sum_{i \in V}\frac{K}{\Delta_i^2}} + \sqrt{C} \cdot \left(\sum_{i \in V}\frac{K}{\Delta_i^2}\right)^{1/4} + K + D\right).$$

$\square$

## B.1 Regret Decomposition for Decouple-Tsallis-INF

In this subsection, we present the proposed regret decomposition using $\bar{p}^{t+1}$ (with the new definition) and $\widetilde{p}^t$.

**Lemma B.1.1.** *For any t, $D^{t,t+1}(p^t, p^{t+1}) - D_U^{t,t+1}(q^t, q^{t+1})$ is bounded by*

$$\underbrace{D_V^t(p^t, \bar{p}^{t+1}) + \phi_V^t(p^{t+1}) - \phi_V^{t+1}(p^{t+1})}_{\text{regret on sub-optimal arms}} + \underbrace{D_U^{t,t+1}(p^t, \bar{p}^{t+1}) - D_U^{t,t+1}(q^t, q^{t+1})}_{\text{regret on optimal arms}} + \underbrace{D^{\widetilde{t}}(\bar{p}^{t+1}, \widetilde{p}^{t+1})}_{\text{residual regret}}.$$

*Proof.* We proceed as follows:

$$D^{t,t+1}(p^t, p^{t+1})$$

$$= \left\langle p^t - p^{t+1}, \widehat{\ell}^t \right\rangle - D^{t+1,t}(p^{t+1}, p^t)$$

$$= \left\langle p^t - p^{t+1}, \widehat{\ell}^t \right\rangle - \left(\widetilde{\phi}^t(p^{t+1}) - \phi^t(p^t) - \left\langle \nabla\phi^t(p^t), p^{t+1} - p^t \right\rangle\right) + \widetilde{\phi}^t(p^{t+1}) - \phi^{t+1}(p^{t+1})$$

$$= \left\langle p^t - p^{t+1}, \widehat{\ell}^t \right\rangle - D^{\widetilde{t},t}(p^{t+1}, p^t) + \widetilde{\phi}^t(p^{t+1}) - \phi^{t+1}(p^{t+1})$$

$$\leq \left\langle p^t - \widetilde{p}^{t+1}, \widehat{\ell}^t \right\rangle - D^{\widetilde{t},t}(\widetilde{p}^{t+1}, p^t) + \widetilde{\phi}^t(p^{t+1}) - \phi^{t+1}(p^{t+1})$$

$$
\begin{aligned}
&= D^{t,\widetilde{t}}(p^t, \widetilde{p}^{t+1}) + \widetilde{\phi}^t(p^{t+1}) - \phi^{t+1}(p^{t+1}) \\
&= D^{t,\widetilde{t}}(p^t, \widetilde{p}^{t+1}) + \phi_V^t(p^{t+1}) - \phi_V^{t+1}(p^{t+1}),
\end{aligned}
\tag{44}
$$

where the first step follows from Lemma C.5.1 according to the definitions of $p^t$; the second step adds and subtracts $\widetilde{\phi}^t(p^{t+1})$; the third step follows from the definition of $D^{\widetilde{t},t}$; the forth step uses Lemma C.5.2; the last step follows from the definition of $\widetilde{\phi}^t$ which implies $\widetilde{\phi}_U^t(p^{t+1}) = \phi_U^{t+1}(p^{t+1})$.

Then, with the help of $\bar{p}^{t+1}$, we can further decompose the term $D^{t,\widetilde{t}}(p^t, \widetilde{p}^{t+1})$ as:

$$
\begin{aligned}
&D^{t,\widetilde{t}}(p^t, \widetilde{p}^{t+1}) \\
&= \phi^t(p^t) - \widetilde{\phi}^t(\widetilde{p}^{t+1}) - \left\langle \nabla\widetilde{\phi}^t(\widetilde{p}^{t+1}), p^t - \widetilde{p}^{t+1} \right\rangle \\
&= \phi^t(p^t) - \widetilde{\phi}^t(\bar{p}^{t+1}) - \left\langle \nabla\widetilde{\phi}^t(\widetilde{p}^{t+1}), p^t - \bar{p}^{t+1} \right\rangle \\
&\quad + \widetilde{\phi}^t(\bar{p}^{t+1}) - \widetilde{\phi}^t(\widetilde{p}^{t+1}) - \left\langle \nabla\widetilde{\phi}^t(\widetilde{p}^{t+1}), \bar{p}^{t+1} - \widetilde{p}^{t+1} \right\rangle \\
&= \phi_U^t(p^t) - \widetilde{\phi}_U^t(\bar{p}^{t+1}) - \left\langle \nabla\widetilde{\phi}_U^t(\widetilde{p}^{t+1}), p_U^t - \bar{p}_U^{t+1} \right\rangle & (45) \\
&\quad + \phi_V^t(p^t) - \widetilde{\phi}_V^t(\bar{p}^{t+1}) - \left\langle \nabla\widetilde{\phi}_V^t(\widetilde{p}^{t+1}), p_V^t - \bar{p}_V^{t+1} \right\rangle & (46) \\
&\quad + D^{\widetilde{t}}(\bar{p}^{t+1}, \widetilde{p}^{t+1}),
\end{aligned}
$$

where the last term is the residual regret.

For the regret on optimal arms, we show that the term $\phi_U^t(p^t) - \widetilde{\phi}_U^t(\bar{p}^{t+1}) - \left\langle \nabla\widetilde{\phi}_U^t(\widetilde{p}^{t+1}), p_U^t - \bar{p}_U^{t+1} \right\rangle$ in Eq. (45) is exactly $D_U^{t,t+1}(p^t, \bar{p}^{t+1})$:

$$
\begin{aligned}
&\phi_U^t(p^t) - \widetilde{\phi}_U^t(\bar{p}^{t+1}) - \left\langle \nabla\widetilde{\phi}_U^t(\widetilde{p}^{t+1}), p_U^t - \bar{p}_U^{t+1} \right\rangle \\
&= \phi_U^t(p^t) - \widetilde{\phi}_U^t(\bar{p}^{t+1}) - \left\langle \nabla\phi_U^{t+1}(\bar{p}^{t+1}) + c \cdot \mathbf{1}_U, p_U^t - \bar{p}_U^{t+1} \right\rangle \\
&= \phi_U^t(p^t) - \widetilde{\phi}_U^t(\bar{p}^{t+1}) - \left\langle \nabla\phi_U^{t+1}(\bar{p}^{t+1}), p_U^t - \bar{p}_U^{t+1} \right\rangle \\
&= D_U^{t,t+1}(p^t, \bar{p}^{t+1}),
\end{aligned}
\tag{47}
$$

where the first step uses the KKT conditions of $\bar{p}^{t+1}$ and $\widetilde{p}^{t+1}$, which indicate $\nabla\widetilde{\phi}_U^t(\widetilde{p}^{t+1}) = \nabla\phi_U^{t+1}(\bar{p}^{t+1}) + c \cdot \mathbf{1}_U$ for a constant $c \in \mathbb{R}$; the second step follows from the fact $\sum_{i \in U} p_i^t = \sum_{i \in U} \bar{p}_i^{t+1}$, which guarantees $\left\langle c \cdot \mathbf{1}_U, p_U^t - \bar{p}_U^{t+1} \right\rangle = 0$ for any $c \in \mathbb{R}$; the third step follows from the definition of $\widetilde{\phi}^t$ which implies $\widetilde{\phi}_U^t(\bar{p}^{t+1}) = \phi_U^{t+1}(\bar{p}^{t+1})$.

Following the same idea of handling Eq. (45), we have:

$$
\begin{aligned}
&\phi_V^t(p^t) - \widetilde{\phi}_V^t(\bar{p}^{t+1}) - \left\langle \nabla\widetilde{\phi}_V^t(\widetilde{p}^{t+1}), p_V^t - \bar{p}_V^{t+1} \right\rangle \\
&= \phi_V^t(p^t) - \phi_V^t(\bar{p}^{t+1}) - \left\langle \nabla\phi_V^t(\bar{p}^{t+1}), p_V^t - \bar{p}_V^{t+1} \right\rangle \\
&= D_V^t(p^t, \bar{p}^{t+1}),
\end{aligned}
\tag{48}
$$

where the first step follows from the definition of $\widetilde{\phi}^t$ that $\widetilde{\phi}_V^t(\bar{p}^{t+1}) = \phi_V^t(\bar{p}^{t+1})$ and the KKT conditions of $\bar{p}^{t+1}$ and $\widetilde{p}^{t+1}$, which indicate that $\nabla\widetilde{\phi}_V^t(\widetilde{p}^{t+1}) = \nabla\phi_V^t(\bar{p}^{t+1}) + c' \cdot \mathbf{1}_V$ for a constant $c' \in \mathbb{R}$. Finally, combining these bounds together, we have

$$
\begin{aligned}
&D^{t,t+1}(p^t, p^{t+1}) \\
&\leq D^{t,\widetilde{t}}(p^t, \widetilde{p}^{t+1}) + \phi_V^t(p^{t+1}) - \phi_V^{t+1}(p^{t+1}) && \text{(By Eq. (44))} \\
&= \phi_V^t(p^{t+1}) - \phi_V^{t+1}(p^{t+1}) + \phi_V^t(p^t) - \widetilde{\phi}_V^t(\bar{p}^{t+1}) - \left\langle \nabla\widetilde{\phi}_V^t(\widetilde{p}^{t+1}), p_V^t - \bar{p}_V^{t+1} \right\rangle \\
&\quad + \phi_U^t(p^t) - \widetilde{\phi}_U^t(\bar{p}^{t+1}) - \left\langle \nabla\widetilde{\phi}_U^t(\widetilde{p}^{t+1}), p_U^t - \bar{p}_U^{t+1} \right\rangle \\
&\quad + D^{\widetilde{t}}(\bar{p}^{t+1}, \widetilde{p}^{t+1})
\end{aligned}
$$

$$\begin{aligned}
&= D_V^t(p^t, \bar{p}^{t+1}) + \phi_V^t(p^{t+1}) - \phi_V^{t+1}(p^{t+1}) && \text{(By Eq. (48))}\\
&\quad + D_U^{t,t+1}(p^t, \bar{p}^{t+1}) + D^{\tilde{t}}(\bar{p}^{t+1}, \widetilde{p}^{t+1}) && \text{(By Eq. (47))}
\end{aligned}$$

which concludes the proof. $\qquad\square$

## B.2 Regret on Sub-Optimal Arms

**Lemma B.2.1.** *For the Decoupled-Tsallis-INF algorithm, the following holds:*

$$\mathbb{E}\left[\sum_{t=T_0+1}^{T} D_V^t(p^t, \bar{p}^{t+1}) + \phi_V^t(p^{t+1}) - \phi_V^{t+1}(p^{t+1})\right] \le \mathcal{O}\left(\mathbb{E}\left[\sum_{t=T_0+1}^{T}\sum_{i\in V}\frac{K^{1/6}\left(p_i^t\right)^{2/3}}{\sqrt{t}}\right]\right).$$

*for any $T_0 \ge 16K$ and any subset $V \subseteq [K]$.*

*Proof.* The stability term $D_V^t(p^t, \bar{p}^{t+1})$, in conditional expectation, is bounded as:

$$\begin{aligned}
&\mathbb{E}^t\left[D_V^t\left(p^t, \bar{p}^{t+1}\right)\right]\\
&= \mathbb{E}^t\left[\left\langle p_V^t - \bar{p}_V^{t+1}, \widehat{\ell}_V^t\right\rangle - D_V^t\left(\bar{p}^{t+1}, p^t\right)\right]\\
&\le \mathcal{O}\left(\frac{1}{\gamma^t}\sum_{i\in V}\frac{(p_i^t)^{4/3}}{g_i^t}\right)\\
&= \mathcal{O}\left(\frac{1}{\gamma^t}\left(\sum_{i\in V}\left(p_i^t\right)^{2/3}\right)\left(\sum_{i\in[K]}\left(p_i^t\right)^{2/3}\right)\right) && (49)\\
&\le \mathcal{O}\left(\frac{K^{1/3}}{\gamma^t}\left(\sum_{i\in V}\left(p_i^t\right)^{2/3}\right)\right)\\
&= \mathcal{O}\left(\frac{K^{1/6}}{\sqrt{t}}\sum_{i\in V}\left(p_i^t\right)^{2/3}\right), && (50)
\end{aligned}$$

where the first step applies Lemma C.5.3; the third step uses the definition of $g_i^t$; the forth step follows from the fact $\sum_{i\in[K]}\left(p_i^t\right)^{2/3} \le K^{1/3}$; the last step uses the definition of $\gamma^t = K^{1/6}\sqrt{t}$.

On the other hand, the penalty term can be bounded as

$$\begin{aligned}
&\mathbb{E}^t\left[\phi_V^t(p^{t+1}) - \phi_V^{t+1}(p^{t+1})\right]\\
&= \mathcal{O}\left((\gamma^{t+1} - \gamma^t)\sum_{i\in V}\left(p_i^{t+1}\right)^{2/3}\right) \le \mathcal{O}\left(\frac{K^{1/6}}{\sqrt{t+1}}\sum_{i\in V}\left(p_i^{t+1}\right)^{2/3}\right), && (51)
\end{aligned}$$

where the second step uses $\gamma^{t+1} - \gamma^t \le \frac{K^{1/6}}{\sqrt{t+1}}$. By summing up Eq. (50) and Eq. (51) over all $t$ from $T_0 + 1$ to $T$, we arrive at the desired bound. $\qquad\square$

## B.3 Regret on Optimal Arms

Since the standard $2/3$-Tsallis entropy regularizer is twice differentiable and its partial derivatives are concave (that is, $(-x^{2/3})' = -2/3 \cdot x^{-1/3}$ is concave on $\mathbb{R}_{>0}$), the first condition of Theorem 4.3 holds. As the Decoupled-Tsallis-INF algorithm adopts $\gamma^t = K^{1/6}\sqrt{t}$, one can easily verify that the second condition of the theorem also holds. Hence, one can apply Theorem 4.3 to bound the regret on optimal arms by zero, i.e., $D_U^{t,t+1}(p^t, \bar{p}^{t+1}) - D_U^{t,t+1}(q^t, q^{t+1}) \le 0$ for $\forall t$.

## B.4 Residual Regret

In this section, our goal is to prove the following lemma, which upper-bounds the residual regret by a self-bounding term.

**Lemma B.4.1.** *For the Decoupled-Tsallis-INF algorithm, the following holds for any $T_0 \geq 16K$ and any subset $V \subseteq [K]$:*

$$\mathbb{E}\left[\sum_{t=T_0+1}^{T} D^{\widetilde{t}}\left(\bar{p}^{t+1}, \widetilde{p}^{t+1}\right)\right] \leq \mathcal{O}\left(\mathbb{E}\left[\sum_{t=T_0+1}^{T} \sum_{i \in V} \frac{K^{1/6}\left(p_i^t\right)^{2/3}}{\sqrt{t}}\right]\right).$$

To show Lemma B.4.1, we start from a decomposition of the residual regret. Similar to the analysis of the residual regret for MAB in Section A.4, here, bounding the residual regret requires us to carefully analyze the following KKT conditions:

$$\nabla \phi_U^{t+1}(\bar{p}^{t+1}) = \nabla \phi_U^t(p^t) - \widehat{\ell}_U^t + \lambda_U \cdot \mathbf{1}_U,$$
$$\nabla \phi_V^t(\bar{p}^{t+1}) = \nabla \phi_V^t(p^t) - \widehat{\ell}_V^t + \lambda_V \cdot \mathbf{1}_V,$$
$$\nabla \widetilde{\phi}_U^t(\widetilde{p}^{t+1}) = \nabla \phi_U^{t+1}(\widetilde{p}^{t+1}) = \nabla \phi_U^t(p^t) - \widehat{\ell}_U^t + \lambda_K \cdot \mathbf{1}_U,$$
$$\nabla \widetilde{\phi}_V^t(\widetilde{p}^{t+1}) = \nabla \phi_V^t(\widetilde{p}^{t+1}) = \nabla \phi_V^t(p^t) - \widehat{\ell}_V^t + \lambda_K \cdot \mathbf{1}_V,$$

where $\lambda_U, \lambda_V, \lambda_K$ are corresponding Lagrange multipliers for $\bar{p}^{t+1}$ and $\widetilde{p}^{t+1}$. Note that, according to the definition of $\bar{p}_V^{t+1}$ in Eq. (36) and the fact that $\widetilde{\phi}^t(x) = \phi_V^t(x) + \phi_U^{t+1}(x)$ for any $x$, the second and the forth conditions are slightly different from those in Section A.4.

We start from the following decomposition of $D^{\widetilde{t}}\left(\bar{p}^{t+1}, \widetilde{p}^{t+1}\right)$. For any $c \in \mathbb{R}$, we have

$$D^{\widetilde{t}}\left(\bar{p}^{t+1}, \widetilde{p}^{t+1}\right)$$
$$= \widetilde{\phi}^t(\bar{p}^{t+1}) - \widetilde{\phi}^t(\widetilde{p}^{t+1}) - \left\langle \nabla \widetilde{\phi}^t(\widetilde{p}^{t+1}), \bar{p}^{t+1} - \widetilde{p}^{t+1} \right\rangle$$
$$= \widetilde{\phi}^t(\bar{p}^{t+1}) - \widetilde{\phi}^t(\widetilde{p}^{t+1}) - \left\langle \nabla \widetilde{\phi}^t(\widetilde{p}^{t+1}) + c \cdot \mathbf{1}_K, \bar{p}^{t+1} - \widetilde{p}^{t+1} \right\rangle$$
$$= \widetilde{\phi}^t(\bar{p}^{t+1}) - \widetilde{\phi}^t(\widetilde{p}^{t+1}) - \left\langle \nabla \widetilde{\phi}^t(\widetilde{p}^{t+1}) - \lambda_V \cdot \mathbf{1}_V - \lambda_U \cdot \mathbf{1}_U + c \cdot \mathbf{1}_K, \bar{p}^{t+1} - \widetilde{p}^{t+1} \right\rangle$$
$$= \left\langle \lambda_V \cdot \mathbf{1}_V - c \cdot \mathbf{1}_K, \bar{p}_V^{t+1} - p_V^{t+1} \right\rangle + \widetilde{\phi}_V^t(\bar{p}^{t+1}) - \widetilde{\phi}_V^t(\widetilde{p}^{t+1}) - \left\langle \nabla \widetilde{\phi}_V^t(\bar{p}^{t+1}), \bar{p}_V^{t+1} - \widetilde{p}_V^{t+1} \right\rangle$$
$$\quad + \left\langle \lambda_U \cdot \mathbf{1}_U - c \cdot \mathbf{1}_K, \bar{p}_U^{t+1} - \widetilde{p}_U^{t+1} \right\rangle + \widetilde{\phi}_U^t(\bar{p}^{t+1}) - \widetilde{\phi}_U^t(\widetilde{p}^{t+1}) - \left\langle \nabla \widetilde{\phi}_U^t(\bar{p}^{t+1}), \bar{p}_U^{t+1} - \widetilde{p}_U^{t+1} \right\rangle$$
$$= \left\langle \lambda_V \cdot \mathbf{1}_V - c \cdot \mathbf{1}_K, \bar{p}_V^{t+1} - p_V^{t+1} \right\rangle + \phi_V^t(\bar{p}^{t+1}) - \phi_V^t(\widetilde{p}^{t+1}) - \left\langle \nabla \phi_V^t(\bar{p}^{t+1}), \bar{p}_V^{t+1} - \widetilde{p}_V^{t+1} \right\rangle$$
$$\quad + \left\langle \lambda_U \cdot \mathbf{1}_U - c \cdot \mathbf{1}_K, \bar{p}_U^{t+1} - \widetilde{p}_U^{t+1} \right\rangle + \phi_U^{t+1}(\bar{p}^{t+1}) - \phi_U^{t+1}(\widetilde{p}^{t+1}) - \left\langle \nabla \phi_U^{t+1}(\bar{p}^{t+1}), \bar{p}_U^{t+1} - \widetilde{p}_U^{t+1} \right\rangle$$
$$= \left\langle \lambda_V \cdot \mathbf{1}_V - c \cdot \mathbf{1}_K, \bar{p}_V^{t+1} - \widetilde{p}_V^{t+1} \right\rangle - D_V^t(\widetilde{p}^{t+1}, \bar{p}^{t+1})$$
$$\quad + \left\langle \lambda_U \cdot \mathbf{1}_U - c \cdot \mathbf{1}_K, \bar{p}_U^{t+1} - \widetilde{p}_U^{t+1} \right\rangle - D_U^{t+1}(\widetilde{p}^{t+1}, \bar{p}^{t+1}),$$

where the second step uses the fact $\left\langle c \cdot \mathbf{1}_K, \bar{p}^{t+1} - \widetilde{p}^{t+1} \right\rangle = 0$ for any $c \in \mathbb{R}$; the third step follows from the facts that $\nabla \widetilde{\phi}_U^{t+1}(\widetilde{p}^{t+1}) = \nabla \phi_U^{t+1}(\widetilde{p}^{t+1}) = \nabla \phi_U^{t+1}(\bar{p}^{t+1}) - \lambda_U \cdot \mathbf{1}_U + \lambda_K \cdot \mathbf{1}_K$ and similarly that $\nabla \widetilde{\phi}_V^{t+1}(\widetilde{p}^{t+1}) = \nabla \phi_V^{t+1}(\bar{p}^{t+1}) - \lambda_V \cdot \mathbf{1}_V + \lambda_K \cdot \mathbf{1}_K$, which are derived from the KKT conditions above, and $\left\langle \lambda_K \cdot \mathbf{1}_K, \bar{p}^{t+1} - \widetilde{p}^{t+1} \right\rangle = 0$; the forth step follows from the fact that $\left\langle \lambda_U \cdot \mathbf{1}_U, \bar{p}_V^{t+1} - \bar{p}_V^{t+1} \right\rangle = \left\langle \lambda_V \cdot \mathbf{1}_V, \bar{p}_U^{t+1} - \bar{p}_U^{t+1} \right\rangle = 0$; the fifth step uses the definition of $\widetilde{\phi}^t$.

Now, we choose $c$ as

$$c = \frac{\lambda_U \sum_{i \in U}\left(\bar{p}_i^{t+1}\right)^{4/3} + \lambda_V \sum_{i \in V}\left(\bar{p}_i^{t+1}\right)^{4/3}}{\sum_{i \in [K]}\left(\bar{p}_i^{t+1}\right)^{4/3}}, \tag{52}$$

which minimizes a subsequent bound (Eq. (53)) below. With this choice of $c$, we apply Lemma C.5.3 (the condition required by the lemma is verified at the end of Appendix B.5) to get for any $t \geq T_0 + 1$,

$$D^{\widetilde{t}}(\bar{p}^{t+1}, \widetilde{p}^{t+1}) \leq \mathcal{O}\left(\sum_{i \in U} \frac{\left(\bar{p}_i^{t+1}\right)^{4/3}}{\gamma^{t+1}}(\lambda_U - c)^2 + \sum_{i \in V} \frac{\left(\bar{p}_i^{t+1}\right)^{4/3}}{\gamma^t}(\lambda_V - c)^2\right)$$
$$\leq \mathcal{O}\left(\sum_{i \in U} \frac{\left(\bar{p}_i^{t+1}\right)^{4/3}}{\gamma^t}(\lambda_U - c)^2 + \sum_{i \in V} \frac{\left(\bar{p}_i^{t+1}\right)^{4/3}}{\gamma^t}(\lambda_V - c)^2\right) \tag{53}$$

$$= \mathcal{O}\left( \frac{\left(\sum_{i \in V} \left(\bar{p}_i^{t+1}\right)^{4/3}\right)\left(\sum_{i \in U} \left(\bar{p}_i^{t+1}\right)^{4/3}\right)}{\gamma^t \sum_{i \in [K]} \left(\bar{p}_i^{t+1}\right)^{4/3}} (\lambda_V - \lambda_U)^2 \right)$$

$$\leq \mathcal{O}\left( \frac{\left(\sum_{i \in V} \left(\bar{p}_i^{t+1}\right)^{4/3}\right)\left(\sum_{i \in U} \left(\bar{p}_i^{t+1}\right)^{4/3}\right)}{\gamma^t \sum_{i \in [K]} \left(\bar{p}_i^{t+1}\right)^{4/3}} (\lambda_U^2 + \lambda_V^2) \right), \qquad (54)$$

where the third step applies the choice of $c$ and the last step uses $(x-y)^2 \leq 2(x^2+y^2)$ for any $x, y \in \mathbb{R}$.

To further bound the regret in Eq. (54), we need to know the range of $\lambda_U$ and $\lambda_V$. According to Lemma C.4.2, we use Eq. (80) and Eq. (81) to obtain the upper and lower bounds of $\lambda_U$, and apply Eq. (82) and Eq. (83) to get upper and lower bounds of $\lambda_V$, which are summarized as:

$$\lambda_U \leq \frac{\sum_{i \in U} (p_i^t)^{4/3} \widehat{\ell}_i^t}{\sum_{i \in U} (p_i^t)^{4/3}}, \quad \lambda_U \geq -2\left(\gamma^{t+1} - \gamma^t\right) \frac{\sum_{i \in U} \bar{p}_i^{t+1}}{\sum_{i \in U} \left(\bar{p}_i^{t+1}\right)^{4/3}},$$

$$\lambda_V \leq \frac{\sum_{i \in V} (p_i^t)^{4/3} \widehat{\ell}_i^t}{\sum_{i \in V} (p_i^t)^{4/3}}, \quad \lambda_V \geq \frac{\sum_{i \in V} \left(\bar{p}_i^{t+1}\right)^{4/3} \widehat{\ell}_i^t}{\sum_{i \in V} \left(\bar{p}_i^{t+1}\right)^{4/3}} \geq 0,$$

where the last step of the lower bound $\lambda_V \geq 0$ follows from $\bar{p}_i^{t+1} \geq 0$ and $\widehat{\ell}_i^t \geq 0$ for any $t, i$.

In what follows, we continue to bound the $\lambda_U$-related part of Eq. (54) and the $\lambda_V$-related part respectively. As our goal is to bound $\mathbb{E}[\sum_{t=T_0+1}^T D^{\widetilde{t}}(\bar{p}^{t+1}, \widehat{p}^{t+1})]$, it suffices to bound these terms in conditional expectation for every round $t \geq T_0 + 1$.

**Bounding the $\lambda_U$-related term** First, we consider the term in Eq. (54) related to $\lambda_U$ and decompose it as:

$$\mathbb{E}^t\left[ \frac{\left(\sum_{i \in V} \left(\bar{p}_i^{t+1}\right)^{4/3}\right)\left(\sum_{i \in U} \left(\bar{p}_i^{t+1}\right)^{4/3}\right)}{\gamma^t \sum_{i \in [K]} \left(\bar{p}_i^{t+1}\right)^{4/3}} \lambda_U^2 \right]$$

$$= \mathbb{E}^t\left[ \frac{\left(\sum_{i \in V} \left(\bar{p}_i^{t+1}\right)^{4/3}\right)\left(\sum_{i \in U} \left(\bar{p}_i^{t+1}\right)^{4/3}\right)}{\gamma^t \sum_{i \in [K]} \left(\bar{p}_i^{t+1}\right)^{4/3}} \lambda_U^2 \left(\mathbb{I}\{\lambda_U \geq 0\} + \mathbb{I}\{\lambda_U < 0\}\right) \right].$$

For the case of $\mathbb{I}\{\lambda_U \geq 0\}$, we use the upper bound of $\lambda_U$ to show

$$\mathbb{E}^t\left[ \frac{\left(\sum_{i \in V} \left(\bar{p}_i^{t+1}\right)^{4/3}\right)\left(\sum_{i \in U} \left(\bar{p}_i^{t+1}\right)^{4/3}\right)}{\gamma^t \sum_{i \in [K]} \left(\bar{p}_i^{t+1}\right)^{4/3}} \lambda_U^2 \mathbb{I}\{\lambda_U \geq 0\} \right]$$

$$\leq \mathcal{O}\left( \mathbb{E}^t\left[ \frac{\left(\sum_{i \in V} \left(\bar{p}_i^{t+1}\right)^{4/3}\right)\left(\sum_{i \in U} \left(\bar{p}_i^{t+1}\right)^{4/3}\right)}{\gamma^t \sum_{i \in [K]} \left(\bar{p}_i^{t+1}\right)^{4/3}} \left( \frac{\sum_{i \in U} (p_i^t)^{4/3} \widehat{\ell}_i^t}{\sum_{i \in U} (p_i^t)^{4/3}} \right)^2 \right] \right). \qquad (55)$$

For the case of $\mathbb{I}\{\lambda_U < 0\}$, we use the lower bound of $\lambda_U$ to show

$$\mathbb{E}^t\left[ \frac{\left(\sum_{i \in V} \left(\bar{p}_i^{t+1}\right)^{4/3}\right)\left(\sum_{i \in U} \left(\bar{p}_i^{t+1}\right)^{4/3}\right)}{\gamma^t \sum_{i \in [K]} \left(\bar{p}_i^{t+1}\right)^{4/3}} \lambda_U^2 \mathbb{I}\{\lambda_U < 0\} \right]$$

$$\leq \mathcal{O}\left( \mathbb{E}^t\left[ \frac{\left(\gamma^{t+1} - \gamma^t\right)^2}{\gamma^t} \frac{\left(\sum_{i \in V} \left(\bar{p}_i^{t+1}\right)^{4/3}\right)\left(\sum_{i \in U} \bar{p}_i^{t+1}\right)^2}{\left(\sum_{i \in [K]} \left(\bar{p}_i^{t+1}\right)^{4/3}\right)\left(\sum_{i \in U} \left(\bar{p}_i^{t+1}\right)^{4/3}\right)} \right] \right). \qquad (56)$$

Now, our goal is to bound Eq. (55) and Eq. (56), which are shown below respectively.

**Bounding Eq. (55)** For this part, we bound it as

$$\mathbb{E}^t \left[ \frac{\left( \sum_{i \in V} \left( \bar{p}_i^{t+1} \right)^{4/3} \right) \left( \sum_{i \in U} \left( \bar{p}_i^{t+1} \right)^{4/3} \right)}{\gamma^t \sum_{i \in [K]} \left( \bar{p}_i^{t+1} \right)^{4/3}} \cdot \left( \frac{\sum_{i \in U} \left( p_i^t \right)^{4/3} \widehat{\ell}_i^t}{\sum_{i \in U} \left( p_i^t \right)^{4/3}} \right)^2 \right]$$

$$= \mathbb{E}^t \left[ \frac{\left( \sum_{i \in V} \left( \bar{p}_i^{t+1} \right)^{4/3} \right) \left( \sum_{i \in U} \left( \bar{p}_i^{t+1} \right)^{4/3} \right)}{\gamma^t \sum_{i \in [K]} \left( \bar{p}_i^{t+1} \right)^{4/3}} \cdot \frac{\sum_{i \in U} \left( \left( p_i^t \right)^{4/3} \widehat{\ell}_i^t \right)^2}{\left( \sum_{i \in U} \left( p_i^t \right)^{4/3} \right)^2} \right]$$

$$\leq \mathbb{E}^t \left[ \left( \sum_{i \in V} \left( \bar{p}_i^{t+1} \right)^{4/3} \right)^{1/2} \left( \sum_{i \in U} \left( \bar{p}_i^{t+1} \right)^{4/3} \right)^{1/2} \frac{\sum_{i \in U} \left( \left( p_i^t \right)^{4/3} \widehat{\ell}_i^t \right)^2}{2\gamma^t \left( \sum_{i \in U} \left( p_i^t \right)^{4/3} \right)^2} \right]$$

$$\leq \mathbb{E}^t \left[ 8 \left( \sum_{i \in V} \left( p_i^t \right)^{4/3} \right)^{1/2} \left( \sum_{i \in U} \left( p_i^t \right)^{4/3} \right)^{1/2} \frac{\sum_{i \in U} \left( \left( p_i^t \right)^{4/3} \widehat{\ell}_i^t \right)^2}{2\gamma^t \left( \sum_{i \in U} \left( p_i^t \right)^{4/3} \right)^2} \right]$$

$$= 4 \left( \sum_{i \in V} \left( p_i^t \right)^{4/3} \right)^{1/2} \frac{\mathbb{E}^t \left[ \sum_{i \in U} \left( \left( p_i^t \right)^{4/3} \widehat{\ell}_i^t \right)^2 \right]}{\gamma^t \left( \sum_{i \in U} \left( p_i^t \right)^{4/3} \right)^{3/2}},$$

where the first step follows from the fact $\widehat{\ell}_i^t \cdot \widehat{\ell}_j^t = 0$ for any $i \neq j$ and the definition of $\widehat{\ell}_i^t$; the second step uses

$$\left( \sum_{i \in [K]} \left( \bar{p}_i^{t+1} \right)^{4/3} \right) = \left( \sum_{i \in U} \left( \bar{p}_i^{t+1} \right)^{4/3} + \sum_{i \in V} \left( \bar{p}_i^{t+1} \right)^{4/3} \right) \geq 2 \left( \sum_{i \in U} \left( \bar{p}_i^{t+1} \right)^{4/3} \right)^{1/2} \left( \sum_{i \in V} \left( \bar{p}_i^{t+1} \right)^{4/3} \right)^{1/2};$$

(57)

the third step applies Lemma B.5.1 and Corollary B.5.2 to obtain the multiplicative relation on $U$ and $V$, respectively; and the last step holds since $p^t$ is deterministic given the history.

Furthermore, by direct calculation, we have

$$\left( \sum_{i \in V} \left( p_i^t \right)^{4/3} \right)^{1/2} \frac{\mathbb{E}^t \left[ \sum_{i \in U} \left( \left( p_i^t \right)^{4/3} \widehat{\ell}_i^t \right)^2 \right]}{\gamma^t \left( \sum_{i \in U} \left( p_i^t \right)^{4/3} \right)^{3/2}}$$

$$\leq \left( \sum_{i \in V} \left( p_i^t \right)^{4/3} \right)^{1/2} \frac{\left( \sum_{i \in U} \left( p_i^t \right)^2 \right) \left( \sum_{i \in [K]} \left( p_i^t \right)^{2/3} \right)}{\gamma^t \left( \sum_{i \in U} \left( p_i^t \right)^{4/3} \right)^{\frac{3}{2}}} \tag{58}$$

$$\leq \frac{1}{\gamma^t} \sqrt{\sum_{i \in V} \left( p_i^t \right)^{4/3}} \left( \sum_{i \in [K]} \left( p_i^t \right)^{2/3} \right) \tag{59}$$

$$\leq \frac{1}{\gamma^t} \left( \sum_{i \in V} \left( p_i^t \right)^{2/3} \right) \left( \sum_{i \in [K]} \left( p_i^t \right)^{2/3} \right) \tag{60}$$

$$\leq \frac{K^{1/6}}{\sqrt{t}} \left( \sum_{i \in V} \left( p_i^t \right)^{2/3} \right), \tag{61}$$

where Eq. (58) follows from

$$\mathbb{E}^t \left[ \sum_{i \in U} \left( \left( p_i^t \right)^{4/3} \widehat{\ell}_i^t \right)^2 \right] = \mathbb{E}^t \left[ \sum_{i \in U} \left( \left( p_i^t \right)^{4/3} \frac{\mathbb{I}\{j^t = i\} \ell_i^t}{g_i^t} \right)^2 \right] \qquad \text{(by Eq. (35))}$$

$$\leq \sum_{i \in U} \frac{\left(p_i^t\right)^{8/3}}{g_i^t} \qquad\qquad (\ell_i^t \leq 1 \text{ and } \mathbb{E}^t\left[\mathbb{I}\{j^t = i\}\right] = g_i^t)$$

$$= \left(\sum_{i \in U} \left(p_i^t\right)^2\right)\left(\sum_{i \in [K]} \left(p_i^t\right)^{2/3}\right); \qquad\qquad \text{(by Eq. (35))}$$

Eq. (59) and Eq. (60) use the following, respectively:

$$\left(\sum_{i \in U} \left(p_i^t\right)^{4/3}\right)^{3/2} \geq \left(\sum_{i \in U} \left(p_i^t\right)^2\right) \text{ since } \|x\|_2 \leq \|x\|_{\frac{4}{3}}, \quad \text{and} \quad \sqrt{\sum_{i \in V} \left(p_i^t\right)^{4/3}} \leq \left(\sum_{i \in V} \left(p_i^t\right)^{2/3}\right);$$

and Eq. (61) holds by the facts $\sum_{i \in [K]} \left(p_i^t\right)^{2/3} \leq K^{1/3}$ and $\gamma^t = K^{1/6}\sqrt{t}$. Thus, the cumulative regret of the term in Eq. (55) from round $T_0 + 1$ to round $T$ can be bounded as:

$$\mathbb{E}\left[\sum_{t=T_0+1}^T \frac{\left(\sum_{i \in V}\left(\bar{p}_i^{t+1}\right)^{4/3}\right)\left(\sum_{i \in U}\left(\bar{p}_i^{t+1}\right)^{4/3}\right)}{\gamma^t \sum_{i \in [K]}\left(\bar{p}_i^{t+1}\right)^{4/3}}\lambda_U^2\mathbb{I}\{\lambda_U \geq 0\}\right] \leq \mathcal{O}\left(\mathbb{E}\left[\sum_{t=T_0+1}^T \frac{K^{1/6}}{\sqrt{t}}\sum_{i \in V}\left(p_i^t\right)^{2/3}\right]\right).$$
$$(62)$$

**Bounding Eq. (56)** For this term, we have

$$\mathbb{E}^t\left[\frac{\left(\gamma^{t+1} - \gamma^t\right)^2}{\gamma^t} \frac{\left(\sum_{i \in V}\left(\bar{p}_i^{t+1}\right)^{4/3}\right)\left(\sum_{i \in U}\bar{p}_i^{t+1}\right)^2}{\left(\sum_{i \in [K]}\left(\bar{p}_i^{t+1}\right)^{4/3}\right)\left(\sum_{i \in U}\left(\bar{p}_i^{t+1}\right)^{4/3}\right)}\right]$$

$$\leq \mathcal{O}\left(\mathbb{E}^t\left[\frac{K^{1/6}}{t^{3/2}}\frac{\sqrt{\sum_{i \in V}\left(\bar{p}_i^{t+1}\right)^{4/3}}\left(\sum_{i \in U}\bar{p}_i^{t+1}\right)^2}{\left(\sum_{i \in U}\left(\bar{p}_i^{t+1}\right)^{4/3}\right)^{3/2}}\right]\right)$$

$$\leq \mathcal{O}\left(\mathbb{E}^t\left[\frac{K^{1/6}}{t^{3/2}}\frac{\sqrt{\sum_{i \in V}\left(p_i^t\right)^{4/3}}\left(\sum_{i \in U}\bar{p}_i^{t+1}\right)^2}{\left(\sum_{i \in U}\left(\bar{p}_i^{t+1}\right)^{4/3}\right)^{3/2}}\right]\right)$$

$$\leq \mathcal{O}\left(\mathbb{E}^t\left[\frac{K^{1/6}}{t^{3/2}}\frac{\left(\sum_{i \in V}\left(p_i^t\right)^{2/3}\right)\left(\sum_{i \in U}\bar{p}_i^{t+1}\right)^2}{\left(\sum_{i \in U}\left(\bar{p}_i^{t+1}\right)^{4/3}\right)^{3/2}}\right]\right)$$

$$= \mathcal{O}\left(\mathbb{E}^t\left[\frac{K^{1/6}\sum_{i \in V}\left(p_i^t\right)^{2/3}}{t^{3/2}\left(\sum_{i \in U}\left(\frac{\left(\bar{p}_i^{t+1}\right)}{\sum_{i \in U}\bar{p}_i^{t+1}}\right)^{4/3}\right)^{3/2}}\right]\right),$$

where the first step follows from $\gamma^{t+1} - \gamma^t \leq \frac{K^{1/6}}{\sqrt{t}}$, $\gamma^t = K^{1/6}\sqrt{t}$, and Eq. (57); the second step applies Corollary B.5.2 to obtain the multiplicative relation; the third step applies $\sqrt{\sum_{i \in V}\left(p_i^t\right)^{4/3}} \leq \sum_{i \in V}\left(p_i^t\right)^{2/3}$; the last step divides $\left(\sum_{i \in U}\bar{p}_i^{t+1}\right)^2 = \left(\left(\sum_{i \in U}\bar{p}_i^{t+1}\right)^{4/3}\right)^{3/2}$ for both the numerator and the denominator. By using the bound above, we have

$$\text{Eq. (56)} \leq \mathcal{O}\left(\mathbb{E}^t\left[\frac{K^{1/6}\sum_{i \in V}\left(p_i^t\right)^{2/3}}{t^{3/2}\left(\sum_{i \in U}\left(\frac{\left(\bar{p}_i^{t+1}\right)}{\sum_{i \in U}\bar{p}_i^{t+1}}\right)^{4/3}\right)^{3/2}}\right]\right)$$

$$\leq \mathcal{O}\left(\mathbb{E}^t\left[\frac{K^{2/3}\sum_{i\in V}\left(p_i^t\right)^{2/3}}{t^{3/2}}\right]\right) \leq \mathcal{O}\left(\mathbb{E}^t\left[\frac{K^{1/6}\sum_{i\in V}\left(p_i^t\right)^{2/3}}{\sqrt{t}}\right]\right),$$

where the second step uses $\sum_{i\in U}\left(\frac{p_i^t}{\sum_{j\in U}p_j^t}\right)^{4/3} \geq \frac{1}{|U|^{1/3}} \geq \frac{1}{K^{1/3}}$ and the last step follows from $T_0 \geq K \geq \sqrt{K}$. Thus, the cumulative regret of this part, starting from $t = T_0 + 1$ to $t = T$, can be bounded by

$$\mathcal{O}\left(\mathbb{E}\left[\sum_{t=T_0+1}^{T}\frac{K^{1/6}\sum_{i\in V}\left(p_i^t\right)^{2/3}}{\sqrt{t}}\right]\right). \tag{63}$$

**Bounding the $\lambda_V$-related term** We now consider the regret of Eq. (54) associated with $\lambda_V$. Note that as $\lambda_V \geq 0$ always holds, we do not need to consider two cases. Repeating the argument used to bound Eq. (55), we have

$$\mathbb{E}^t\left[\frac{\left(\sum_{i\in V}\left(\bar{p}_i^{t+1}\right)^{4/3}\right)\left(\sum_{i\in U}\left(\bar{p}_i^{t+1}\right)^{4/3}\right)}{\gamma^t\sum_{i\in[K]}\left(\bar{p}_i^{t+1}\right)^{4/3}}\lambda_V^2\right]$$

$$\leq \mathbb{E}^t\left[\frac{\left(\sum_{i\in V}\left(\bar{p}_i^{t+1}\right)^{4/3}\right)\left(\sum_{i\in U}\left(\bar{p}_i^{t+1}\right)^{4/3}\right)}{\gamma^t\sum_{i\in[K]}\left(\bar{p}_i^{t+1}\right)^{4/3}}\frac{\sum_{i\in V}\left(\left(p_i^t\right)^{4/3}\widehat{\ell}_i^t\right)^2}{\left(\sum_{i\in V}\left(p_i^t\right)^{4/3}\right)^2}\right]$$

$$\leq \mathbb{E}^t\left[\left(\sum_{i\in V}\left(\bar{p}_i^{t+1}\right)^{4/3}\right)\frac{\sum_{i\in V}\left(\left(p_i^t\right)^{4/3}\widehat{\ell}_i^t\right)^2}{\gamma^t\left(\sum_{i\in V}\left(p_i^t\right)^{4/3}\right)^2}\right] \leq \mathcal{O}\left(\mathbb{E}^t\left[\frac{\sum_{i\in V}\left(\left(p_i^t\right)^{4/3}\widehat{\ell}_i^t\right)^2}{\gamma^t\sum_{i\in V}\left(p_i^t\right)^{4/3}}\right]\right),$$

where the first step applies the upper bound of $\lambda_V$ and uses the fact that $\widehat{\ell}_i^t \cdot \widehat{\ell}_j^t = 0$ for any $i \neq j$ together with the definition of $\widehat{\ell}_i^t$; the second step bounds the fraction $\sum_{i\in U}\left(\bar{p}_i^{t+1}\right)^{4/3} / \sum_{i\in[K]}\left(\bar{p}_i^{t+1}\right)^{4/3}$ by one; the last step applies Corollary B.5.2 to obtain the multiplicative relation. By a similar argument used for $\lambda_U$, we have

$$\mathbb{E}^t\left[\frac{\sum_{i\in V}\left(\left(p_i^t\right)^{4/3}\widehat{\ell}_i^t\right)^2}{\gamma^t\sum_{i\in V}\left(p_i^t\right)^{4/3}}\right] \leq \mathcal{O}\left(\frac{\left(\sum_{i\in V}\left(p_i^t\right)^2\right)\left(\sum_{i\in[K]}\left(p_i^t\right)^{2/3}\right)}{\gamma^t\sum_{i\in V}\left(p_i^t\right)^{4/3}}\right)$$

$$\leq \mathcal{O}\left(\frac{K^{1/6}}{\sqrt{t}}\left(\sum_{i\in V}\left(p_i^t\right)^{2/3}\right)\right), \tag{64}$$

where the last step uses $\sum_{i\in V}\left(p_i^t\right)^2 \leq \left(\sum_{i\in V}\left(p_i^t\right)^{2/3}\right)\left(\sum_{i\in V}\left(p_i^t\right)^{4/3}\right)$, $\sum_{i\in[K]}\left(p_i^t\right)^{2/3} \leq K^{1/3}$, and $\gamma^t = K^{1/6}\sqrt{t}$. To conclude, the cumulative regret of the term related to $\lambda_V$ is bounded as

$$\mathbb{E}\left[\sum_{t=T_0+1}^{T}\frac{\left(\sum_{i\in V}\left(\bar{p}_i^{t+1}\right)^{4/3}\right)\left(\sum_{i\in U}\left(\bar{p}_i^{t+1}\right)^{4/3}\right)}{\gamma^t\sum_{i\in[K]}\left(\bar{p}_i^{t+1}\right)^{4/3}}\lambda_V^2\right] \leq \mathcal{O}\left(\mathbb{E}\left[\sum_{t=T_0+1}^{T}\frac{K^{1/6}}{\sqrt{t}}\sum_{i\in V}\left(p_i^t\right)^{2/3}\right]\right). \tag{65}$$

Finally, combining Eq. (62), Eq. (63), and Eq. (65) yields the result of Lemma B.4.1.

## B.5 Auxiliary Lemmas for Analysis of Decoupled-Tsallis-INF

Since the regularizer of Decoupled-Tsallis-INF does not have an extra log-barrier, we cannot expect an entry-wise multiplicative relation as used in our MAB analysis. Hence, we follow Lemma 22 and Lemma 28 in [Ito, 2021] to show *group multiplicative relation* for $2/3$-Tsallis entropy.

**Lemma B.5.1.** *For any given $t$ and index set $\mathcal{I} \subseteq [K]$, if $x, y \in [0, 1]^K$ satisfies $\sum_{i\in\mathcal{I}}x_i = \sum_{i\in\mathcal{I}}y_i$ and for $2/3$-Tsallis entropy $\phi$ and $c \in \mathbb{R}$, we have*

$$\phi_{\mathcal{I}}^{t+1}(y) = \phi_{\mathcal{I}}^t(x) - \widehat{\ell}_{\mathcal{I}}^t + c \cdot \mathbf{1}_{\mathcal{I}}, \tag{66}$$

*then, the following holds.*

$$\sum_{i \in \mathcal{I}} (y_i)^{4/3} \le 2 \left( 1 + \frac{1}{t} \right)^2 \sum_{i \in \mathcal{I}} (x_i)^{4/3} \le 8 \sum_{i \in \mathcal{I}} (x_i)^{4/3}. \tag{67}$$

*Proof.* From Eq. (66), we have

$$\frac{3\sqrt{t+1}}{K^{-1/6}} (y_i)^{-1/3} = \frac{3\sqrt{t}}{K^{-1/6}} (x_i)^{-1/3} + \widehat{\ell}_i^t - c,$$

which implies that

$$\sqrt{t+1} \, (y_i)^{-1/3} = \sqrt{t} \, (x_i)^{-1/3} + \frac{K^{-1/6}}{3} \left( \widehat{\ell}_i^t - c \right).$$

If $c \le 0$, then, we have

$$\sqrt{t+1} \, (y_i)^{-1/3} \ge \sqrt{t} \, (x_i)^{-1/3} , \quad \text{which implies} \;\; (y_i)^{4/3} \le \left( \frac{t+1}{t} \right)^2 (x_i)^{4/3}.$$

Since the above holds for every $i \in \mathcal{I}$, the desired claim is immediate. Now, we consider $c \ge 0$. For all $i \in \mathcal{I} \backslash \{i^t\}$, we have $\sqrt{t+1} \, (y_i)^{-1/3} \le \sqrt{t} \, (x_i)^{-1/3}$, which gives $\frac{(x_i)^{1/3}}{\sqrt{t}} \le \frac{(y_i)^{1/3}}{\sqrt{t+1}}$. Rearranging it, we arrive at

$$\frac{y_i}{(t+1)^{3/2}} \ge \frac{x_i}{(t)^{3/2}}.$$

Let us define

$$z_i^t = \frac{x_i}{\sum_{i \in \mathcal{I}} x_i}, \quad \text{and} \quad \overline{z}_i^{t+1} = \left( \frac{t}{t+1} \right)^{3/2} \frac{y_i}{\sum_{i \in \mathcal{I}} y_i}.$$

By $\overline{z}_i^{t+1} \ge z_i^t$ for all $i \in \mathcal{I} \backslash \{i^t\}$ and $\sum_{i \in \mathcal{I}} \overline{z}_i^{t+1} \le \sum_{i \in \mathcal{I}} z_i^t = 1$, we can show

$$\frac{\sum_{i \in \mathcal{I}} (y_i)^{4/3}}{\sum_{i \in \mathcal{I}} (x_i)^{4/3}} = \left( \frac{t+1}{t} \right)^2 \frac{\sum_{i \in \mathcal{I}} \left( \overline{z}_i^{t+1} \right)^{4/3}}{\sum_{i \in \mathcal{I}} \left( z_i^t \right)^{4/3}} \le 2 \left( \frac{t+1}{t} \right)^2,$$

where the last step uses Lemma 28 in [Ito, 2021] by replacing the power $3/2$ by $4/3$. $\qquad \square$

Recall that $\nabla \phi_U^{t+1}(\bar{p}^{t+1}) = \nabla \phi_U^t(p^t) - \widehat{\ell}_U^t + \lambda_U \cdot \mathbf{1}_U$, which implies that Eq. (66) and $\sum_{i \in \mathcal{I}} x_i = \sum_{i \in \mathcal{I}} y_i$ in Lemma B.5.1 hold by applying $\mathcal{I} = U$, $y = \bar{p}^{t+1}$, and $x = p^t$. Hence, we have $\sum_{i \in U} \left( \bar{p}_i^{t+1} \right)^{4/3} \le 8 \sum_{i \in U} (p_i^t)^{4/3}$. By repeating the same reasoning in Lemma B.5.1 and changing $\gamma^{t+1}$ to $\gamma^t$, the following corollary is immediate.

**Corollary B.5.2.** *For any given $t$ and index set $\mathcal{I} \subseteq [K]$, if $x, y \in [0,1]^K$ satisfies $\sum_{i \in \mathcal{I}} x_i = \sum_{i \in \mathcal{I}} y_i$ and for $2/3$-Tsallis entropy $\phi$ and $c \in \mathbb{R}$, we have $\phi_\mathcal{I}^t(y) = \phi_\mathcal{I}^t(x) - \widehat{\ell}_\mathcal{I}^t + c \cdot \mathbf{1}_\mathcal{I}$, then, $\sum_{i \in \mathcal{I}} (y_i)^{4/3} \le 8 \sum_{i \in \mathcal{I}} (x_i)^{4/3}$.*

Recall that as $\bar{p}_V^{t+1}$ is computed via $\phi^t$ rather than $\phi^{t+1}$ and $\nabla \phi_V^t(\bar{p}^{t+1}) = \nabla \phi_V^t(p^t) - \widehat{\ell}_V^t + \lambda_V \cdot \mathbf{1}_V$, Corollary B.5.2 implies the multiplicative relation: $\sum_{i \in V} \left( \bar{p}_i^{t+1} \right)^{4/3} \le 8 \sum_{i \in V} (p_i^t)^{4/3}$.

Next, we present a lemma that is used to bound the regret for the first $T_0$ rounds by simply $\mathcal{O}(\sqrt{KT_0})$.

**Lemma B.5.3.** *For $q^t$ defined in Eq. (5), $p^t$ defined in Algorithm 1, and any $t \in [T]$, we have*

$$\mathbb{E}\left[ D^{t,t+1}(p^t, p^{t+1}) - D_U^{t,t+1}(q^t, q^{t+1}) \right] \le \mathcal{O}\left( \frac{\sqrt{K}}{\sqrt{t}} \right). \tag{68}$$

*Proof.* By similar arguments of [Ito, 2021, Lemma 24], we arrive at

$$D^{t,t+1}(p^t, p^{t+1}) - D_U^{t,t+1}(q^t, q^{t+1}) \le D^{t,t+1}(p^t, z^{t+1}),$$

where $z^{t+1} \in \mathbb{R}^K$ is the unconstrained projection such that $\nabla \phi^{t+1}(z^{t+1}) = \nabla \phi^t(p^t) - \widehat{\ell}^t$ (see more details in [Ito, 2021]). Then, we have for any $i \in [K]$ and any $t$,

$$\frac{K^{1/6}\sqrt{t+1}}{\left(z_i^{t+1}\right)^{1/3}} = \frac{K^{1/6}\sqrt{t}}{\left(p_i^t\right)^{1/3}} + \widehat{\ell}_i^t \ge \frac{K^{1/6}\sqrt{t}}{\left(p_i^t\right)^{1/3}},$$

which implies that for any $i \in [K]$ and any $t$,

$$\left(z_i^{t+1}\right)^{2/3} \le \left(\frac{1}{t} + 1\right)\left(p_i^t\right)^{2/3} \le 2\left(p_i^t\right)^{2/3}. \tag{69}$$

By using $\nabla \phi^{t+1}(z^{t+1}) = \nabla \phi^t(p^t) - \widehat{\ell}^t$, we add and subtract $\phi^t(z^{t+1})$ to show

$$D^{t,t+1}(p^t, z^{t+1}) = \phi^t(p^t) - \phi^{t+1}(z^{t+1}) - \left\langle \nabla \phi^{t+1}(z^{t+1}), p^t - z^{t+1} \right\rangle + \left(\phi^t(z^{t+1}) - \phi^t(z^{t+1})\right)$$

$$= \phi^t(p^t) - \phi^t(z^{t+1}) - \left\langle \nabla \phi^t(p^t) - \widehat{\ell}^t, p^t - z^{t+1} \right\rangle + \phi^t(z^{t+1}) - \phi^{t+1}(z^{t+1})$$

$$= \left\langle p^t - z^{t+1}, \widehat{\ell}^t \right\rangle - D^t\left(z^{t+1}, p^t\right) + \phi^t(z^{t+1}) - \phi^{t+1}(z^{t+1}).$$

One can bound the stability term in conditional expectation by

$$\mathbb{E}^t\left[\left\langle p^t - z^{t+1}, \widehat{\ell}^t \right\rangle - D^t\left(z^{t+1}, p^t\right)\right] \le \frac{1}{\gamma^t}\left(\sum_{i \in [K]}\left(p_i^t\right)^{2/3}\right)^2 \le \frac{1}{\gamma^t}\left(K^{1/3}\right)^2 = \frac{K^{1/2}}{\sqrt{t}}, \tag{70}$$

where the first step applies Lemma C.5.3 and uses a similar approach of Eq. (49) (changing $V$ to $[K]$); the second step uses the fact that when $p_i^t = 1/K$ for all $i \in [K]$, the square of sum is maximized.

The penalty term is bounded by

$$\phi^t(z^{t+1}) - \phi^{t+1}(z^{t+1}) \le \mathcal{O}\left(\frac{K^{1/6}}{\sqrt{t}}\sum_{i \in [K]}\left(z_i^{t+1}\right)^{2/3}\right) \le \mathcal{O}\left(\frac{K^{1/6}}{\sqrt{t}}\sum_{i \in [K]}\left(p_i^t\right)^{2/3}\right) \le \mathcal{O}\left(\frac{\sqrt{K}}{\sqrt{t}}\right), \tag{71}$$

where the first step uses a similar argument of Eq. (51); the second step applies Eq. (69); the last step follows from the fact $\sum_{i \in [K]}\left(p_i^t\right)^{2/3} \le K^{1/3}$ since $p_i^t = 1/K$ maximizes the value of $\sum_{i \in [K]}\left(p_i^t\right)^{2/3}$. Combining Eq. (70) and Eq. (71), we complete the proof. $\qquad\square$

**Sanity check for the condition in Lemma C.5.3**  Recall that we defer the sanity check for the condition of Lemma C.5.3 in Appendix B.4. This condition requires us to check that $\frac{\left(\bar{p}_j^{t+1}\right)^{1/3}(\lambda_V - c)}{\gamma^t}$ for any $j \in V$ and $\frac{\left(\bar{p}_j^{t+1}\right)^{1/3}(\lambda_U - c)}{\gamma^t}$ for any $j \in U$ can be lower-bounded by a fixed negative constant (that is, $\frac{\beta}{1-\beta}\left(e^{\frac{\beta-1}{\beta}} - 1\right) = 2\left(e^{-\frac{1}{2}} - 1\right) \approx -0.78693$ when $\beta = 2/3$).

Because the lower bound of $\lambda_V$ is zero, which is larger than that of $\lambda_U$, we will verify that $\frac{\left(\bar{p}_j^{t+1}\right)^{1/3}(\lambda_U - c)}{\gamma^t}$ can be lower-bounded for any $j \in U$ and $t \ge T_0 + 1$, and the other one can be similarly bounded. For any $j \in U$, we first show that $\frac{\left(\bar{p}_j^{t+1}\right)^{1/3}\lambda_U}{\gamma^t}$ is lower bounded as:

$$\frac{\left(\bar{p}_j^{t+1}\right)^{1/3}\lambda_U}{\gamma^t} \ge -\frac{2\left(\bar{p}_j^{t+1}\right)^{1/3}}{\gamma^t}\frac{\left(\gamma^{t+1} - \gamma^t\right)\sum_{i \in U}\bar{p}_i^{t+1}}{\sum_{i \in U}\left(\bar{p}_i^{t+1}\right)^{4/3}}$$

$$\ge -\frac{2\sum_{i \in U}\left(\bar{p}_j^{t+1}\right)^{1/3}\left(\bar{p}_i^{t+1}\right)}{t\sum_{i \in U}\left(\bar{p}_i^{t+1}\right)^{4/3}}$$

$$\geq -\frac{2|U|\left(\bar{p}_j^{t+1}\right)^{4/3} + 2\sum_{i\in U}\left(\bar{p}_i^{t+1}\right)^{4/3}}{t\sum_{i\in U}\left(\bar{p}_i^{t+1}\right)^{4/3}}$$

$$\geq -\frac{2\left(|U|+1\right)}{t} \geq -\frac{1}{4},$$

where the first step uses the lower bound of $\lambda_U$; the second step follows from the definition of learning rate $\gamma^t$; the third step follows from the fact that $x^{1/3}y \leq x^{4/3} + y^{4/3}$ for any $x, y \geq 0$; the last step uses $t \geq T_0 + 1 \geq 16K + 1$.

Then, we show that $\frac{\left(\bar{p}_j^{t+1}\right)^{1/3}c}{\gamma^t}$ is upper-bounded as:

$$\frac{\left(\bar{p}_j^{t+1}\right)^{1/3} c}{\gamma^t} = \frac{\left(\bar{p}_j^{t+1}\right)^{1/3}}{\gamma^t} \frac{\lambda_U \sum_{i\in U}\left(\bar{p}_i^{t+1}\right)^{4/3} + \lambda_V \sum_{i\in V}\left(\bar{p}_i^{t+1}\right)^{4/3}}{\sum_{i\in[K]}\left(\bar{p}_i^{t+1}\right)^{4/3}}$$

$$\leq \frac{\left(p_{i^t}^t\right)^{1/3}}{\gamma^t \sum_{i\in[K]}\left(\bar{p}_i^{t+1}\right)^{4/3}} \leq \frac{K^{1/3}\left(p_{i^t}^t\right)^{1/3}}{\gamma^t} = \frac{K^{1/6}}{\sqrt{t}} \leq \frac{1}{4},$$

where first step applies the choice of $c$ (see Eq. (52)); the second step uses upper bounds of $\lambda_U$ and $\lambda_V$; the third step bounds $\sum_{i\in[K]}\left(\bar{p}_i^{t+1}\right)^{4/3} \geq K^{-1/3}$; the fourth step uses $\gamma^t = K^{1/6}\sqrt{t}$ and bounds $\left(p_{i^t}^t\right)^{1/3} \leq 1$; the last step uses $t \geq T_0 + 1 \geq 16K + 1$.

Finally, combining the bounds above, we have

$$\frac{\left(\bar{p}_j^{t+1}\right)^{1/3}\left(\lambda_U - c\right)}{\gamma^t} \geq -\frac{1}{8} - \frac{1}{4} \geq -0.5 \geq 2\left(e^{-\frac{1}{2}} - 1\right),$$

which satisfies the condition of Lemma C.5.3.

# C  Supplementary Lemmas

## C.1  Proof of Lemma 4.1

*Proof.* As $D^{t,t+1}(p^t, p^{t+1}) = D_V^{t,t+1}(p^t, p^{t+1}) + D_U^{t,t+1}(p^t, p^{t+1})$, we first decompose $D_V^{t,t+1}(p^t, p^{t+1})$. By the definition of skewed Bregman divergence, we have

$$D_V^{t,t+1}(p^t, p^{t+1})$$
$$= \phi_V^t(p^t) - \phi_V^{t+1}(p^{t+1}) - \left\langle \nabla \phi_V^{t+1}(p^{t+1}), p_V^t - p_V^{t+1} \right\rangle$$
$$= \phi_V^t(p^t) - \phi_V^{t+1}(p^{t+1}) - \left\langle \nabla \phi_V^{t+1}(p^{t+1}), p_V^t - \bar{p}_V^{t+1} \right\rangle - \left\langle \nabla \phi_V^{t+1}(p^{t+1}), \bar{p}_V^{t+1} - p_V^{t+1} \right\rangle$$
$$= \phi_V^t(p^t) - \phi_V^{t+1}(p^{t+1}) - \left\langle \nabla \phi_V^{t+1}(\bar{p}^{t+1}), p_V^t - \bar{p}_V^{t+1} \right\rangle - \left\langle \nabla \phi_V^{t+1}(p^{t+1}), \bar{p}_V^{t+1} - p_V^{t+1} \right\rangle$$
$$= \phi_V^t(p^t) - \phi_V^{t+1}(\bar{p}^{t+1}) - \left\langle \nabla \phi_V^{t+1}(\bar{p}^{t+1}), p_V^t - \bar{p}_V^{t+1} \right\rangle$$
$$\quad + \phi_V^{t+1}(\bar{p}^{t+1}) - \phi_V^{t+1}(p^{t+1}) - \left\langle \nabla \phi_V^{t+1}(p^{t+1}), \bar{p}_V^{t+1} - p_V^{t+1} \right\rangle$$
$$= D_V^{t,t+1}\left(p^t, \bar{p}^{t+1}\right) + D_V^{t+1}\left(\bar{p}^{t+1}, p^{t+1}\right),$$

where the third step follows from the fact $\nabla \phi_V^{t+1}(\bar{p}^{t+1}) - \nabla \phi_V^{t+1}(p^{t+1}) = c \cdot \mathbf{1}_V$ for a Lagrange multiplier $c \in \mathbb{R}$ and the fact $\sum_{i \in V} p_i^t = \sum_{i \in V} \bar{p}_i^{t+1}$.

By similar arguments, we have

$$D_U^{t,t+1}(p^t, p^{t+1}) = D_U^{t,t+1}\left(p^t, \bar{p}^{t+1}\right) + D_U^{t+1}\left(\bar{p}^{t+1}, p^{t+1}\right).$$

Combining these two parts together, we have

$$D^{t,t+1}(p^t, p^{t+1}) = D_V^{t,t+1}\left(p^t, \bar{p}^{t+1}\right) + D_U^{t,t+1}\left(p^t, \bar{p}^{t+1}\right) + D^{t+1}\left(\bar{p}^{t+1}, p^{t+1}\right).$$

Subtracting $D_U^{t,t+1}(q^t, q^{t+1})$ from both sides completes the proof. $\qquad \square$

## C.2  Proof of Lemma 4.2

To prove Lemma 4.2, we first consider a more general version that takes a weight sequence $\{a_t\}_{t=1}^T$ into consideration. Note that, by simply setting $a_t = 1$ for all $t$, one recovers Lemma 4.2.

**Lemma C.2.1.** *Let $\{x_t\}_{t=1}^T$ and $\{a_t\}_{t=1}^T$ be some sequences with that $x_t, a_t > 0$ for all $t$. Then, for any $\alpha \in [0,1]$, we have*

$$\sum_{t=1}^T \frac{x_t^{1-\alpha} a_t}{\sqrt{1 + \sum_{s=1}^t x_s^{1-2\alpha}}} \leq 2 \sqrt{\left( \sum_{t=1}^T x_t a_t^2 \right) \log \left( 1 + \sum_{t=1}^T x_t^{1-2\alpha} \right)}.$$

*Proof.* For any $\eta > 0$, we have

$$\sum_{t=1}^T \frac{x_t^{1-\alpha} a_t}{\sqrt{1 + \sum_{s=1}^t x_s^{1-2\alpha}}} = \sum_{t=1}^T \frac{\left( a_t x_t^{1/2} \right) \cdot x_t^{1/2 - \alpha}}{\sqrt{1 + \sum_{s=1}^t x_s^{1-2\alpha}}} \leq \sum_{t=1}^T \eta \cdot x_t a_t^2 + \frac{1}{\eta} \cdot \frac{x_t^{1-2\alpha}}{1 + \sum_{s=1}^t x_s^{1-2\alpha}},$$

where the second step uses the AM-GM inequality.

Note that, we have

$$\sum_{t=1}^T \frac{x_t^{1-2\alpha}}{1 + \sum_{s=1}^t x_s^{1-2\alpha}} \leq \sum_{t=1}^T \int_{1 + \sum_{s=1}^{t-1} x_s^{1-2\alpha}}^{1 + \sum_{s=1}^t x_s^{1-2\alpha}} \frac{du}{u} = \int_1^{1 + \sum_{s=1}^T x_s^{1-2\alpha}} \frac{du}{u} = \log \left( 1 + \sum_{s=1}^T x_s^{1-2\alpha} \right),$$

where the first step bounds the fraction with the integral of $\frac{1}{u}$ from $1 + \sum_{s=1}^{t-1} x_s^{1-2\alpha}$ to $1 + \sum_{s=1}^t x_s^{1-2\alpha}$, and the last step follows from the Newton-Leibniz formula.

Therefore, the following bound holds for any $\eta > 0$

$$\sum_{t=1}^T \frac{x_t^{1-\alpha} a_t}{\sqrt{1 + \sum_{s=1}^t x_s^{1-2\alpha}}} \leq \eta \left( \sum_{t=1}^T x_t a_t^2 \right) + \frac{1}{\eta} \log \left( 1 + \sum_{s=1}^T x_s^{1-2\alpha} \right).$$

Finally, picking the optimal $\eta$ finishes the proof. $\qquad \square$

**Remark C.2.2.** *For any $\beta \in (0,1)$, when using the $\beta$-Tsallis entropy regularizer with $C_{\text{LOG}} \geq \frac{162\beta}{1-\beta}$, $\alpha = \beta$, and $\theta = \sqrt{\frac{1-\beta}{\beta}}$,* [Algorithm 1]{.underline} *ensures (ignoring some lower-order terms)*

$$\text{Reg}^T = \mathcal{O}\left(\sqrt{\frac{1}{\beta(1-\beta)}} \sum_{i \in [K]} \sum_{t=1}^{T} \frac{(p_i^t)^{1-\beta}}{\sqrt{1 + \sum_{s \leq t} \max\{p_i^s, 1/T\}^{1-2\beta}}}\right),$$

*in the adversarial setting. Note that, when $\beta = \frac{1}{2}$, our learning rate schedule becomes $\gamma_i^t = \sqrt{t}$ which is the same as that of the Tsallis-INF algorithm, and the regret bound above can be simplified as*

$$\text{Reg}^T = \mathcal{O}\left(\sum_{t=1}^{T} \sum_{i \in [K]} \sqrt{\frac{p_i^t}{t+1}}\right) \leq \mathcal{O}\left(\sum_{t=1}^{T} \sqrt{\frac{K}{t+1}}\right) = \mathcal{O}\left(\sqrt{KT}\right),$$

*where the second step follows from the Cauchy-Schwarz inequality:* $\sum_{i \in [K]} \sqrt{p_i^t} \leq \sqrt{K \sum_{i \in [K]} p_i^t} = \sqrt{K}$, *and the last step uses the fact that $\sum_{t=1}^{T} \frac{1}{\sqrt{t+1}} = \mathcal{O}\left(\sqrt{T}\right)$. On the other hand, for the extreme case where $\beta = 0$, we have*

$$\sum_{i \in [K]} \sum_{t=1}^{T} \frac{p_i^t}{\sqrt{1 + \sum_{s \leq t} \max\{p_i^s, 1/T\}}} \leq \sum_{i \in [K]} \sum_{t=1}^{T} \int_{\sum_{s=1}^{t-1} p_i^s}^{\sum_{s=1}^{t} p_i^s} \frac{du}{\sqrt{1+u}}$$

$$\leq 2 \sum_{i \in K} \sqrt{1 + \sum_{t=1}^{T} p_i^t} = \mathcal{O}\left(\sqrt{KT}\right),$$

*where the first step bounds the fraction by the integral of $\frac{1}{\sqrt{1+u}}$ from $\sum_{s=1}^{t-1} p_i^s$ to $\sum_{s=1}^{t} p_i^s$; the second step follows from the Newton-Leibniz formula; the last step follows from the Cauchy-Schwarz inequality.*

*These two bounds for $\beta = 0$ and $\beta = 1/2$ inspire us to conjecture the following: for any $\beta \in (0,1)$, any sequence of distributions $\{p^t\}_{t=1}^{T}$, we have*

$$\sum_{i \in [K]} \sum_{t=1}^{T} \frac{(p_i^t)^{1-\beta}}{\sqrt{1 + \sum_{s \leq t} \max\{p_i^s, 1/T\}^{1-2\beta}}} = \mathcal{O}\left(\sqrt{KT}\right). \tag{72}$$

*Clearly, if this conjecture holds, then, the extra $\sqrt{\log T}$ factor in the regret bounds for the adversarial setting can be removed when $\beta \neq 1/2$.*

## C.3 Multiplicative Relation

The added fixed amount log-barrier in the regularizer ensures multiplicative relation among $\bar{p}^{t+1}$, $p^t$, and $p^{t+1}$. To show this, we start from the following general lemma.

**Lemma C.3.1.** *Consider any constant $C_{\text{LOG}} > 0$, any learning rate vectors $\gamma, \gamma' \in \mathbb{R}_{>0}^d$ satisfying $\gamma_i' \geq \gamma_i$ for all $i \in [d]$, any loss vectors $L, L' \in \mathbb{R}_{>0}^d$, and any convex function $\psi(x)$ which ensures $\frac{\psi''(x)}{4} \leq \psi''(z) \leq 4\psi''(x)$ for any $x \in \mathbb{R}_{>0}$ and any $z \in \left[\frac{x}{2}, 2x\right]$. Define $\phi(p) = \sum_{i \in [d]} \gamma_i \psi(p_i)$ as the regularizer with learning rate $\gamma$, $\phi'(p) = \sum_{i \in [d]} \gamma_i' \psi(p_i)$ as the regularizer with learning rate $\gamma'$, $\phi_L(p) = -C_{\text{LOG}} \sum_{i \in [d]} \log p_i$ as the log-barrier regularizer, $F(p) = \langle p, L \rangle + \phi(p) + \phi_L(p)$, and $F'(p) = \langle p, L' \rangle + \phi'(p) + \phi_L(p)$. Further define $x, y \in \Omega \subseteq \mathbb{R}_{>0}^d$ as: $x = \operatorname{argmin}_{p \in \Omega} F(p)$ and $y = \operatorname{argmin}_{p \in \Omega} F'(p)$. If $C_{\text{LOG}}$ satisfies:*

$$C_{\text{LOG}} \geq \max\left\{9, 32 \sum_{i \in [d]} (L_i' - L_i)^2 x_i^2, 32 \sum_{i \in [d]} (\gamma_i' - \gamma_i)^2 \psi'(x_i)^2 x_i^2\right\}, \tag{73}$$

*then, for any $i \in [d]$, we have*

$$\frac{1}{2} x_i \leq y_i \leq 2 x_i. \tag{74}$$

*Proof.* For simplicity, we denote $H$ as the Hessian $\nabla^2 F(x)$, and $H_L$ as the Hessian $\nabla^2 \phi_L(x)$ which is a diagonal matrix with $\frac{C_{\text{LOG}}}{x_i^2}$ on its diagonal for every entry $i \in [d]$. Our goal is to show that $\|y - x\|_H \leq 1$, which is enough to guarantee Eq. (74) because $1 \geq \|y - x\|_H \geq \|y - x\|_{H_L}$, and

$$\|y - x\|_{H_L} \leq 1 \Rightarrow C_{\text{LOG}} \sum_{i \in [d]} \left( \frac{y_i - x_i}{x_i} \right)^2 \leq 1 \Rightarrow \left| \frac{y_i - x_i}{x_i} \right| \leq \frac{1}{3}, \text{ for } \forall i \in [d],$$

where the last step follows from the condition $C_{\text{LOG}} \geq 9$.

To prove $\|y - x\|_H \leq 1$, it suffices to show that for any $z \in \Omega$ that ensures $\|z - x\|_H = 1$, it holds that $F'(z) \geq F'(x)$. To see this, note that the level set $A = \{p \in \Omega : F'(p) \leq F'(x)\}$ is a convex set that contains $x$ and $y$. Clearly, any intermediate point between $x$ and $y$ belongs to $A$, and does not belong to the boundary set $\partial B = \{p : \|p - x\|_H = 1\}$ where $B = \{p : \|p - x\|_H < 1\}$ also contains $x$. Therefore, $y$ belongs to the set $B$ and guarantees that $\|y - x\|_H \leq 1$.

To this end, we first bound $F'(z)$ for any $z \in \Omega$ with $\|z - x\|_H = 1$ as

$$
\begin{aligned}
F'(z) &= F'(x) + \nabla F'(x)^\top (z - x) + \frac{1}{2} \|z - x\|_{\nabla^2 F'(\xi)}^2 \\
&= F'(x) + (\nabla F'(x) - \nabla F(x))^\top (z - x) + \nabla F(x)^\top (z - x) + \frac{1}{2} \|z - x\|_{\nabla^2 F'(\xi)}^2 \\
&\geq F'(x) + (\nabla F'(x) - \nabla F(x))^\top (z - x) + \frac{1}{2} \|z - x\|_{\nabla^2 F'(\xi)}^2 \\
&\geq F'(x) + (\nabla F'(x) - \nabla F(x))^\top (z - x) + \frac{1}{2} \|z - x\|_{\nabla^2 F(\xi)}^2 \\
&\geq F'(x) + (\nabla F'(x) - \nabla F(x))^\top (z - x) + \frac{1}{8} \|z - x\|_{\nabla^2 F(x)}^2 \\
&= F'(x) + (\nabla F'(x) - \nabla F(x))^\top (z - x) + \frac{1}{8} \|z - x\|_H^2 \\
&= F'(x) + (\nabla F'(x) - \nabla F(x))^\top (z - x) + \frac{1}{8},
\end{aligned}
$$

where the first step follows the Taylor expansion of $F'(z)$ at $x$ with $\xi$ being an intermediate point between $x$ and $z$; the third step holds due to the optimality condition of $x$, i.e., $\nabla F(x)^\top (z - x) \geq 0$; the forth step uses the fact that $\nabla^2 F'(\xi) \succeq \nabla^2 F(\xi)$; the fifth step applies the multiplicative relation between $\xi$ and $x$: $\frac{1}{2} x_i \leq \xi_i \leq 2 x_i$ for any $i \in [d]$, which indicates $\psi''(\xi_i) \geq \frac{\psi''(x_i)}{4}$ according to the property of $\psi$; the last step uses the condition that $\|z - x\|_H = 1$.

To finish the proof, we only need to show that $(\nabla F'(x) - \nabla F(x))^\top (z - x) \geq -\frac{1}{8}$. We bound $(\nabla F'(x) - \nabla F(x))^\top (z - x)$ as

$$
\begin{aligned}
&(\nabla F'(x) - \nabla F(x))^\top (z - x) \\
&= (L' - L + \nabla \phi'(x) - \nabla \phi(x))^\top (z - x) \\
&\geq -\|L' - L + \nabla \phi'(x) - \nabla \phi(x)\|_{H^{-1}} \|z - x\|_H \\
&= -\|L' - L + \nabla \phi'(x) - \nabla \phi(x)\|_{H^{-1}}^2 \\
&\geq -\|L' - L + \nabla \phi'(x) - \nabla \phi(x)\|_{H_L^{-1}}^2 \\
&= -\sum_{i \in [d]} (L'_i - L_i + (\gamma'_i - \gamma_i) \psi'(x_i))^2 \frac{x_i^2}{C_{\text{LOG}}}, \quad (75)
\end{aligned}
$$

where the second step uses Hölder's inequality, and the forth step follows from the fact that $H^{-1} \preceq H_L^{-1}$, which ensures $\|p\|_{H^{-1}} \leq \|p\|_{H_L^{-1}}$ for any $p \in \mathbb{R}^d$.

Note that, we have

$$\sum_{i \in [d]} (L'_i - L_i + (\gamma'_i - \gamma_i) \psi'(x_i))^2 \frac{x_i^2}{C_{\text{LOG}}}$$

$$\leq \frac{2}{C_{\text{LOG}}} \sum_{i \in [d]} \left(L_i' - L_i\right)^2 x_i^2 + \left(\gamma_i' - \gamma_i\right)^2 \psi'(x_i)^2 x_i^2$$

$$\leq \frac{2}{32} + \frac{2}{32} = \frac{1}{8},$$

where the second step follows from the fact that $(x + y)^2 \leq 2\left(x^2 + y^2\right)$ for any $x, y$; the second step uses Eq. (73). Plugging this inequality back to Eq. (75) finishes the proof. $\square$

In what follows, we show three applications of Lemma C.3.1 for the $\beta$-Tsallis entropy (Lemma C.3.2), the Shannon entropy (Lemma C.3.3), and the log-barrier (Lemma C.3.4), respectively.

**Lemma C.3.2** (Multiplicative Relation for $\beta$-Tsallis Entropy). *For any $\beta \in (0, 1)$, when using the $\beta$-Tsallis entropy regularizer with $C_{\text{LOG}} \geq \frac{162\beta}{1-\beta}$, $\alpha = \beta$, and $\theta = \sqrt{\frac{1-\beta}{\beta}}$, Algorithm 1 guarantees*

$$\frac{1}{2} p_i^t \leq \bar{p}_i^{t+1} \leq 2p_i^t, \quad \frac{1}{2} p_i^t \leq p_i^{t+1} \leq 2p_i^t,$$

*for all $t \in [T]$ and arm $i \in [K]$.*

*Proof.* We first apply Lemma C.3.1 to show $\frac{1}{2} p_i^t \leq p_i^{t+1} \leq 2p_i^t$. In particular, we set $\Omega$ as a probability simplex, $L = \sum_{\tau < t} \ell_i^\tau$, $L' = \sum_{\tau \leq t} \ell_i^\tau$, $\gamma = \gamma^t$, and $\gamma' = \gamma^{t+1}$ where the loss estimators and learning rates are defined in Algorithm 1. These choices naturally give $x = p^t$ and $y = p^{t+1}$. Then, we only need to check that choosing $C_{\text{LOG}} = \frac{162\beta}{1-\beta}$ can guarantee that Eq. (73) holds. Clearly, we have $C_{\text{LOG}} \geq 9$. From the definition of $\widehat{\ell}_i^t$, one can show

$$32 \sum_{i \in [K]} \left(\widehat{\ell}_i^t\right)^2 \left(p_i^t\right)^2 \leq 32 \sum_{i \in [K]} \frac{\mathbb{I}\{i^t = i\}}{\left(p_i^t\right)^2} \left(p_i^t\right)^2 = 32 \sum_{i \in [K]} \mathbb{I}\{i^t = i\} = 32.$$

Finally, by Eq. (21), we have

$$32 \sum_{i \in [K]} \left(\gamma_i^{t+1} - \gamma_i^t\right)^2 \left(p_i^t\right)^2 \left(\psi'(p_i^t)\right)^2$$

$$= 32 \sum_{i \in [K]} \left(\gamma_i^{t+1} - \gamma_i^t\right)^2 \left(p_i^t\right)^2 \left(\frac{\beta \left(p_i^t\right)^{\beta-1}}{1-\beta}\right)^2$$

$$\leq 32 \sum_{i \in [K]} \frac{\beta^2}{(1-\beta)^2} \cdot \frac{(1-\beta)}{\beta} \cdot \frac{\left(\max\left\{p_i^t, 1/T\right\}\right)^{2-4\beta}}{1 + \sum_{k=1}^{t} \left(\max\left\{p_i^k, 1/T\right\}\right)^{1-2\beta}} \left(p_i^t\right)^{2\beta}$$

$$\leq 32 \sum_{i \in [K]} \frac{\beta}{(1-\beta)} \left(\max\left\{p_i^t, 1/T\right\}\right)^{1-2\beta} \left(\max\left\{p_i^t, 1/T\right\}\right)^{2\beta}$$

$$\leq 32 \sum_{i \in [K]} \frac{\beta}{(1-\beta)} \left(p_i^t + \frac{1}{T}\right)$$

$$\leq 32 \cdot \frac{\beta}{1-\beta} \left(\sum_{i \in [K]} p_i^t + \frac{K}{T}\right) \leq 64 \cdot \frac{\beta}{1-\beta},$$

where the third step uses the inequality that $\max\left\{p_i^t, 1/T\right\} \leq p_i^t + \frac{1}{T}$.

To prove $\frac{1}{2} p_i^t \leq \bar{p}_i^{t+1} \leq 2p_i^t$ for all arms $i \in [K]$, we show $\frac{1}{2} p_i^t \leq \bar{p}_i^{t+1} \leq 2p_i^t$ for all arms $i \in U$ and all arms $i \in V$, respectively. Since the proof ideas of both are the same, we only show the one related to set $U$. We set $\Omega = \{x \in \mathbb{R}_{\geq 0}^K : \sum_{i \in U} x_i = \sum_{i \in U} p_i^t, \sum_{i \in V} x_i = 0\}$ and maintain all settings the same as the above. Note that, Eq. (73) only depends on loss estimators and learning rate schedules, and thus, it holds for new decision space $\Omega$. Repeating a similar argument, we can obtain the multiplicative relation related to arms in $V$. $\square$

**Lemma C.3.3** (Multiplicative Relation for Shannon Entropy). *When using the Shannon entropy regularizer with $C_{\mathrm{LOG}} \geq 162 \log K$, $\alpha = 1$, and $\theta = \sqrt{\frac{1}{\log T}}$, Algorithm 1 guarantees*

$$\frac{1}{2} p_i^t \leq \bar{p}_i^{t+1} \leq 2 p_i^t, \quad \frac{1}{2} p_i^t \leq p_i^{t+1} \leq 2 p_i^t,$$

*for all $t \in [T]$ and arm $i \in [K]$.*

*Proof.* We consider applying Lemma C.3.1 to prove the multiplicative relation between $p^t$ and $p^{t+1}$ with similar setups in the proof Lemma C.3.2. For the Shannon entropy regularizer in Algorithm 1, we have $\psi(x) = x \log\left(\frac{x}{e}\right)$ and $\psi'(x) = \log(x)$.

By the same argument, we know that $162 \log K \geq 32 \geq 32 \sum_{i \in [K]} \left(\widehat{\ell}_i^t\right)^2 (p_i^t)^2$ and $162 \log K \geq 9$.
Then, we only need to verify that $162 \log K \geq 32 \sum_{i \in [K]} \left(\gamma_i^{t+1} - \gamma_i^t\right)^2 \psi'(p_i^t)^2 (p_i^t)^2$ as:

$$32 \sum_{i \in [K]} \left(\gamma_i^{t+1} - \gamma_i^t\right)^2 \left(p_i^t \log p_i^t\right)^2$$

$$\leq 32 \sum_{i \in [K]} \frac{1}{\log T} \frac{\left(\max\left\{p_i^t, 1/T\right\}\right)^{-2}}{1 + \sum_{k=1}^t \left(\max\left\{p_i^k, 1/T\right\}\right)^{-1}} \left(p_i^t \log p_i^t\right)^2$$

$$\leq 32 \sum_{i \in [K]} \frac{1}{\log T} \left(\max\left\{p_i^t, 1/T\right\}\right)^{-1} \left(\max\left\{p_i^t, 1/T\right\} \log\left(\max\left\{p_i^t, 1/T\right\}\right)\right)^2$$

$$\leq 32 \sum_{i \in [K]} \frac{1}{\log T} \max\left\{p_i^t, 1/T\right\} \left(\log\left(\max\left\{p_i^t, 1/T\right\}\right)\right)^2$$

$$\leq 32 \sum_{i \in [K]} - \left(\max\left\{p_i^t, 1/T\right\}\right) \log\left(\max\left\{p_i^t, 1/T\right\}\right)$$

$$\leq 32 K \cdot \left(-\frac{\sum_{i \in [K]} \max\left\{p_i^t, 1/T\right\}}{K} \log\left(\frac{\sum_{i \in [K]} \max\left\{p_i^t, 1/T\right\}}{K}\right)\right)$$

$$\leq 64 \log K,$$

where the first step follows from Eq. (21) that $\gamma_i^{t+1} - \gamma_i^t \leq \frac{1}{\log T} \frac{\max\left\{p_i^t, 1/T\right\}^{-1}}{\gamma_i^{t+1}}$; the second step follwos from the fact that $0 \geq p_i^t \log\left(p_i^t\right) \geq \max\left\{p_i^t, 1/T\right\} \log\left(\max\left\{p_i^t, 1/T\right\}\right)$ as $x \log x$ is monotonically decreasing in $[0, 1/T]$; the forth step follows from the fact that $0 \leq -\log\left(\max\left\{p_i^t, 1/T\right\}\right) \leq \log T$; the fifth step utilizes the concavity of $-x \log x$.

Finally, following the same steps in Lemma C.3.2 finishes the proof of the multiplicative relation between $\bar{p}^{t+1}$ and $p^t$. $\qquad\square$

**Lemma C.3.4** (Multiplicative Relation for Log-barrier). *When using the log-barrier regularizer with $C_{\mathrm{LOG}} \geq 162$, $\alpha = 0$ and $\theta = \sqrt{\frac{1}{\log T}}$, Algorithm 1 guarantees*

$$\frac{1}{2} p_i^t \leq \bar{p}_i^{t+1} \leq 2 p_i^t, \quad \frac{1}{2} p_i^t \leq p_i^{t+1} \leq 2 p_i^t,$$

*for all $t \in [T]$ and arm $i \in [K]$.*

*Proof.* Similarly, we only need to verify that $162 \log K \geq 32 \sum_{i \in [K]} \left(\gamma_i^{t+1} - \gamma_i^t\right)^2 \psi'(p_i^t)^2 (p_i^t)^2$ as:

$$32 \sum_{i \in [K]} \left(\gamma_i^{t+1} - \gamma_i^t\right)^2 (p_i^t)^2 \psi'(p_i^t)^2$$

$$= 32 \sum_{i \in [K]} \left(\gamma_i^{t+1} - \gamma_i^t\right)^2 (p_i^t)^2 \left(\frac{1}{p_i^t}\right)^2$$

$$\leq 32 \sum_{i \in [K]} \frac{1}{(\log T)} \frac{(\max \{p_i^t, 1/T\})}{1 + \sum_{k=1}^{t} (\max \{p_i^k, 1/T\})}$$

$$\leq \frac{32}{\log T} \sum_{i \in [K]} (\max \{p_i^t, 1/T\})$$

$$\leq \frac{32}{\log T} \left( \sum_{i \in [K]} p_i^t + \frac{K}{T} \right) \leq 64,$$

where the first step follows from the definition of the log-barrier regularizer that $\psi'(p_i^t) = -\frac{1}{p_i^t}$; the second step applies Eq. (21). $\qquad\square$

**Lemma C.3.5.** *With $C_{\text{LOG}} \geq 162$, Algorithm 1 guarantees $\frac{1}{2} p_i^t \leq z_i \leq 2 p_i^t$ for any arm $i \in [K]$ and round $t \in [T]$ where $z \in \mathbb{R}_{\geq 0}^K$ is defined as*

$$z = \operatorname*{argmin}_{\substack{x_i = 0, \forall i \in U \\ x_i \geq 0, \forall i \in V \\ \sum_{i \in V} x_i = \sum_{i \in V} p_i^t}} \left\langle \sum_{\tau \leq t} \widehat{\ell_V^\tau}, x \right\rangle + \phi_V^t(x).$$

*Proof.* Clearly, $p_V^t$ is the solution of the following optimization problem:

$$p_V^t = \operatorname*{argmin}_{\substack{x_i = 0, \forall i \in U \\ x_i \geq 0, \forall i \in V \\ \sum_{i \in V} x_i = \sum_{i \in V} p_i^t}} \left\langle \sum_{\tau < t} \widehat{\ell_V^\tau}, x \right\rangle + \phi_V^t(x),$$

since $p_V^t$ satisfies all the KKT conditions.

Therefore, we are able to apply Lemma C.3.1 by setting $L = \sum_{\tau < t} \widehat{\ell_V^\tau}$, $L' = \sum_{\tau \leq t} \widehat{\ell_V^\tau}$, $\Omega$ being the corresponding simplex, $\gamma = \gamma' = \gamma^t$ where the loss estimators and learning rates are defined in Algorithm 1. As $\gamma' = \gamma$ and $C_{\text{LOG}} = 162 > 9$, we only need to verify that $C_{\text{LOG}} \geq 32 \sum_{i \in V} \left( \widehat{\ell_i^t} \right)^2 (p_i^t)^2$. By the definition of the importance weighted loss estimator $\widehat{\ell^t}$, we have

$$32 \sum_{i \in V} \left( \widehat{\ell_i^t} \right)^2 (p_i^t)^2 \leq 32 \sum_{i \in V} \mathbb{I}\{i^t = i\} \leq 32 \leq 162,$$

which finishes the proof. $\qquad\square$

## C.4 Generalization of [Ito, 2021, Lemma 21]

In this section, we greatly generalize [Ito, 2021, Lemma 21] which is critical to analyze the residual regret. The approach of Ito [2021] requires a closed form solution of the optimization problem from the FTRL framework, which is not always guaranteed. Our approach removes this constraint, and thus, it can be applied to $\beta$-Tsallis entropy regularizers when $\beta \neq 1/2$, and even more complicated regularizers such as those hybrid ones.

**Lemma C.4.1.** *Given $\Omega \subseteq \mathbb{R}$ and twice-differentiable functions $f, g : \Omega \to \mathbb{R}$, if their corresponding first-order derivatives $f', g'$ are invertible and the inverse functions $(f')^{-1}, (g')^{-1}$ are differentiable and convex, then, for any $p, q \in \Omega$ and $z \in \mathbb{R}$ satisfying $g'(q) = f'(p) - z$, we have*

$$q \geq p + \frac{1}{g''(p)} \cdot (f'(p) - g'(p) - z), \tag{76}$$

$$q \leq p + \frac{1}{g''(q)} \cdot (f'(p) - g'(p) - z), \tag{77}$$

$$q \geq p + \frac{1}{f''(p)} \cdot (f'(q) - g'(q) - z), \tag{78}$$

$$q \leq p + \frac{1}{f''(q)} \cdot (f'(q) - g'(q) - z). \tag{79}$$

*Proof.* We first show Eq. (76):

$$
\begin{aligned}
q &= (g')^{-1}\left(f'(p) - z\right) \\
&= (g')^{-1}\left(g'(p) + (f'(p) - g'(p)) - z\right) \\
&\geq p + \left[\left((g')^{-1}\right)'(x)\right]\bigg|_{x=g'(p)} \cdot (f'(p) - g'(p) - z) \\
&= p + \frac{1}{g''(p)} \cdot (f'(p) - g'(p) - z),
\end{aligned}
$$

where the third step follows from the convexity of $(g')^{-1}$, which ensures $(g')^{-1}(u) \geq (g')^{-1}(v) + \left((g')^{-1}\right)'(v) \cdot (u - v)$ for any $u, v$ in the domain of $(g')^{-1}$; the last step uses the inverse function theorem.

On the other hand, we have

$$
\begin{aligned}
p &= (g')^{-1}\left(g'(p)\right) \\
&= (g')^{-1}\left(g'(q) + g'(p) - g'(q)\right) \\
&= (g')^{-1}\left(g'(q) + g'(p) - f'(p) + f'(p) - g'(q)\right) \\
&= (g')^{-1}\left(g'(q) - (f'(p) - g'(p) - z)\right) \\
&\geq q - \left[\left((g')^{-1}\right)'(x)\right]\bigg|_{x=g'(q)} \cdot (f'(p) - g'(p) - z) \\
&= q - \frac{1}{g''(q)} \cdot (f'(p) - g'(p) - z),
\end{aligned}
$$

where the forth step uses $g'(q) = f'(p) - z$; the fifth step follows from the convexity of $(g')^{-1}$; the last step uses the inverse function theorem. Rearranging this inequality yields Eq. (77).

By swapping $p$ with $q$, $f$ with $g$, and flipping $z$ to $-z$, repeating the steps above yields that

$$
p \geq q + \frac{1}{f''(q)} \cdot (g'(q) - f'(q) + z),
$$

$$
p \leq q + \frac{1}{f''(p)} \cdot (g'(q) - f'(q) + z).
$$

Rearranging these inequalities finishes the proof of Eq. (79) and Eq. (78), respectively. $\qquad\square$

In the following, we show an application of Lemma C.4.1 in the FTRL framework.

**Lemma C.4.2.** *Let $\mathcal{I} \subseteq [K]$ be an index set. Suppose $\nabla \phi_{\mathcal{I}}^{t+1}(q) = \nabla \phi_{\mathcal{I}}^{t}(p) - \widehat{\ell}_{\mathcal{I}}^{t} + c \cdot \mathbf{1}_{\mathcal{I}}$ where $\phi^t$ is the regularizer with $\phi^t(x) = \sum_{i\in S} \gamma_i^t \psi(x_i)$, $\psi : \Omega \to \mathbb{R}$ for $\Omega \subseteq \mathbb{R}$ is a strictly convex function such that $\gamma_i^t \psi$ satisfies all conditions in Lemma C.4.1 for all $t, i$, $\widehat{\ell}_{\mathcal{I}}^{t}$ is the loss estimator, and $c \in \mathbb{R}$. If $p, q$ satisfy $\sum_{i\in\mathcal{I}} p_i = \sum_{i\in\mathcal{I}} q_i$, then we have:*

$$
c \leq \left(\sum_{i\in\mathcal{I}} \frac{1}{\gamma_i^{t+1}\psi''(p_i)}\right)^{-1} \left(\sum_{i\in\mathcal{I}} \frac{(\gamma_i^{t+1} - \gamma_i^t)\psi'(p_i) + \widehat{\ell}_i^t}{\gamma_i^{t+1}\psi''(p_i)}\right), \tag{80}
$$

$$
c \geq \left(\sum_{i\in\mathcal{I}} \frac{1}{\gamma_i^{t}\psi''(q_i)}\right)^{-1} \left(\sum_{i\in\mathcal{I}} \frac{(\gamma_i^{t+1} - \gamma_i^t)\psi'(q_i) + \widehat{\ell}_i^t}{\gamma_i^{t}\psi''(q_i)}\right). \tag{81}
$$

*Consequently, for $\nabla\phi_{\mathcal{I}}^{t}(q) = \nabla\phi_{\mathcal{I}}^{t}(p) - \widehat{\ell}_{\mathcal{I}}^{t} + c \cdot \mathbf{1}_{\mathcal{I}}$ where all definitions remain the same as above, we have*

$$
c \leq \left(\sum_{i\in\mathcal{I}} \frac{1}{\gamma_i^{t}\psi''(p_i)}\right)^{-1} \left(\sum_{i\in\mathcal{I}} \frac{\widehat{\ell}_i^t}{\gamma_i^{t}\psi''(p_i)}\right), \tag{82}
$$

$$c \geq \left( \sum_{i \in \mathcal{I}} \frac{1}{\gamma_i^t \psi''(q_i)} \right)^{-1} \left( \sum_{i \in \mathcal{I}} \frac{\widehat{\ell}_i^t}{\gamma_i^t \psi''(q_i)} \right). \tag{83}$$

*Proof.* By directly applying Eq. (76) in Lemma C.4.1 with $f(\cdot) = \gamma_i^t \psi(\cdot)$, $g(\cdot) = \gamma_i^{t+1} \psi(\cdot)$, $p = p_i$, $q = q_i$, and $z = \widehat{\ell}_i^t - c$, we have

$$q_i - p_i \geq \frac{\left( \gamma_i^t - \gamma_i^{t+1} \right) \psi'(p_i) - \widehat{\ell}_i^t + c}{\gamma_i^{t+1} \psi''(p_i)}. \tag{84}$$

Since $\sum_{i \in \mathcal{I}} p_i = \sum_{i \in \mathcal{I}} q_i$, we sum Eq. (84) over all $i \in \mathcal{I}$ and then rearrange to get Eq. (80). On the other hand, by applying Eq. (79) in Lemma C.4.1 with the same choices, we have

$$q_i - p_i \leq \frac{\left( \gamma_i^{t+1} - \gamma_i^t \right) \psi'(q_i) - \widehat{\ell}_i^t + c}{\gamma_i^t \psi''(q_i)}. \tag{85}$$

We again sum Eq. (85) over all $i \in \mathcal{I}$ and then rearrange to get Eq. (81). □

Lemma C.4.2 is applicable to the analysis of the Decoupled-Tsallis-INF algorithm. For the analysis of the MAB algorithms, since we add some extra log-barrier to the regularizers, we will need the following similar results.

**Lemma C.4.3.** *Under the same setup of Lemma C.4.2 and let $\theta \in \mathbb{R}_{>0}$ be given, if the regularizer is $\phi^t(x) = \sum_{i \in S} \gamma_i^t \psi(x_i) + \theta r(x_i)$ where $\psi(\cdot), r(\cdot)$ are strictly convex and twice differentiable, $\gamma_i^t \psi(\cdot) + \theta r(\cdot)$ satisfies all conditions in Lemma C.4.1 for all $t, i$, we have*

$$c \leq \left( \sum_{i \in \mathcal{I}} \frac{1}{\gamma_i^{t+1} \psi''(p_i) + \theta r''(p_i)} \right)^{-1} \left( \sum_{i \in \mathcal{I}} \frac{\left( \gamma^{t+1} - \gamma^t \right) \psi'(p_i) + \widehat{\ell}_i^t}{\gamma_i^{t+1} \psi''(p_i) + \theta r''(p_i)} \right),$$

$$c \geq \left( \sum_{i \in \mathcal{I}} \frac{1}{\gamma_i^t \psi''(q_i) + \theta r''(q_i)} \right)^{-1} \left( \sum_{i \in \mathcal{I}} \frac{\left( \gamma_i^{t+1} - \gamma_i^t \right) \psi'(q_i) + \widehat{\ell}_i^t}{\gamma_i^t \psi''(q_i) + \theta r''(q_i)} \right),$$

$$c \leq \left( \sum_{i \in \mathcal{I}} \frac{1}{\gamma_i^t \psi''(p_i) + \theta r''(p_i)} \right)^{-1} \left( \sum_{i \in \mathcal{I}} \frac{\left( \gamma^{t+1} - \gamma^t \right) \psi'(q_i) + \widehat{\ell}_i^t}{\gamma_i^t \psi''(p_i) + \theta r''(p_i)} \right),$$

$$c \geq \left( \sum_{i \in \mathcal{I}} \frac{1}{\gamma_i^{t+1} \psi''(q_i) + \theta r''(q_i)} \right)^{-1} \left( \sum_{i \in \mathcal{I}} \frac{\left( \gamma_i^{t+1} - \gamma_i^t \right) \psi'(p_i) + \widehat{\ell}_i^t}{\gamma_i^{t+1} \psi''(q_i) + \theta r''(q_i)} \right).$$

*Proof.* For any $i \in \mathcal{I}$, applying Eq. (76) in Lemma C.4.1 with $f = \gamma_i^{t+1} \psi(\cdot) + \theta r(\cdot)$, $g = \gamma_i^t \psi(\cdot) + \theta r(\cdot)$, $p = p_i$, $q = q_i$, and $z = \widehat{\ell}_i^t - c$ yields that

$$q_i - p_i \geq \frac{\gamma_i^t \psi'(p_i) + \theta r'(p_i) - \gamma_i^{t+1} \psi'(p_i) - \theta r'(p_i) - \widehat{\ell}_i^t + c}{\gamma_i^{t+1} \psi''(p_i) + \theta r''(p_i)} = \frac{\left( \gamma_i^t - \gamma_i^{t+1} \right) \psi'(p_i) - \widehat{\ell}_i^t + c}{\gamma_i^{t+1} \psi''(p_i) + \theta r''(p_i)}.$$

By the condition that $\sum_{i \in \mathcal{I}} p_i = \sum_{i \in \mathcal{I}} q_i$, taking the summation on both sides gives us:

$$0 \geq \sum_{i \in \mathcal{I}} \frac{\left( \gamma_i^t - \gamma_i^{t+1} \right) \psi'(p_i) - \widehat{\ell}_i^t + c}{\gamma_i^{t+1} \psi''(p_i) + \theta r''(p_i)}$$

$$\xrightarrow{\text{Rearranging}} c \cdot \sum_{i \in \mathcal{I}} \frac{1}{\gamma_i^{t+1} \psi''(p_i) + \theta r''(p_i)} \leq \sum_{i \in \mathcal{I}} \frac{\left( \gamma_i^{t+1} - \gamma_i^t \right) \psi'(p_i) + \widehat{\ell}_i^t}{\gamma_i^{t+1} \psi''(p_i) + \theta r''(p_i)},$$

which proves the first inequality. Similarly, the other inequalities in the statement can be obtained by applying Eq. (77), Eq. (78), and Eq. (79) in Lemma C.4.1. □

## C.5 Analysis of FTRL Framework and Bregman Divergence

**Lemma C.5.1** (Lemma 17 in Ito [2021])**.** *For any regularizers $\phi, \phi'$, and $L, L' \in \mathbb{R}^K$, denote $z, z'$ as*

$$z = \operatorname*{argmin}_{x \in \Omega} \langle L, x \rangle + \phi(x), \quad z' = \operatorname*{argmin}_{x \in \Omega} \langle L', x \rangle + \phi'(x),$$

*where $\Omega = \{x \in \mathcal{D} \mid Ax = b\}$ for some convex set $\mathcal{D}$ and $A \in \mathbb{R}^{K \times K}, b \in \mathbb{R}^K$. Define the divergences $F^1$ and $F^2$ as:*

$$F^1(x, y) = \phi(x) - \phi'(y) - \langle \nabla \phi'(y), x - y \rangle, \quad F^2(y, x) = \phi'(y) - \phi(x) - \langle \nabla \phi(x), y - x \rangle.$$

*We then have $F^1(z, z') + F^2(z', z) = -\langle L' - L, z' - z \rangle$.*

Since the following lemma only uses standard Bregman divergence and the analysis is for a given round $t$, we drop $t$ for $D^t(x, y)$ and use $D(x, y)$.

**Lemma C.5.2.** *For any $L, \ell \in \mathbb{R}^K$ and a Legendre regularizer $\phi$, let $p, q$ be the vectors such that*

$$p = \operatorname*{argmin}_{x \in \Omega_K} \langle x, L \rangle + \phi(x), \text{ and } q = \operatorname*{argmin}_{x \in \Omega_K} \langle x, L + \ell \rangle + \phi(x).$$

*We then have*

$$\langle p - z, \ell \rangle - D(z, p) \le \langle p - q, \ell \rangle - D(q, p), \forall z \in \Omega_K.$$

*Moreover, there exists an intermediate point $\xi = \eta \cdot p + (1 - \eta) \cdot q$ for some $\eta \in [0, 1]$, such that*

$$\langle p - q, \ell \rangle - D(q, p) \le 2 \|\ell\|_{\nabla^{-2} \phi(\xi)}^2.$$

*Proof.* Since $\phi$ is Legendre, we have $\nabla \phi(p) = -L + c \cdot \mathbf{1}_K$ for some $c \in \mathbb{R}$. Thus, it holds for any $z \in \Omega_K$ that

$$\begin{aligned}
&\langle p - z, \ell \rangle - D(z, p) \\
&= \langle p - z, \ell \rangle - \phi(z) + \phi(p) + \langle \nabla \phi(p), z - p \rangle \\
&= \langle p - z, \ell \rangle - \phi(z) + \phi(p) + \langle -L, z - p \rangle \\
&= \langle p, L + \ell \rangle + \phi(p) - \langle z, L + \ell \rangle - \phi(z) \\
&= \langle p, L \rangle + \phi(p) + \langle p, \ell \rangle - \langle z, L + \ell \rangle - \phi(z).
\end{aligned}$$

Since $q$ maximizes the last two terms, we have for any $z \in \Omega_K$,

$$\langle p - z, \ell \rangle - D(z, p) \le \langle p - q, \ell \rangle - D(q, p).$$

Let $F(x) = \langle x, L \rangle + \phi(x)$ and $G(x) = \langle x, L + \ell \rangle + \phi(x)$ so that $p$ is the minimizer of $F(x)$ and $q$ is the minimizer of $G(x)$. By $\nabla \phi(p) = -L + c \cdot \mathbf{1}_K$ and direct calculation, we have

$$\begin{aligned}
\langle p - q, \ell \rangle - D(q, p) &= F(p) + \langle p, \ell \rangle - G(q) = G(p) - G(q) \\
&= \langle p - q, \nabla G(q) \rangle + \frac{1}{2} \|p - q\|_{\nabla^2 G(\xi)}^2 \\
&\ge \frac{1}{2} \|p - q\|_{\nabla^2 \phi(\xi)}^2,
\end{aligned}$$

where we apply Taylor's expansion with $\xi = \eta \cdot p + (1 - \eta) \cdot q$ for some $\eta \in [0, 1]$ being an intermediate point between $p$ and $q$, followed by the first-order optimality condition. On the other hand, we have

$$\begin{aligned}
\langle p - q, \ell \rangle - D(q, p) &= \langle p - q, \ell \rangle + F(p) - F(q) \\
&\le \langle p - q, \ell \rangle \\
&\le \|\ell\|_{\nabla^{-2} \phi(\xi)} \|p - q\|_{\nabla^2 \phi(\xi)},
\end{aligned}$$

where the second step follows from the optimality of $p$, and the last step applies Hölder's inequality. Combining these two inequalities gives us that

$$\langle p - q, \ell \rangle - D(q, p) \le 2 \|\ell\|_{\nabla^{-2} \phi(\xi)}^2,$$

for some $\eta \in [0, 1]$ such that $\xi = \eta \cdot p + (1 - \eta) \cdot q$. $\qquad\square$

**Lemma C.5.3.** *Consider the following regularizer for any $\beta \in (0,1)$ and learning rate $\gamma_i > 0$:*

$$\phi(x) = -\frac{1}{1-\beta} \sum_{i \in [K]} \gamma_i x_i^{\beta}.$$

*For any $q, x \in \mathbb{R}_{\geq 0}^K$ and $\ell \in \mathbb{R}^K$ satisfying $\frac{(q_i)^{1-\beta}\ell_i}{\gamma_i} \geq \frac{\beta}{1-\beta}\left(e^{\frac{\beta-1}{\beta}} - 1\right)$ for all arm $i$, we have*

$$\langle q - x, \ell \rangle - D(x, q) \leq \sum_{i \in [K]} \frac{(q_i)^{2-\beta}}{\beta\gamma_i} (\ell_i)^2.$$

*Proof.* Let $w = \operatorname{argmin}_{x \in \mathbb{R}_{>0}^K} \langle q - x, \ell \rangle - D(x, q)$ be the maximizer. By setting the gradient to zero, we have $-\ell - \nabla\phi(w) + \nabla\phi(q) = 0$, which equivalently implies that for any arm $i \in [K]$,

$$\frac{1}{(w_i)^{1-\beta}} = \frac{1}{(q_i)^{1-\beta}} + \frac{1-\beta}{\gamma_i \beta} \ell_i. \tag{86}$$

By direct calculation, we have

$$
\begin{aligned}
&\langle q - w, \ell \rangle - D(w, q) \\
&= \phi(q) - \phi(w) - \langle \nabla\phi(w), q - w \rangle \\
&= \frac{1}{1-\beta} \sum_{i \in [K]} \gamma_i \left( (w_i)^{\beta} - (q_i)^{\beta} + \beta (w_i)^{\beta-1} (q_i - w_i) \right) \\
&= \frac{1}{1-\beta} \sum_{i \in [K]} \gamma_i \left( (1-\beta)(w_i)^{\beta} - (1-\beta)(q_i)^{\beta} + \beta \left( (w_i)^{\beta-1} - (q_i)^{\beta-1} \right) q_i \right) \\
&= \frac{1}{1-\beta} \sum_{i \in [K]} \gamma_i \left( (1-\beta)(w_i)^{\beta} - (1-\beta)(q_i)^{\beta} + \frac{1-\beta}{\gamma_i} \ell_i q_i \right) \\
&= \sum_{i \in [K]} \gamma_i \left( (w_i)^{\beta} - (q_i)^{\beta} + \frac{\ell_i q_i}{\gamma_i} \right),
\end{aligned} \tag{87}
$$

where the first step uses $-\ell - \nabla\phi(w) + \nabla\phi(q) = 0$, and the fifth step uses Eq. (86).

Moreover, we have for any arm $i$

$$
\begin{aligned}
(w_i)^{\beta} &= (q_i)^{\beta} \left( \frac{(q_i)^{1-\beta}}{(w_i)^{1-\beta}} \right)^{\frac{\beta}{\beta-1}} \\
&= (q_i)^{\beta} \left( 1 + \frac{1-\beta}{\gamma_i \beta} (q_i)^{1-\beta} \ell_i \right)^{\frac{\beta}{\beta-1}} \\
&\leq (q_i)^{\beta} \left( 1 - \frac{1}{\gamma_i} (q_i)^{1-\beta} \ell_i + \frac{1}{\beta(\gamma_i)^2} (q_i)^{2-2\beta} (\ell_i)^2 \right) \\
&= (q_i)^{\beta} - \frac{1}{\gamma_i} q_i \ell_i + \frac{1}{\beta(\gamma_i)^2} (q_i)^{2-\beta} (\ell_i)^2,
\end{aligned} \tag{88}
$$

where the second step uses Eq. (86), and the third step follows from the fact that $(1+z)^{\alpha} < 1 + \alpha z + \alpha(\alpha-1)z^2$ for $\alpha < 0$ and $z \geq \exp\left(\frac{1}{\alpha}\right) - 1$ with $\alpha = \frac{\beta}{\beta-1} = 1 - \frac{1}{1-\beta} < 0$ for $\beta \in (0,1)$ and $z = \frac{1-\beta}{\gamma_i \beta} (q_i)^{1-\beta} \ell_i \geq e^{\frac{\beta-1}{\beta}} - 1 = \exp\left(\frac{1}{\alpha}\right) - 1$.

Plugging Eq. (88) into Eq. (87) yields

$$\langle q - w, \ell \rangle - D(w, q) \leq \sum_{i \in [K]} \gamma_i \left( \frac{\ell_i q_i}{\gamma_i} - \frac{1}{\gamma_i} q_i \ell_i + \frac{1}{\beta(\gamma_i)^2} (q_i)^{2-\beta} (\ell_i)^2 \right) = \sum_{i \in [K]} \frac{(q_i)^{2-\beta}}{\beta\gamma_i} (\ell_i)^2,$$

which concludes the proof. $\square$

