# OpenReview forum: "Improved Best-of-Both-Worlds Guarantees for Multi-Armed Bandits: FTRL with General Regularizers and Multiple Optimal Arms"
_NeurIPS.cc/2023/Conference — NeurIPS 2023 poster_

### Official Review · Reviewer_36bt · 2023-06-28

**Soundness:** 3 good
**Presentation:** 3 good
**Contribution:** 2 fair
**Rating:** 6
**Confidence:** 4

**Summary:**

The paper introduces a new algorithm for multi-armed bandit problems, leveraging the FTRL framework and the flexible $\beta$-Tsallis entropy family of regularizers, where $\beta \in [0,1]$. This algorithm firstly uses a new learning rate schedule to offer best-of-both-worlds guarantees for a wide range of regularization parameters. Secondly, it eliminates the requirement of the uniqueness assumption of the optimal arm. Thirdly, it improves the stochastic bounds for Shannon entropy and Log-barrier regularization.

**Strengths:**

- The paper introduces new elegant learning rates for $\beta$-Tsallis entropy, providing a best-of-both-worlds guarantee.
- It generalizes the regret analysis approach by Ito (2021) by removing the uniqueness assumption of the optimal arm.
- As $\beta$-Tsallis entropy is an important regularization in bandit algorithm, the results could be useful in other settings too.


**Weaknesses:**

- The results presented in the paper for the plain multi-armed bandit, in my view, lack interest and significance compared to the algorithmic and analysis novelties. Specifically, there is no improvement in terms of regret bounds in the plain multi-armed bandit problem, as FTRL with $1/2$-Tsallis regularizer (1/2-Tsallis-INF algorithm) already achieves the optimal bound in both adversarial and stochastic regimes, and the uniqueness assumption has already been addressed by Ito (2021). The only potential value lies in applying the same ideas to other settings, such as the decoupled exploration and exploitation problem explored, as discussed in the paper.

- The bounds in intermediate regimes between stochastic and adversarial, where $C \neq 0$ seem to be suboptimal as they do not interpolate well between the optimal bounds of the two regime. Read question 2 and 3 for further clarification of this issue.

- There are few undefined notations used in the analysis. For instance, Equation (6) introduces the notation $D_U$ without providing a clear definition, and the same issue applies to $\phi_U(x)$ and $\phi_V(x)$ in Equation (7). If the used notation for this part were consistent with Ito (2021), $\phi_U(x)$ must accept $x$ from $\mathbb{R}^{|U|}$ but in Equation (7) it takes $x \in \mathbb{R}^{K}$. This inconsistency requires revision for clarity and accuracy.


**Questions:**

1- As stated in the paper, the regularizer always includes a small amount of extra log-barrier. Typically, the use of log-barrier introduces an additional multiplicative factor of $\sqrt{\log T}$ in the adversarial bound. However, your adversarial bound in Theorem 3.3 does not reflect this when $\beta = 1/2$. Can you elaborate on this?

2- Regarding the results on regime so called adversarial regime with a self-bounding constraint, In lines 183 and 184, you claim that your bound smoothly interpolates between $\log T$ and $\sqrt{T \log T}$ as $C$ ranges from $0$ to $T$. However, the presence of $D$ in your bound raises doubts about this smooth interpolation since $D$ can be as large as $C$, and $C$ can be on the order of $\mathcal{O}(T)$. This essentially shows that your bound has no robustness to corruption. Could you address this inconsistency?

3- Following up on the previous question, there is an improvement in the analysis of the $1/2$-Tsallis-INF algorithm by Masoudian and Seldin (2021), which demonstrates smooth interpolation between the two optimal bounds, $\mathcal{O}(\sum_{i \neq i^*} \frac1{\Delta_i}\log T)$ and $\mathcal{O}(\sqrt{KT})$ as $C$ increases. Is there any way to achieve a similar improvement for your algorithm? If not, discuss the challenges involved in obtaining the same improvement for intermediate regimes?

4- In the contribution section, for log-barrier regularization, the authors claim to have removed the uniqueness assumption of the optimal arm utilized in Ito (2021) and improved the stochastic bound. However, it should be noted that the algorithm proposed by Ito does not align completely with the algorithm presented in this paper, as Ito employs optimistic follow the regularized leader with different learning rates. Please provide further clarification.


References:

- Shinji Ito, Parameter-Free Multi-Armed Bandit Algorithms with Hybrid Data-Dependent Regret Bounds, COLT 2021

- Saeed Masoudian and Yevgeny Seldin, Improved analysis of the tsallis-inf algorithm in stochastically constrained adversarial bandits and stochastic bandits with adversarial corruptions, COLT 2021

---

> ### Author Rebuttal · Authors · 2023-08-10
>
> Thanks for your valuable feedback. Please see our responses below:
>
> ***
> **Q1:** There are few undefined notations used in the analysis. For instance, Equation (6) introduces the notation $D_U$
>  without providing a clear definition, and the same issue applies to $\phi_U(x)$
>  and $\phi_V(x)$
>  in Equation (7). If the used notation for this part were consistent with [Ito, 2021], $\phi_U(x)$
>  must accept $x$
>  from $\mathbb{R}^{|U|}$
>  but in Equation (7) it takes $x \in \mathbb{R}^K$
> . This inconsistency requires revision for clarity and accuracy.
>
> **A1:** The definitions of $D_\mathcal{I}$ and $\phi_\mathcal{I}$ for any subset $\mathcal{I} \subseteq [K]$ are provided in lines 253-254, and they indeed take $x \in \mathbb{R}^K$ as input.
>
> For your convenience, we also copy the definitions here: For any subset $\mathcal{I} \subseteq [K]$, we define $D^{s,t}_I(x,y) =  \phi_I (x) -\phi_I^t (y) - \langle \nabla  \phi^t_I(y), x-y \rangle$,
>
> where $\phi^t_{\mathcal{I}} (x)= -C_{log}\sum_{i\in\mathcal{I}} \log x_i - \frac{1}{1-\beta} \sum_{i \in \mathcal{I}} \gamma^{t}_i x_i^{\beta}$ (that is, $\phi^t$ restricted to $\mathcal{I}$). We will certainly improve the readability in the future version.
>
> ***
> **Q2:** As stated in the paper, the regularizer always includes a small amount of extra log-barrier. Typically, the use of log-barrier introduces an additional multiplicative factor of $\sqrt{\log T}$ in the adversarial bound. However, your adversarial bound in Theorem 3.3 does not reflect this when $\beta=1/2$. Can you elaborate on this?
>
> **A2:** In fact, a small (here, small means constant) amount of extra log-barrier usually only introduces an additive $K\log T$ term to the regret (which we omit in the bound for simplicity), but not a multiplicative factor of $\sqrt{\log T}$.
> The multiplicative factor of $\sqrt{\log T}$ in our bound for $\beta \neq 1/2$ comes from the specific learning rate that we propose, and this factor does not show up for $\beta = 1/2$ because in that case we simply use the arm-independent learning rate $\gamma_i^t = \Theta(\sqrt{t})$ of [31].
>
> ***
> **Q3:** Regarding the results on regime so called adversarial regime with a self-bounding constraint, In lines 183 and 184, you claim that your bound smoothly interpolates between
> $\log T$ and $\sqrt{T\log T}$
> as $C$
> ranges from $0$
> to $T$. However, the presence of $D$
> in your bound raises doubts about this smooth interpolation since $D$
> can be as large as $C$, and
> can be on the order of $O(T)$. This essentially shows that your bound has no robustness to corruption. Could you address this inconsistency?
>
> **A3:** You are right, and we will revise our statement. However, note that the same $D$ dependence also appears in [Ito, 2021]. Whether this can be removed in the absence of the uniqueness assumption is indeed an important question.
>
> ***
> **Q4:** Following up on the previous question, there is an improvement in the analysis of the
> $1/2$-Tsallis-INF algorithm by Masoudian and Seldin (2021), which demonstrates smooth interpolation between the two optimal bounds, $O(\sum_{i\neq i^\star}\log T/\Delta_i)$ and $\sqrt{KT}$ as $C$ increases. Is there any way to achieve a similar improvement for your algorithm? If not, discuss the challenges involved in obtaining the same improvement for intermediate regimes?
>
> **A4:**
> We do not think the technique in Masoudian and Seldin (2021) is helpful here, since this issue only shows up when we remove the uniqueness assumption (while Masoudian and Seldin still make this assumption).
> Specifically, recall that to use the self-bounding constraint, we typically rewrite the regret as $\text{Reg}^T = (1+\lambda)\text{Reg}^T-\lambda \text{Reg}^T$ for any $\lambda>0$. The term $(1+\lambda)D$ then shows up when decomposing the regret in the first term if we do not have uniqueness.
>
>
> ***
> **Q5:** In the contribution section, for log-barrier regularization, the authors claim to have removed the uniqueness assumption of the optimal arm utilized in [Ito, 2021] and improved the stochastic bound. However, it should be noted that the algorithm proposed by Ito does not align completely with the algorithm presented in this paper, as Ito employs optimistic follow the regularized leader with different learning rates. Please provide further clarification.
>
>
> **A5:**
> We will clarify this in the final version. In fact, if one removes the hint vector  in [Ito, 2021] (which is used to obtain data-dependent bounds) and set $\nu^t_i=p^t_i$ in his algorithm, our techniques can also remove the uniqueness assumption for this algorithm.
>
> ***
> [Ito, 2021] Shinji Ito. Parameter-free multi-armed bandit algorithms with hybrid data-dependent regret bounds. In Proceedings of Thirty Fourth Conference on Learning Theory, 2021.

---

> > ### Comment · Reviewer_36bt · 2023-08-15
> >
> > Thank the authors for addressing my questions. I don't have further questions and will keep my score as is.

---

### Official Review · Reviewer_aiz3 · 2023-07-05

**Soundness:** 4 excellent
**Presentation:** 3 good
**Contribution:** 3 good
**Rating:** 7
**Confidence:** 5

**Summary:**

The authors focus on best-of-both-worlds (BOBW) algorithms based on follow-the-regularized-leader (FTRL) in multi-armed bandits.
The theoretical guarantees for most existing FTRL-based BOBW algorithms were based on the assumption that the best arm is unique in order to take advantage of self-bounding techniques.
It is known that this assumption can be removed by the paper in [13], but its analysis was only applicable to the case of Tsallis-INF (FTRL with 1/2-Tsallis entropy), one of the most representative BOBW algorithms.
Extending the analysis of [13], the authors show that a BOBW guarantee can be obtained with FTRL with more general regularizers, i.e., negative Shannon entropy, log-barrier, and FTRL with $\beta$-Tsallis entropy, without the assumption of an unique optimal arm.
Furthermore, by using the new theory, the authors improve the regret upper bound in the stochastic regime in the decoupled setting.

**Strengths:**

- The paper is very well organized and well written.
- The paper greatly advances the theory of [13], excluding the unique optimal arm assumption for a wide range of typical regularizers. The assumption have been employed for constructing BOBW algorithms with FTRL, and this is an interesting and important technical contribution to the community.
- In addition, the authors affirmatively answer the question of whether it is possible to achieve BOBW without knowing $\Delta_{\min}$ when $\beta$-Tsallis entropy with $\beta \neq 1/2$ (which was unresolved in Zimmert and Seldin [31]), and the question of whether it is possible to achieve BOBW with Shannon entropy using ${\Delta_{\min}}$ to $\Delta_i$-wise in the stochastic setting (unresolved in Ito et al. [14]). The both of contributions are interesting and important. The related points are listed in weakness.

**Weaknesses:**

- There do not appear to be any major weaknesses in this paper.
- One weakness would be a discussion of whether removing the assumption of unique optimal arm actually improves or worsens the performance of algorithms (The reviewer expects the algorithm becomes more conservative, and the performance becomes worse.)
- Since the discussion excluding the assumption of unique optimal arm is cumbersome on its own, it would be desirable to have a discussion of which techniques in the paper contributed to resolving the problems. More specifically, which components of the algorithm play an important role in resolving the problems of [31] and [14], which are mentioned in the above Strengths part? In addition, if we accept the assumption of unique optimal arm, can we achieve improvements in [31] and [14] with a much more similar argument?

Minor issues and typos:
- line 87: Sepcifically -> Specifically

**Questions:**

- Shannon entropy relies on $(\log T)^2$ rather than $\log T$, which is the case for all published algorithms using FTRL with Shannon entropy, but do the authors think it is possible to make it to $\log T$?
- In addition to the above, the reviewer expect the authors to answer the questions listed in Weaknesses.
- Along with them, the reviewer hopes that the author will address the questions that have been pointed out in the "Weaknesses" section above.

---

> ### Author Rebuttal · Authors · 2023-08-10
>
>
>
>
> ***
> **Q1:** One weakness would be a discussion of whether removing the assumption of unique optimal arm actually improves or worsens the performance of algorithms (The reviewer expects the algorithm becomes more conservative, and the performance becomes worse.)
>
> **A1:**
> We in fact do not believe this would be the case. For example, Ito [2021] removes the uniqueness requirement without changing the algorithm at all (i.e., this is merely an improvement in the analysis).
> While our results unfortunately require making slight algorithmic modifications (such as adding a log-barrier regularizer) and also suffer an extra $\frac{|U|\log T}{\Delta_{\min}}$ term in the regret, we believe that these are all artifacts of our analysis.
>
>
> ***
> **Q2(a):** Since the discussion excluding the assumption of unique optimal arm is cumbersome on its own, it would be desirable to have a discussion of which techniques in the paper contributed to resolving the problems. More specifically, which components of the algorithm play an important role in resolving the problems of [31] and [14], which are mentioned in the above Strengths part?
>
> **A2(a):** Thanks for your suggestion, and we will add such  a discussion to the final version.
> From an algorithmic perspective, our novel learning rate schedule plays the most important role.
> On the other hand, the novelty of our analysis has been highlighted in Sec 4, starting L263 in particular.
>
> ***
> **Q2(b):** In addition, if we accept the assumption of unique optimal arm, can we achieve improvements in [31] and [14] with a much more similar argument?
>
> **A2(b):** We do not think we can ``improve'' their results under the uniqueness assumption (especially for [31] since it is already optimal), but for the standard MAB problem considered in [31], we can at least recover their result, and the analysis indeed become much simpler. On the other hand, it is currently unclear to us whether the same holds for the case with graph feedback that is considered in [14].
>
> ***
> **Q3:** Minor issues and typos: line 87, Sepcifically -> Specifically.
>
>
> **A3:** Thank you for spotting the typo. It will be fixed in the final version.
>
>
> ***
> **Q4:** Shannon entropy relies on $(\log T)^2$ rather than $\log T$, which is the case for all published algorithms using FTRL with Shannon entropy, but do the authors think it is possible to make it to $\log T$?
>
> **A4:** It is unclear to us at this point whether $\log T$ is achievable, but we point out that it is possible to improve $(\log T)^2$ to $\log T \log K$ in the MAB setting as shown in [Dann et al., 2023].
>
> ***
> [Ito, 2021] Shinji Ito. Parameter-free multi-armed bandit algorithms with hybrid data-dependent regret bounds. In Proceedings of Thirty Fourth Conference on Learning Theory, 2021.
>
> [Dann et al., 2023] Chris Dann, Chen-Yu Wei, Julian Zimmert, A blackbox approach to best of both worlds in bandits and beyond. In Proceedings of Thirty Sixth Conference on Learning Theory, 2023.

---

> > ### Comment · Reviewer_aiz3 · 2023-08-12
> > **Response confirmed**
> >
> > Thank you for your response.
> > All questions have been answered.
> > The rating will remain the same.

---

### Official Review · Reviewer_VdeD · 2023-07-06

**Soundness:** 3 good
**Presentation:** 3 good
**Contribution:** 3 good
**Rating:** 6
**Confidence:** 3

**Summary:**

This paper studies the problem of designing adaptive multi-armed bandit algorithms that perform optimally in both the stochastic setting and the adversarial setting simultaneously (often known as a best-of-both-world guarantee). The authors show that the uniqueness assumption is unnecessary for FTRL with a broad family of regularizers and a new learning rate schedule. For some regularizers, their regret bounds also improve upon prior results even when uniqueness holds.

**Strengths:**

1.	The considered problem, i.e., best-of-both-world for multi-armed bandit, is important in the bandit literature.
2.	The theoretical analysis looks sound, and the improvement is significant.
3.	This paper is well-written and clearly organized.


**Weaknesses:**

1.	This paper does not provide any experimental result. It would improve the paper if the authors could conduct empirical evaluation for their algorithms and compare to existing BOBW algorithms, to validate their theoretical results.

**Questions:**

Please see the weaknesses above.

**Limitations:**

Please see the weaknesses above.

---

> ### Author Rebuttal · Authors · 2023-08-10
>
> ***
> **Q1:**  This paper does not provide any experimental result. It would improve the paper if the authors could conduct empirical evaluation for their algorithms and compare to existing BOBW algorithms, to validate their theoretical results.
>
> **A1:**  We thank the reviewer for this suggestion. As in most previous work along this line (e.g. the closest one by Ito [2021]), our work focuses only on the theoretical sides. We do think empirical evaluations would be interesting and plan to do so in the future though.
>
> ***
> [Ito, 2021] Shinji Ito. Parameter-free multi-armed bandit algorithms with hybrid data-dependent regret bounds. In Proceedings of Thirty Fourth Conference on Learning Theory, 2021.

---

> > ### Comment · Reviewer_VdeD · 2023-08-15
> > **Thank the authors for their response**
> >
> > Thank the authors for their response. This paper can be improved by including experiments. I tend to keep my score.

---

### Official Review · Reviewer_toae · 2023-07-06

**Soundness:** 3 good
**Presentation:** 4 excellent
**Contribution:** 3 good
**Rating:** 6
**Confidence:** 3

**Summary:**

This paper considers the problem of proving best of both worlds guarantees for algorithms based on the FTRL framework for the multi-armed bandits problem. While it has been demonstrated in (Zimmert and Seldin (2019,2021)) that Tsallis-INF (FTRL with the $1/2$-Tsallis entropy regularizer) achieves optimal regret in both the adversarial and stochastic settings simultaneously, their analysis for the stochastic case relied on the assumption that the optimal arm is unique. The more recent work of Ito (2021) showed that Tsallis-INF still enjoys $\\log T$ regret in the stochastic case even if the optimal arm is not unique. In this paper, the authors generalize the analysis of Ito (2021) to other regularizers. Namely, they prove, without the uniqueness assumption, best-of-both-worlds guarantees for FTRL with any $\\beta$-Tsallis regularizer (including the log barrier and the Shannon entropy regularizers) using a new arm-dependent learning rate, albeit all regularizers are mixed with the log barrier for technical reasons.

**Strengths:**

- This work provides best of both worlds guarantees without the unique optimal arm assumption for FTRL with a broad family of regularizers. While the use of the $1/2$-Tsallis regularizer is the optimal choice (and already analyzed by Ito (2021)) for the standard bandits problem, other choices are still useful in closely related problems as illustrated in the decoupled exploration and exploitation problem.
- Moreover, this work seems to be the first to provide BOBW guarantees (without requiring prior knowledge of the suboptimality gaps) for the $\\beta$-Tsallis regularizer when $\\beta$ is not $1/2$.
- Overall, the paper is well written and the presentation is clear. A concise sketch of the analysis technique is provided in the last section, and the proofs seem mostly well written and easy to follow.

**Weaknesses:**

- Unlike Ito(2021), the provided bounds include an added term of order $|U| \\log(T) / \\Delta_\\min$ where $U$ is the set of optimal arms and $\\Delta_\\min$ is the smallest sub-optimality gap. Thus, the bounds are negatively affected when there are many optimal arms. While this is still an improvement in cases where prior works only achieved a $K \\log(T) / \\Delta_\\min$ dependence (as in the Shannon entropy case), in other cases (most notably for the $1/2$-Tsallis regularizer analyzed by Ito (2021) without the uniqueness assumption) the provided results are inferior to prior works.
- The fact that all regularizers are summed with a log barrier term is a little unsatisfactory. For instance, in the Shannon entropy case, we potentially lose the appealing property of having closed form expressions for the predictions of FTRL.
- Though probably curable with a doubling trick, the fact that the proposed approach sometimes requires prior knowledge of the time horizon is a minor weakness.

**Questions:**

- It seems that the log barrier was not added to the Tsallis regularizer in the decoupled exploration and exploitation problem, was it to avoid the $K \\log T$ term? Why did the analysis go through in this case but not in the standard bandits problem?

**Limitations:**

The authors did address some of the limitations of their work. Notably the fact that their analysis requires adding a log barrier term to all the considered regularizers. The authors also acknowledged, though a little less explicitly, the extraneous dependence of their bounds on the number of optimal arms.

---

> ### Author Rebuttal · Authors · 2023-08-09
>
> Thanks for your valuable feedback. Please see our responses below:
>
> ***
> **Q1:**  Unlike Ito[2021], the provided bounds include an added term of order $\frac{|U|\log T}{\Delta_{\min}}$
>  where $U$
>  is the set of optimal arms and $\Delta_{\min}$
>  is the smallest sub-optimality gap. Thus, the bounds are negatively affected when there are many optimal arms. While this is still an improvement in cases where prior works only achieved a $\frac{K\log T}{\Delta_{\min}}$
>  dependence (as in the Shannon entropy case), in other cases (most notably for the
> $1/2$-Tsallis regularizer analyzed by Ito [2021] without the uniqueness assumption) the provided results are inferior to prior works.
>
>
> **A1:**
> We agree that our bounds are negatively affected when there are many optimal arms due to an extra $\frac{|U|\log T}{\Delta_{\min}}$ term. However, it is worth noting that when using $1/2$-Tsallis entropy regularizer, our result in fact can recover those of Ito \[2021\](that is, without paying this extra $\frac{|U|\log T}{\Delta_{\min}}$ term).
> We did not write this down explicitly only because we wanted to unify all cases in a concise way, but this can be verified by bounding Eq. (12) using the argument immediately below that equation, instead of the more complicated one we provide that handles general $\beta$.
> Therefore, our result is only more general than previous ones, and importantly is the only one that achieves the best of both worlds guarantee without the uniqueness assumption for other regularizers.
>
>
> ***
> **Q2:**  The fact that all regularizers are summed with a log barrier term is a little unsatisfactory. For instance, in the Shannon entropy case, we potentially lose the appealing property of having closed form expressions for the predictions of FTRL.
>
>
> **A2:**  Indeed, the extra log barrier is not ideal.
> Removing it is an interesting but also challenging direction, which we plan to work on in the future.
>
> ***
> **Q3:**  Though probably curable with a doubling trick, the fact that the proposed approach sometimes requires prior knowledge of the time horizon is a minor weakness.
>
> **A3:**
> Indeed, this is curable via a doubling trick (without hurting any of our regret bounds), as already mentioned in Section 5.3 of Ito [2021].
> More specifically, we divide all the rounds $1, 2, \ldots$ into segments $\\{C_k\\}_{k=1}^{\infty}$ where $C_k=\\{S_k+1,S_k+2,\ldots,S_k+T_k\\}$ with $T_k=2^{2^k}$ and $S_k=\sum_1^{k-1}T_h$, and run a new instance of our algorithm parameterized with $T_k$ for the rounds in $C_k$.
> The analysis is then similar to Ito [2021].
>
> ***
> **Q4:** It seems that the log barrier was not added to the Tsallis regularizer in the decoupled exploration and exploitation problem, was it to avoid the $K\log T$ term? Why did the analysis go through in this case but not in the standard bandits problem?
>
> **A4:**
> This is because unlike the MAB problem, an arm-independent learning rate is used in this case (note that $\alpha$ is set to $1/2$ and thus $\gamma_i^t$ is simply $\theta\sqrt{t}$).
> With such an arm-independent learning rate, we can show the desired stability without the help of an extra log-barrier regularizer.
> As you mentioned, this avoids paying the $K\log T$ term.
>
> ***
> [Ito, 2021] Shinji Ito. Parameter-free multi-armed bandit algorithms with hybrid data-dependent regret bounds. In Proceedings of Thirty Fourth Conference on Learning Theory, 2021.

---

> > ### Comment · Reviewer_toae · 2023-08-17
> >
> > Thank you for your response. My opinion remains generally the same: though the obtained bounds do not feature the ideal dependence on the suboptimality gaps, this work still offers a solid contribution towards a better understanding of the BOBW performance of FTRL-based algorithms, both in lifting the uniqueness assumption and providing improved bounds for some regularizers (with the impact of these contributions partly hinging upon their applicability beyond the standard bandits problem, as illustrated in one case by the authors for the DEE problem).

---

### Decision · Program_Chairs · 2023-09-21

**Decision:**

Accept (poster)

**Comment:**

This paper continues a long line of work on best-of-both-worlds bandit algorithms. The contribution is purely theoretical, extending the guarantees from prior work to a (much) broader family of algorithms and relaxing some assumptions. The reviewers and this AC are in consensus that this is a useful and interesting contribution (although not a spectacular one).  We think it is worth publishing and definitely passes the bar for NeurIPS.